# Implementation of a comprehensive ice crystal formation parameterization for cirrus and mixed-phase clouds into the EMAC model (based on MESSy 2.53)

Sara Bacer[1], Sylvia C. Sullivan[2], Vlassis A. Karydis[1,*], Donifan Barahona[3], Martina Krämer[4], Athanasios Nenes[2,5,6,7,8], Holger Tost[9], Alexandra P. Tsimpidi[1], Jos Lelieveld[1,10], and Andrea Pozzer[1]

[1]Atmospheric Chemistry Department, Max Planck Institute for Chemistry, Mainz, Germany
[2]School of Chemical and Biomolecular Engineering, Georgia Institute of Technology, Atlanta, USA
[3]NASA Goddard Space Flight Center, Greenbelt, USA
[4]Institute for Energy and Climate Research – 7, Research Center Jülich, Jülich, Germany
[5]School of Earth and Atmospheric Sciences, Georgia Institute of Technology, Atlanta, USA
[6]ICE-HT, Foundation for Research and Technology, Hellas, Greece
[7]IERSD, National Observatory of Athens, Athens, Greece
[8]Laboratory of Atmospheric Processes and Their Impacts, École Polytechnique Fédérale de Lausanne, Lausanne, Switzerland
[9]Institute for Atmospheric Physics, Johannes Gutenberg University Mainz, Mainz, Germany
[10]Energy, Environment and Water Research Center, The Cyprus Institute, Nicosia, Cyprus
*now at: Institute for Energy and Climate Research – 8, Research Center Jülich, Jülich, Germany

Correspondence: Sara Bacer (sara.bacer@mpic.de)

**Abstract.** A comprehensive ice nucleation parameterization has been implemented in the global chemistry-climate model EMAC to improve the representation of ice crystal number concentrations (ICNCs). The parameterization of Barahona and Nenes (2009, hereafter BN09) allows the treatment of ice nucleation taking into account the competition for water vapour between homogeneous and heterogeneous nucleation in cirrus clouds. Furthermore, the influence of chemically-heterogeneous, polydisperse aerosols is considered by applying one of the multiple ice nucleating particle parameterizations which are included in BN09 to compute the heterogeneously formed ice crystals. BN09 has been modified in order to consider the pre-existing ice crystal effect and implemented to operate both in the cirrus and in the mixed-phase regimes. Compared to the standard EMAC parameterizations, BN09 produces fewer ice crystals in the upper troposphere but higher ICNCs in the middle troposphere, especially in the Northern Hemisphere where ice nucleating mineral dust particles are relatively abundant. Overall, ICNCs agree well with the observations, especially in cold cirrus clouds (at temperatures below 205 K), although they are underestimated between 200 K and 220 K. As BN09 takes into account processes which were previously neglected by the standard version of the model, it is recommended for future EMAC simulations.

## 1 Introduction

Clouds play an important role in the Earth System by affecting the global radiative energy budget, the hydrologic cycle, the scavenging of gaseous and particulate substances, and by providing a medium for aqueous-phase chemical reactions. Nevertheless, clouds remain one of the less understood components of the atmospheric system, and their representation in models

(including processes like cloud droplet formation, ice nucleation, cloud phase transitions, secondary ice production, aerosol-cloud interactions) is one of the major challenges in climate studies (IPCC, 2013; Seinfeld et al., 2016). Compared to the liquid droplet activation process, the ice crystal formation (in mixed-phase and cirrus clouds) is affected by large uncertainties because of the poor understanding of the chemical and physical principles underlying ice nucleation and the complexity

of ice nucleation mechanisms and aerosol-ice interactions (Cantrell and Heymsfield, 2005; Gultepe and Heymsfield, 2016; Heymsfield et al., 2017; Korolev et al., 2017).

Cirrus clouds form at high altitudes and very low temperatures (below 238 K) and consist purely of ice crystals. They strongly impact the transport of water vapour entering the stratosphere (Jensen et al., 2013) and play an important role as modulator of radiation fluxes in the global radiative budget: they scatter solar radiation back into the space (albedo effect) and absorb and

re-emit longwave terrestrial radiation (greenhouse effect). Differently from other types of clouds, cirrus clouds produce a net warming at the top of the atmosphere (TOA) (e.g. Chen et al., 2000; Hong et al., 2016; Matus and L'Ecuyer, 2017). In addition, mixed-phase clouds consist of both supercooled liquid cloud droplets and ice crystals and appear at subfreezing temperatures above 238 K. Mixed-phase clouds generates a net cooling at TOA, although the estimates of their radiative effects are complicated by the coexistence of both ice and liquid cloud phases (Matus and L'Ecuyer, 2017). Due to the difference between vapour

pressure over water and over ice, ice crystals grow at the expense of water droplets (Wegener-Bergeron-Findeisen process), thus, mixed-phase clouds are thermodynamically unstable and can convert into ice-only clouds (e.g. Korolev, 2007; Korolev et al., 2017). As ice crystals can grow quickly to precipitation-sized particles, precipitation is mainly formed in mixed-phase clouds, while precipitation from cirrus clouds does not usually reach the surface (Lohmann, 2017). The mixed phase is also important for cloud electrification and intracloud lightning, which occur through the in-cloud charge separation via a transition

from supercooled raindrops to graupel over the mixed-phase temperature range (Korolev et al., 2017). The fraction of cloud ice has a profound impact on the cloud forcing in global climate models, one of the reasons why cloud radiative forcing is so diverse and uncertain (McCoy et al., 2016; Tan et al., 2016; Vergara-Temprado et al., 2018).

Ice crystal formation takes place via homogeneous and heterogeneous nucleation, depending on environmental conditions (e.g. temperature, supersaturation, vertical velocity) and aerosol populations (i.e. aerosol number concentrations and physical-

chemical characteristics) (Pruppacher and Klett, 1997; Kanji et al., 2017; Heymsfield et al., 2017; Korolev et al., 2017). Homogeneous nucleation occurs through the freezing of supercooled liquid droplets at low temperatures ($T < 238$ K) and high supersaturation over ice ($140\% - 160\%$) (Koop et al., 2000). Heterogeneous nucleation refers to the formation of ice on an aerosol surface, which reduces the energy barrier for ice nucleation and lets ice crystals form at lower supersaturations and/or at higher (subfreezing) temperatures than homogeneous nucleation. The aerosols that lead to the generation of ice crystals are

called ice nucleating particles (INPs) and are mostly insoluble, like mineral dust, soot, organics, and biological particles (Pruppacher and Klett, 1997). Heterogeneous nucleation occurs via different mechanisms called "nucleation modes" (deposition, condensation, immersion, and contact nucleation). In several modeling studies, homogeneous nucleation has been considered the dominant process for cirrus formation (e.g. Haag et al., 2003; Hendricks et al., 2011; Gettelman et al., 2012; Barahona et al., 2014) because the concentration of liquid droplets is higher than that of INPs in the upper troposphere. However, some

field measurements found a predominance of heterogeneous nucleation and lower ice crystal number concentrations (ICNCs)

than produced by homogeneous nucleation (e.g. Cziczo et al., 2013; Jensen et al., 2013). What process is dominant is still under debate, although recent studies suggested the overestimation of the vertical velocity as possible cause of the discrepancy between modeled results and observations (e.g. Barahona and Nenes, 2011; Zhou et al., 2016; Barahona et al., 2017).

Overall, two different regimes for ice crystal formation are distinguished. The *cirrus regime* at low temperatures ($T < 238$ K), where ice crystals originate via heterogeneous and homogeneous nucleation to form cirrus clouds. The *mixed-phase regime* at subfreezing temperatures between 238 K and 273 K, where ice crystals form exclusively via heterogeneous nucleation and alter the phase composition of the mixed-phase clouds. In the latter regime, besides primary nucleation, another mechanism which controls ICNCs is the secondary ice production, i.e. the production of new ice crystals via the multiplication of pre-existing ice particles without the action of INPs.

As heterogeneous nucleation takes place at lower supersaturation than homogeneous nucleation, the available water vapour and the degree of supersaturation decrease, reducing or inhibiting the formation of ice crystals from homogeneous nucleation. This competition between homogeneous and heterogeneous nucleation for water vapour drastically affects the ICNC in the cirrus regime, even at low INP concentrations (Kärcher and Lohmann, 2003; Spichtinger and Cziczo, 2010). On the other hand, both in the cirrus regime and in the mixed-phase regime, water vapour can also be reduced by depositional growth onto pre-existing ice crystals and ice crystals carried into the cloud via convective detrainment and advective transport, thus, inhibiting ice nucleation. The impact of pre-existing ice crystals (PREICE) can be especially important in cirrus clouds, when ice crystals are of small size and have low sedimentation rates at low temperatures (Barahona and Nenes, 2011), leading to optically thinner cirrus clouds characterized by fewer ice crystals with a diverse age distribution and high supersaturation levels, especially in the case of tropical upper troposphere/lowermost stratosphere (UTLS) cirrus clouds (Barahona and Nenes, 2011; Hendricks et al., 2011; Kuebbeler et al., 2014).

Cloud schemes in atmospheric and climate models have evolved from using only macrophysical properties like cloud cover to representing the microphysics explicitly, e.g. formation, evolution, and removal of cloud droplets and ice crystals (Kärcher et al., 2006; Lohmann et al., 2008; Gettelman et al., 2010; Barahona et al., 2014). Including sophisticated schemes in general circulation models (GCMs) allows for a more realistic description of the variability of cloud properties and cloud radiative effects, improving the model climate predictions (Lohmann and Feichter, 2005; Barahona et al., 2014). Recently, sophisticated parameterizations have been developed, taking into account the aerosol influence on ice formation and different modes of heterogeneous nucleation. Liu and Penner (2005) presented an ice nucleation scheme based on numerical parcel model simulations which considers the competition between homogeneous and heterogeneous nucleation following the classical nucleation theory (CNT). Kärcher et al. (2006) developed a physically based parameterization scheme of ice initiation and ice crystal initial growth in cirrus clouds, considering the PREICE effect and allowing for the competition between heterogeneous and homogeneous nucleation. Barahona and Nenes (2009) introduced an ice cloud formation parameterization, based on the analytical solution of the cloud parcel model equations, which calculates the competition for water vapour between homogeneous and heterogeneous nucleation and takes into account the variability (in size and chemical composition) of different aerosol species through a variety of INP parameterizations. Since then, these parameterizations have been included in GCMs in order to better predict cloud phase partitioning. Hendricks et al. (2011) and Kuebbeler et al. (2014) have implemented the parameterization

of Kärcher et al. (2006) into the ECHAM4 and ECHAM5-HAM models, respectively. Liu et al. (2007) and Liu et al. (2012) have implemented the parameterization of Liu and Penner (2005) into the CAM3 and CAM5 models, respectively. Also, Liu et al. (2012) and Barahona et al. (2014) have implemented the scheme of Barahona and Nenes (2009) in CAM5 and GEOS-5, respectively.

In this study the parameterization of Barahona and Nenes (2009, hereafter BN09) has been implemented into the ECHAM/MESSy Atmospheric Chemistry (EMAC) global model to improve the representation of ice nucleation. The BN09 algorithm has been modified in order to include the PREICE effect and has been used to compute the new ice crystals formed both in the cirrus regime and/or in the mixed-phase regime. Its performance has been compared with the results generated via the standard model configuration, and the model evaluation has been carried out paying particular attention to the ice-related results. The paper is

organised as follows: the description of the operational model and the BN09 scheme are in Section 2, as well as the information about the implementation work and the simulations run for this study, Section 3 describes the modeled ice-related products, Section 4 contains the evaluation of the model, and Section 5 presents our conclusions.

## 2   Model description and set-up of simulations

### 2.1   EMAC model

The EMAC model is a numerical chemistry-climate model (CCM) which describes tropospheric and middle-atmosphere processes and their interactions with ocean, land, and human influences. Such interactions are simulated via dedicated submodels in the MESSy framework (Modular Earth Submodel System, Jöckel et al., 2010), while the 5th generation European Centre Hamburg GCM (ECHAM5, Roeckner et al., 2006) is used as core of the atmospheric dynamics. For the present study we have used ECHAM5 version 5.3.02 and MESSy version 2.53.

The EMAC model has been extensively described and evaluated against in-situ observations and satellite data, e.g. aerosol optical depth, acid deposition, meteorological parameters (e.g. Pozzer et al., 2012, 2015; Karydis et al., 2016; Tsimpidi et al., 2016; Klingmüller et al., 2017). It computes gas-phase species on-line through the MECCA (Module Efficiently Calculating the Chemistry of the Atmosphere) submodel (Sander et al., 2011) and provides a comprehensive treatment of chemical processes and dynamical feedbacks through radiation (Dietmüller et al., 2016). Aerosol microphysics and gas/aerosol partitioning

are calculated by the GMXe (Global Modal-aerosol eXtension) submodel (Pringle et al., 2010), a two-moment aerosol module which predicts the number concentration and the mass mixing ratio of the aerosol modes, along with the mixing state. The aerosol size distribution is described by 7 lognormal modes (defined by total number concentration, number mean radius, and geometric standard deviation): 4 hydrophilic modes, which cover the aerosol size spectrum of nucleation, Aitken, accumulation, and coarse modes, and 3 hydrophobic modes, which have the same size range except for the nucleation mode which

is not required. The aerosol composition within each mode is uniform in size (internally mixed) but it varies among modes (externally mixed). The aging of aerosols, through coagulation or condensation of water vapour and sulfuric acid, is described by GMXe by transferring aerosols from the externally mixed to the internally mixed modes. Convective and large-scale clouds are separately treated and individually calculated by the submodels CONVECT and CLOUD, respectively. The CONVECT

submodel contains multiple convection parameterizations (Tost et al., 2006b). In this work the scheme of Tiedtke (1989) with modifications by Nordeng (1994) has been used. Convective cloud microphysics is highly simplified and neither explicit aerosol activation into liquid droplets nor aerosol effects in the ice formation processes are taken into account, i.e. convective microphysics is solely based on temperature and updraft strength. Detrainment from convection is treated by taking updraft (and downdraft) concentrations of water vapour and cloud condensate and the corresponding massflux detrainment rates into account. These are merged including turbulent detrainment (i.e. exchange of mass through the cloud edges) and organised detrainment (i.e. organized outflow at cloud top). The detrained water vapour is added to the large-scale water vapour field, while the detrained cloud condensate is directly used as a source term for cloud condensate by the large-scale cloud scheme (i.e. the CLOUD submodel), which considers the detrained condensate either liquid or ice depending on the temperature (if $T < 238$ K the phase is ice) and the updraft velocity. The number of detrained ice crystals is estimated from the ice condensate detrained from convection by assuming an only temperature dependent radius. The CLOUD submodel uses a double-moment stratiform cloud microphysics scheme for cloud droplets and ice crystals (Lohmann et al., 1999; Lohmann and Kärcher, 2002; Lohmann et al., 2007) which defines prognostic equations for specific humidity, liquid cloud mixing ratio, ice cloud mixing ratio, cloud droplet number concentration ($CDNC$), and ICNC. The advantage of using a two-moment scheme is that it allows aerosol-cloud interactions improving calculations of cloud microphysical processes and radiative transfer. In the CLOUD submodel, ice crystals form via homogeneous nucleation in the cirrus regime, via immersion and contact freezing in the mixed-phase regime (more details about ice nucleation are given in the next Subsection). Cloud droplet formation is parameterized by the "unified activation framework" (UAF) (Kumar et al., 2011; Karydis et al., 2011). It is an advanced physically based parameterization which merges two theories: $\kappa$-Köhler theory (KT) (Petters and Kreidenweis, 2007), which governs the activation of soluble aerosols, and Frenkel-Halsey-Hill adsorption activation theory (FHH-AT) (Kumar et al., 2009), which describes the droplet activation due to water adsorption onto insoluble aerosols (e.g. mineral dust). Aerosol modes that consist of only soluble material follow the KT, and the required effective hygroscopicity ($\kappa$) is calculated based on the chemical composition of the mode as described by the ISORROPIA thermodynamic equilibrium model (Fountoukis and Nenes, 2007). Aerosol modes that consist of an insoluble core with soluble coating follow the UAF scheme, which takes into account the effects of adsorption and absorption on the cloud condensation nuclei (CCN) activity of the mixed aerosol. More details about the UAF scheme and its implementation in the EMAC model can be found in Karydis et al. (2017). The diagnostic cloud cover scheme of Sundqvist et al. (1989) based on the grid mean relative humidity is used; it assumes that a grid box is partly covered by clouds when the relative humidity exceeds a threshold and is totally covered when saturation is reached. Other microphysical processes, like phase transitions, autoconversion, aggregation, accretion, evaporation of rain, melting of snow, sedimentation of cloud ice, are also taken into account by the CLOUD submodel. An evaluation of the double-moment cloud microphysics scheme used by ECHAM5 was presented in Lohmann et al. (2007, 2008) and Lohmann and Hoose (2009), applying the two-moment aerosol microphysics scheme HAM (Stier et al., 2005). Lauer et al. (2007) and Righi et al. (2013, 2015, 2016) showed an evaluation of the CLOUD submodel in conjunction with the aerosol microphysics submodel MADE (Ackermann et al., 1998), while Tost (2017) evaluated the CLOUD submodel in combination with the GMXe submodel. In Section 4 we will extend the comparison

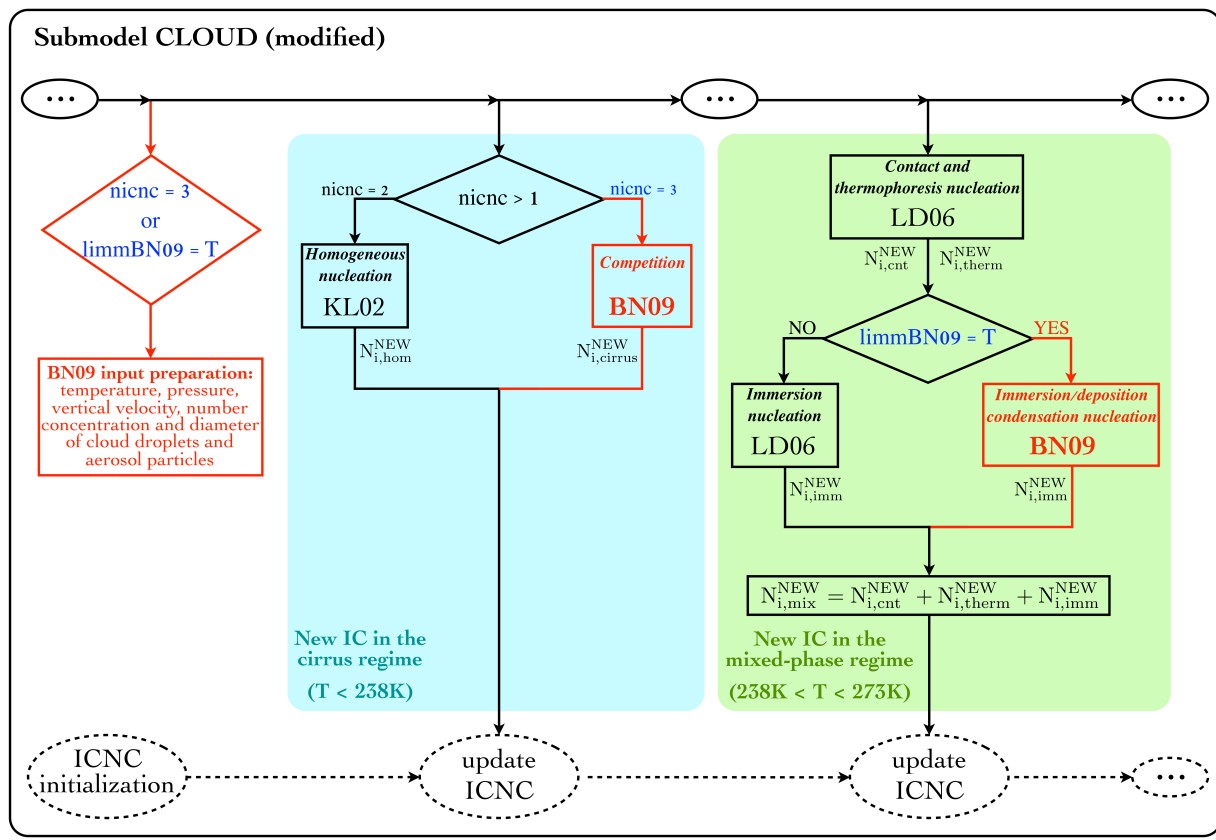

**Figure 1.** Scheme of the new ice crystal formation in the CLOUD submodel: different ice nucleation schemes can be used in the cirrus and in the mixed-phase regimes. nicnc and limm_BN09 are variables defined in the namelist-file "cloud.nml"; red parts are new; three dots indicate other processes coded in the CLOUD submodel.

with various observations. Finally, physical loss processes, like dry deposition, wet deposition, and sedimentation of aerosol, are explicitly considered by the submodels DRYDEP, SEDI, and SCAV (Kerkweg et al., 2006; Tost et al., 2006a).

## 2.2 Default ice nucleation in EMAC

The CLOUD submodel describes the evolution of the prognostic variables which undergo all cloud microphysical processes
5 (e.g. precipitation, deposition, evaporation/sublimation). As far as the formation of new ice crystals is concerned, they are computed via two independent parameterizations, as shown in black in Figure 1.

In the cirrus regime ($T \leq 238.15$ K) it is assumed that cirrus clouds form exclusively homogeneously using the parameterization of Kärcher and Lohmann (2002, hereafter referred to as KL02). Such parameterization computes the newly formed ice crystals via homogeneous nucleation ($N_{i,hom}^{NEW}$) of supercooled solution droplets and allows supersaturation with respect to ice.
10 Alternatively, it is possible to use the parameterization of Kärcher and Lohmann (2003) to simulate cirrus formation via pure

heterogeneous freezing, however, by default the model operates with KL02, under the assumption that the dominant freezing mechanism for cirrus clouds is homogeneous nucleation.

In the mixed-phase regime (238.15 K $< T \leq$ 273.15 K) heterogeneous nucleation occurs via immersion ($N_{i,imm}^{NEW}$) and contact ($N_{i,cnt}^{NEW}$) freezing as described in Lohmann and Diehl (2006, hereafter referred to as LD06). Insoluble dust can initiate contact nucleation in the presence of supercooled water droplets following the parameterization of Levkov et al. (1992). Soluble dust and black carbon can act as immersion nuclei, according to the stochastic freezing hypothesis described in Diehl and Wurzler (2004). Possibly, contact freezing via thermophoresis can be included ($N_{i,therm}^{NEW}$), but it is usually not considered (i.e. $N_{i,therm}^{NEW} = 0$) since its contribution is negligible (Lohmann and Hoose, 2009). The Wegener-Bergeron-Findeisen (WBF) process at subfreezing temperatures is parameterized, so liquid water is forced to evaporate from cloud droplets and deposit onto existing ice crystals.

In the CLOUD submodel, a single updraft velocity ($w$) is used for the whole grid cell, although $w$ can vary strongly in reality within the cell horizontal dimension (e.g. Guo et al., 2008). This is a simplification which is commonly used by GCMs, nevertheless, important progress has been recently achieved on this front to describe the subgrid-scale variability of updraft velocity using high resolution simulations (Barahona et al., 2017). In EMAC, the subgrid-scale variability of vertical velocity is introduced by a turbulent component ($w_{sub}$) which depends on the subgrid-scale turbulent kinetic energy ($TKE$) described by Brinkop and Roeckner (1995), such that $w_{sub} = 0.7\sqrt{TKE}$. The vertical velocity is given by the sum of the grid mean vertical velocity ($\overline{w}$) and the turbulent contribution: $w = \overline{w} + 0.7\sqrt{TKE}$ (Kärcher and Lohmann, 2002). Zhou et al. (2016) analysed the effect of different updraft velocity representations on ice number concentrations and showed that using $w_{sub}$ overestimates the ICNCs at temperatures below 205 K, but agrees better with the observations at higher temperatures. Other studies, e.g. Kärcher and Ström (2003) and Joos et al. (2008), showed that $w$ is in good agreement with vertical velocity observations. Given the importance of updraft velocities for ice formation (Donner et al., 2016; Sullivan et al., 2016), future studies could implement a complete probability distribution of updrafts. Finally, the influence of the pre-existing ice particles is not taken into account. The CLOUD submodel simply reduces the number of aerosol particles available for ice nucleation by the existing ice particle number in the cirrus regime.

## 2.3 Ice nucleation parameterization BN09

### 2.3.1 Scheme characteristics

The BN09 parameterization is computationally efficient and suitable for large-scale atmospheric models. It explicitly considers the competition for water vapour between homogeneous and heterogeneous nucleation in the cirrus regime, the influence of chemically-heterogeneous, polydisperse aerosols acting as INPs, and allows to use different heterogeneous nucleation parameterizations.

The BN09 algorithm can be divided in three subsequent parts. First, the limiting number of INPs ($N_{lim}$) needed to inhibit homogeneous nucleation is computed if temperatures are below 238 K. Indeed, at such low temperatures homogeneous and heterogeneous nucleation compete for water vapour decreasing ice supersaturation. When INPs exceed $N_{lim}$ and the maximum

supersaturation ($s_{max}$) is less than the threshold for homogeneous nucleation ($s_{hom}$), homogeneous nucleation is suppressed and ice crystals are formed only heterogeneously. $N_{lim}$ is determined by computing the number of INPs required to keep $s_{max}$ below $s_{hom}$.

In the second step, ice crystals nucleated heterogeneously ($N_{i,het}$) are computed via the selected INP parameterization at $s_{hom}$, then two cases can follow. If the condition $N_{i,het}(s_{hom}) \geq N_{lim}$ is satisfied, ice crystals are formed only heterogeneously at $s_{max}$ (i.e. $N_{i,het}(s_{max})$), as homogeneous nucleation is suppressed. Here, the $s_{max}$ is determined using a bisection method to balance the supersaturation within the air parcel. If $N_{i,het}(s_{hom}) < N_{lim}$, the competition between homogeneous and heterogeneous nucleation is simulated. The ice crystals nucleated homogeneously ($N_{i,hom}$) are determined via the homogeneous nucleation parameterization of Barahona and Nenes (2008, 2009) (hereafter BNhom):

$$f_c \quad = \quad f_{c,hom}\left[1 - \left(\frac{N_{i,het}(s_{hom})}{N_{lim}}\right)^{3/2}\right]^{3/2} \tag{1}$$

$$N_{i,hom} \quad = \quad N_c e^{-f_c}(1 - e^{-f_c}) \tag{2}$$

where $N_c$ is the number concentration of supercooled liquid cloud droplets and $f_c$ is the fraction of freezing soluble aerosol. The first factor of $f_c$ (i.e. $f_{c,hom}$) is defined by Barahona and Nenes (2008), while the second factor is the correction introduced by Barahona and Nenes (2009) to take into account the reduction of the probability of homogeneous nucleation due to the change in the droplet size distribution during crystal formation.

Third, the total concentration of new ice crystals formed in the cirrus regime ($N_{i,cirrus}^{NEW}$) is determined by the contribution of both heterogeneous and homogeneous nucleation. i.e. $N_{i,cirrus}^{NEW} = N_{i,het} + N_{i,hom}$ (see Figure 1). On the other hand, if the temperature is higher than $238\ K$, the algorithm uses the INP parameterization to compute $N_{i,het}(s_{max})$.

It is important to stress that the BN09 code actually includes five INP parameterizations to deal with heterogeneous nucleation (as mentioned before) and these are described by *(i)* Meyers et al. (1992), *(ii)* Phillips et al. (2007), *(iii)* Phillips et al. (2008), *(iv)* Phillips et al. (2013), and *(v)* Barahona and Nenes (2009). They are all empirically based except the latter, which is derived from the CNT. Sensitivity studies have shown that global means of ICNC vary up to a factor twenty according to the INP parameterization used (when the competition between homogeneous and heterogeneous nucleation is taken into account) and empirical based parameterizations better agree with observations, while CNT overestimates the number of ice crystals (Barahona et al., 2010; Sullivan et al., 2016). Therefore, the simulations described in Subsection 2.4 use the parameterization of Phillips et al. (2013, hereafter referred to as P13) to simulate heterogeneous nucleation, since it better agrees with observations (Sullivan et al., 2016). P13 is the improved version of Phillips et al. (2008), a comprehensive empirical formulation which takes into account the surface area contribution of different insoluble aerosols (with diameters larger than $0.1\ \mu m$) to deposition and immersion/condensation nucleation modes, besides the temperature and the supersaturation with respect to ice. The aerosol particles responsible for ice nucleation are divided in four groups: mineral dust (DU), inorganic black carbon (BC), biological aerosols (BIO), and soluble organics (OCsol). Dust and soot, the aerosol species considered in this work for

the reasons explained in Subsection 2.4, contribute to determine $N_{i,het}$ in the following way:

$$n_{INP,X} \quad = \quad \int_{log(0.1\mu m)}^{\infty} \left[ 1 - e^{-\mu_X(D_X, S_i, T)} \right] \frac{dn_X}{d\log D_X} d\log D_X \qquad X = DU, \, BC; \quad T < 273.15 \, K \tag{3}$$

$$N_{i,het}(s_{max}) \quad = \quad \sum_{X=1}^{N_X} n_{INP,X} \tag{4}$$

where $n_{INP,X}$ is the number concentration of INPs activated at a saturation ratio with respect to ice $S_i$ and temperature $T$ for the aerosol species $X$, $\mu_X$ represents the mean number of activated ice embryos per insoluble aerosol particle of species $X$ with diameter $D_X > 0.1$ μm, $n_X$ is the number concentration of aerosol particles (interstitial and INP immersed in cloud droplets) of species $X$, and $N_X$ is the number of different aerosol species. Equation (3) can be further extended for biological aerosols and soluble organics, as shown in Phillips et al. (2013), and $N_{i,het}$ denotes the new ice crystals formed via deposition and immersion/condensation nucleation modes.

Summarizing (see Figure 1), according to BN09 the new ice crystals formed in the cirrus regime are:

$$N_{i,cirrus}^{NEW} = \begin{cases} N_{i,hom} + N_{i,het}(s_{hom}) & N_{i,het}(s_{hom}) < N_{lim}, \, s_{max} = s_{hom} \\ N_{i,het}(s_{max}) & N_{i,het}(s_{hom}) \geq N_{lim}, \, s_{max} < s_{hom} \end{cases} \tag{5}$$

while in the mixed-phase regime they are computed as:

$$N_{i,imm}^{NEW} = N_{i,het}(s_{max}) \tag{6}$$

In order to account for sub-grid variabilities, the output variables of BN09 which depend on the vertical velocity ($f(w)$) are weighted over a Gaussian updraft velocity distribution by numerically calculating the integral (Morales and Nenes, 2010; Sullivan et al., 2016):

$$\overline{f(w)} = \frac{\int_0^{\infty} f(w')P(w')dw'}{\int_0^{\infty} P(w')dw'} \tag{7}$$

where $P(w')$ is the Gaussian probability density function of sub-grid vertical velocities ($w'$) with mean $0.1 \, \text{cm s}^{-1}$ and standard deviation equal to $w_{sub}$.

## 2.3.2 Implementation

The BN09 parameterization has been added in the MESSy framework in order to compute the newly formed ice crystals in the cirrus regime and/or in the mixed-phase regime. The input variables of BN09 are: temperature ($T$, $[\text{K}]$); pressure ($P$, $[\text{Pa}]$); width of the vertical velocity distribution ($w_{sub}$, $[\text{m s}^{-1}]$) with upper limit $3 \, \text{m s}^{-1}$ and lower limit $0.01 \, \text{m s}^{-1}$; number concentration of activated cloud droplets ($N_c$, $[\text{m}^{-3}]$), dry diameter of Aitken soluble mode ($D_c$, $[\text{m}]$, see Appendix A) and standard deviation of Aitken soluble mode ($\sigma_c$); number concentrations ($N_X$, $[\text{m}^{-3}]$), geometric mean dry diameters ($D_M$, $[\text{m}]$), and lognormal standard deviations ($\sigma_M$) of interstitial aerosol of species $X$ (which can be DU, BC, OCsol, BIO,

depending on the choice of the INP parameterization). Given the internally mixed representation of aerosols in EMAC, the diameters $D_M$ are not distinguished among aerosol species but only among the modes (Aitken ($K$), accumulation ($A$), coarse ($C$), i.e. $M = K, A, C$) which the species belong to. Similarly, the standard deviations $\sigma_M$ are constant depending only on the mode (in EMAC $\sigma_K = \sigma_A = 1.59$ and $\sigma_C = 2.0$).

A schematic overview of how BN09 has been implemented in EMAC through the CLOUD submodel is shown in Figure 1. Moreover, the PREICE effect has been included in the BN09 code. This effect is parameterized by reducing the vertical velocity for ice nucleation ($w_{sub}$) by a factor depending on the pre-existing ice crystal number concentration and size, limiting the expansion cooling. Such "corrected" vertical velocity ($w_{sub,pre}$) has been computed as defined in equation (24) by Barahona et al. (2014). Further information about the implementation is given in Appendix A.

## 2.4    Setup of simulations

In this study EMAC simulations have been carried out with T42L31ECMWF resolution, which corresponds to a spherical truncation of T42 (i.e. quadratic Gaussian grid of approx. $2.8° \times 2.8°$ in latitude and longitude) and 31 vertical hybrid pressure levels up to $10$ hPa (approx. $25$ km) at the lower stratosphere. All simulations have been run for 6 years (1 year as spin-up time plus 5 years for the analysis) using emissions starting from the year 2000 (GFEDv3.1 from van der Werf et al. 2010 for

biomass burning and CMIP5-RCP4.5 from Clarke et al. 2007 for anthropogenic emissions). As in Pozzer et al. (2012), dust is offline prescribed using monthly emission files based on the AEROCOM data set (Dentener et al., 2006). Also volcanic and secondary organic aerosol emissions are based on AEROCOM, while GFEDv3.1 and CMIP5-RCP4.5 have been used to simulate emissions of black carbon and organic aerosols, respectively. Finally, aerosol climatologies have been used for the interactions with radiation (Tanre et al., 1984) and heterogeneous chemistry (Aquila et al., 2011). Prescribed climatologies

of sea surface temperatures (SST) and sea-ice concentrations (SIC) from AMIP (30 years: 1980-2009) have been used as boundary conditions. Daily means have been saved as output, and monthly means have been used for the analysis in Sections 3 and 4.1.

Table 1 lists all simulations of this study and summarises their main characteristics. The default experiment (DEF or KL+LD) is performed with the standard configuration of the EMAC model, i.e. using the parameterization of Kärcher and Lohmann

(2002) for cirrus clouds and the parameterization of Lohmann and Diehl (2006) for immersion nucleation in the mixed-phase regime. The UAF scheme is used as cloud droplet formation parameterization, like in Karydis et al. (2017). In order to investigate the model performace using the BN09 scheme, we carried out three other experiments where BN09 operates in the two cloud regimes in different combinations: BN09 computing the new ice crystals in the cirrus regime (BN+LD), in the mixed-phase regime (KL+BN), and in both regimes (BN+BN).

In all experiments, contact nucleation is computed according to LD06, while thermophoresis contact nucleation is not considered since its contribution is negligible (as remarked in Subsection 2.2). The P13 parameterization is used to simulate deposition and immersion/condensation nucleation whenever BN09 is called (for the reasons explained in Subsection 2.3). Since LD06 takes into account only dust and soot for immersion nucleation, we set the same aerosol species as contributions for P13 and turned off the biological and organic contributions.

| Experiment name | Ice nucleation scheme | |
| --- | --- | --- |
| | *Cirrus regime* | *Mixed-phase regime* |
| KL+LD or DEF | KL02, pure homogeneous nucleation | LD06, immersion nucleation |
| BN+LD | BN09, competition and PREICE | |
| KL+BN | KL02, pure homogeneous nucleation | BN09, immersion/condensation |
| BN+BN | BN09, competition and PREICE | and deposition nucleation via P13 |

**Table 1.** All experiments carried out in this study.

## 3 Model results

BN09 improves the ice nucleation representation in EMAC by taking into account processes (e.g. water vapour competition, influence of polydisperse aerosols, PREICE effect) which were previously neglected by KL02 and LD06. In this Section we investigate the changes and the effects obtained by using BN09 in the different regimes.

### 3.1 Annual zonal means

The annual zonal means of ICNC and ice water content ($IWC$) are shown as a function of latitude and altitude in Figure 2, where the isolines at 273 K and 238 K indicate the approximate bounds of cirrus and mixed-phase regimes. Despite the different ice nucleation parameterizations, ICNCs show similar qualitative patterns in all simulations, indicating the important role of atmospheric dynamics. Their numbers decrease towards lower altitudes (Figure 2*a*) because the ice nucleation rate reduces with increasing temperature, while they are much higher over the mid-latitudes in the Northern Hemisphere (NH) because of larger INP concentrations and the influence of big mountain chains, e.g. Rocky Mountains and the Himalayas. Looking at the relative changes, we note that ICNCs computed with BN09 in the cirrus regime are much lower than the default ICNCs in the upper troposphere and at high latitudes in the Southern Hemisphere (SH), where they are lower by up to $80\%$ (Figure 2*b*). The absolute changes of ICNC annual zonal means computed as a function of latitude and temperature (Figure S1 in the supplement file) show that ICNCs in BN+KL are lower than the default case by $300 \, \mathrm{L^{-1}}$ at temperatures below 220 K. As ice crystals are formed almost exclusively via homogeneous nucleation here (not shown) and BNhom and KL02 produce the same order of magnitude of ICNCs (Barahona and Nenes, 2008), the negative bias is likely due to the PREICE effect predicted by BN09. Indeed, it has been demonstrated that homogeneous nucleation dominates in the upper troposphere in the tropics and in the SH (Haag et al., 2003; Liu et al., 2012; Barahona et al., 2017), while heterogeneous nucleation is important in the NH (Li et al., 2012; Kuebbeler et al., 2014; Storelvmo and Herger, 2014; Shi et al., 2015; Gasparini and Lohmann, 2016; Barahona et al., 2017) where cirrus clouds are formed from a combination of homogeneous and heterogeneous processes. Interestingly, ICNCs at lower altitudes are also influenced by the ice nucleation parameterization used in the cirrus regime. In fact, there is an increase of ICNCs in the mixed-phase regime probably due to a faster sedimentation of the larger ice crystals produced by BN09 in cirrus clouds, especially in the NH where there are larger sources of efficient ice-nucleating mineral dust. Overall, as remarked later in Subsection 4.1, the total ICNC in BN+LD globally decreases. The changes produced by applying

BN09 in the mixed-phase regime (Figure 2$c$) result from the different heterogeneous ice nucleation parameterizations used to simulate immersion nucleation, P13 vs. LD06. The changes are especially evident in the NH (more than 40%), where mineral dust is more abundant than in the SH. As P13 produces fewer new ice crystals than LD06 (not shown), the positive biases in the mixed-phase regime are possibly due to influences from the cirrus regime (e.g. ice crystal sedimentation) and convective detrainment. Overall, the ICNC deviations in the mixed-phase regime obtained using the two different parameterizations are smaller (mostly within $\pm 20\%$) than in the cirrus regime. This is also evident from Figure S1 in the supplement file (last row), where the absolute changes are, in average, between 200 and $-200$ L$^{-1}$ when BN09 is used in the cirrus regime and between 50 and $-50$ L$^{-1}$ when comparing KL+BN with KL+LD. Possibly, the rate associated to heterogeneous nucleation in the mixed-phase regime is masked by other processes, like sedimentation and aggregation, which also contribute to ICNC in this regime. Finally, the simulation using BN09 in both regimes combines the effects described so far (Figure2$d$). Since cirrus clouds do not occur throughout the whole year, we present in the supplement file (Figure S2) the ICNC seasonal means for summer (June-July-August, JJA) and winter (December-January-February, DJF). The seasonal analysis helps to understand why there is cirrus occurrence at temperatures higher than 238 K, showing that the ICNC growth in the mixed-phase region predicted by BN+LD is actually very small, as expected given that the ice scheme used in the mixed-phase regime is the same as the default simulation.

The IWC pattern (Figure 2$e$) qualitatively follows the ICNC distribution. It is quite symmetrical between the two hemispheres except at high latitudes in the NH, where IWC is slightly higher because of the higher values of ICNC. Particularly, IWC exhibits three local maxima: two over the mid-latitudes in both hemispheres and one in the tropics, associated to storm tracks and deep convections, respectively (Li et al., 2012), in agreement with satellite observations, e.g. Waliser et al. (2009), Li et al. (2012). The relative changes in Figure 2$f$ show a pattern very similar to Figure 2$b$, therefore, IWC decreases where ICNC reduces (and vice versa) when BN09 is used in the cirrus regime. On the other hand, IWC in KL+BN slightly reduces (up to 20%) in the mixed-phase regime in areas where ICNC increases, especially in the NH at high latitudes (Figure 2$g$). This could be due to the the different sizes of ice crystals, however, the areas with significance are rather small. Finally, BN+BN in Figure 2$h$ simulates an overall reduction of IWC except in the three areas with higher values of IWC described in Figure 2$e$.

## 3.2 Global distributions

Figure 3 shows the global distributions of ICNC annual means at two different altitudes: 200 hPa (where temperatures vary between 200 K and 220 K) to represent the cirrus regime and 600 hPa (where temperatures are approximately between 240 K and 260 K) to represent the mixed-phase regime. ICNCs in the cirrus regime (Figure 3$a$) show areas with high values over land and in correspondence with mountainous regions, e.g. the Rocky Mountains, Andes, and Tibetan Plateau with ICNCs $> 500$ L$^{-1}$. Such pattern is strongly related to the turbulent contribution of the vertical velocity $w_{sub}$ and in agreement with Gryspeerdt et al. (2017), who detected in these areas mostly orographic cirrus clouds. Figure 3$a$ also shows higher ICNCs around the edge of the Antarctic ice sheet and over those regions which experience a strong convective activity, i.e. the Inter Tropical Convergence Zone (ITCZ) and the Tropical Warm Pool (TWP), as observed in Sourdeval et al. (2018). The annual global mean of ICNC at 200 hPa is about 200 L$^{-1}$ ($\sim 390$ L$^{-1}$ over land and $\sim 124$ L$^{-1}$ over ocean). The relative changes

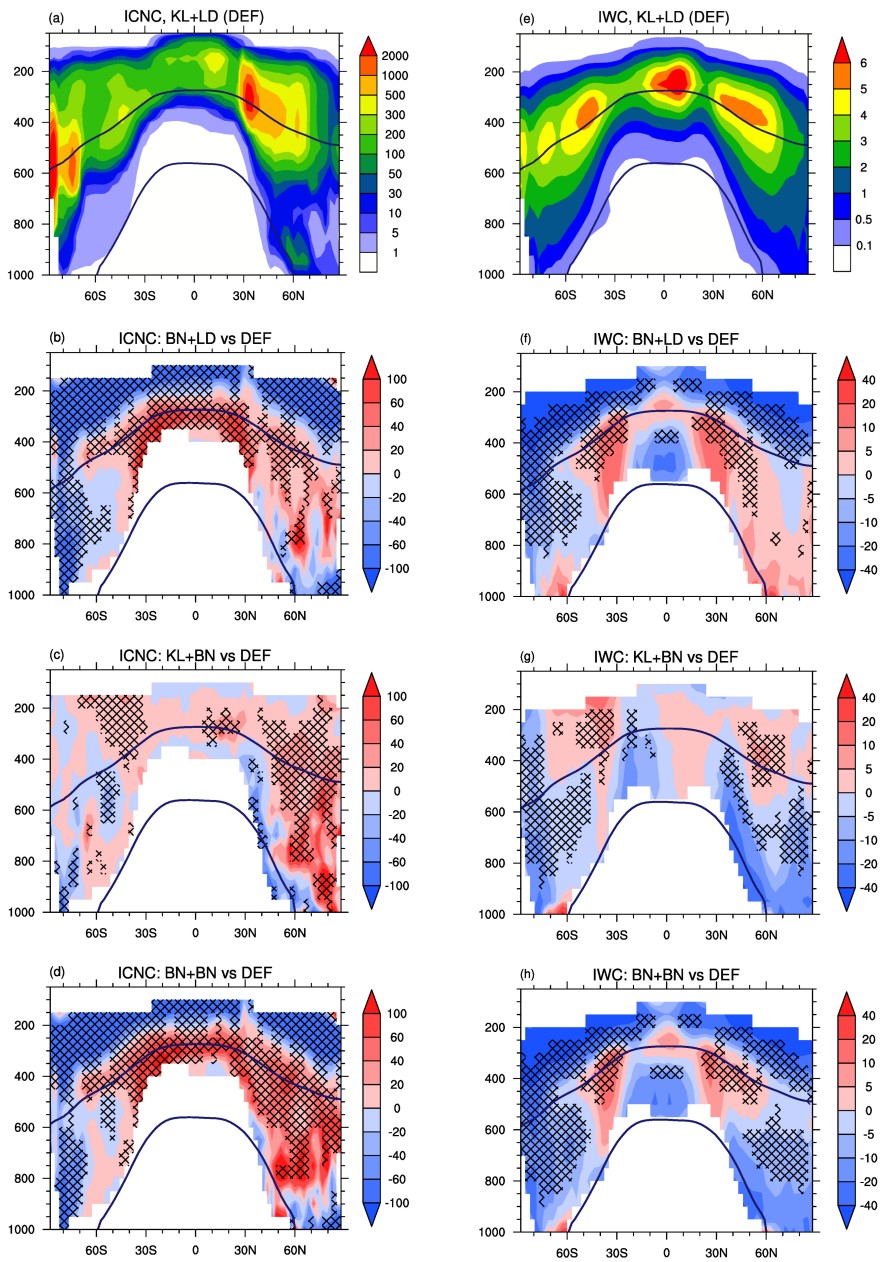

**Figure 2.** Annual zonal means of (grid-averaged) ice crystal number concentration ($ICNC$, $[\mathrm{L}^{-1}]$) and non-precipitable ice water content ($IWC$, $[\mathrm{mg\,kg}^{-1}]$) for the default simulation KL+LD and the relative percentage changes of BN+LD, KL+BN, and BN+BN with respect to it (i.e. $(experiment - DEF)/|DEF| \cdot 100$), computed where $ICNC^{DEF} \geq 1\,\mathrm{L}^{-1}$ and $IWC^{DEF} \geq 0.1\,\mathrm{mg\,kg}^{-1}$. The isotherms at 273 K and 238 K are annual means, the crossed pattern indicates areas with a significance level of 95%.

clearly show that BN09 used in the cirrus regime (Figure 3*b, d*) reduces ICNC (up to 60%) worldwide with respect to the default experiment, and the ICNC annual global mean drops to 137 L$^{-1}$ (i.e. more than 30%). As the ice crystals are mainly of homogeneous origin at this altitude, such a reduction is probably due to the PREICE effect. However, there are positive biases along the ITCZ and over the TWP area. As the concentrations of new ice crystals produced by BN09 are not particularly

remarkable in these regions (not shown), convective detrainment is likely to play a role. Indeed, there is a certain response of the convective activity to the choice of the ice nucleation scheme used in the cirrus regime. On the contrary, KL+BN is characterised by a general increase of ICNC (Figure 3*c*). However, most of the areas with strong positive changes (larger than 60%) correspond to regions characterized by low ICNC ($< 30$ L$^{-1}$), thus, the global annual mean increases just up to 218 L$^{-1}$ (i.e. +9%). At 600 hPa, ICNCs increase towards high latitudes, in particular over Greenland (up to 2000 L$^{-1}$) and Antarctica

(mostly $> 2000$ L$^{-1}$) (Figure 3*e*). It must be said that, due to the very low temperatures in the the latter region, even at 600 hPa the conditions are typical of the cirrus regime, and the high ICNCs can be related to the high values of both $w_{sub}$ and ice supersaturation. Gryspeerdt et al. (2017) found that cirrus clouds over Antarctica have primarily synoptic origin. However, differently from Figure 3*e*, observations do not present such a high peak of ICNC over Antarctica (Gryspeerdt et al., 2018; Sourdeval et al., 2018). The annual global mean is about 53 L$^{-1}$, which means about one quarter with respect to the ICNC

global mean at 200 hPa. Figure 3*f* confirms what already noticed in Figure 2*b*, that is the ice nucleation scheme used in the cirrus regime affects the ICNC at the mixed-phase regime altitudes predicting higher ICNCs especially in the NH. However, the largest differences occur in areas where ICNCs are very low and slightly affect the absolute ICNC values. As a result, the annual global mean actually decreases to 47 L$^{-1}$ because of the negative contribution in the SH. Figure 3*g* also shows strong positive biases, but ICNCs do not change globally (52 L$^{-1}$). Thus, we can reiterate that the ICNC in the mixed-phase regime is

less sensitive to the ice nucleation scheme changes than the ICNC in the cirrus regime. Vertically integrated ice crystal number concentrations ($ICNC_{burden}$, Figure S3 in the supplement file) clearly show that concentrations are higher over continents ($\sim 48 \cdot 10^8$ m$^{-2}$), where vertical updrafts are stronger and aerosol concentrations more abundant, than over oceans ($\sim 11 \cdot 10^8$ m$^{-2}$).

IWC at 200 hPa and 600 hPa (Figure 4) presents a pattern qualitatively similar to the ICNCs at the corresponding heights.

Nevertheless, two interesting features appear. First, the high IWC values ($> 10$ mg kg$^{-1}$) over the TWP at 200 hPa, where ICNCs are not particularly high. This is probably caused by the larger radius of ice crystals simulated in this area. Second, IWC at 600 hPa is rather low over Antarctica (likely because of the low water vapour concentration), which is instead one of the regions with the highest ICNC. The relative changes of IWC with respect to the default simulation (Figure S4 in the supplement file) approximately follow the changes obtained for ICNC, i.e. IWC reduces where ICNC decreases and vice versa.

## 4   Model comparisons and observations

### 4.1   Annual global means

Table 2 shows an overview of the global annual means of cloud microphysical variables and radiative fluxes computed for different observations and for all experiments, and the percentage changes with respect to the default simulation. The global

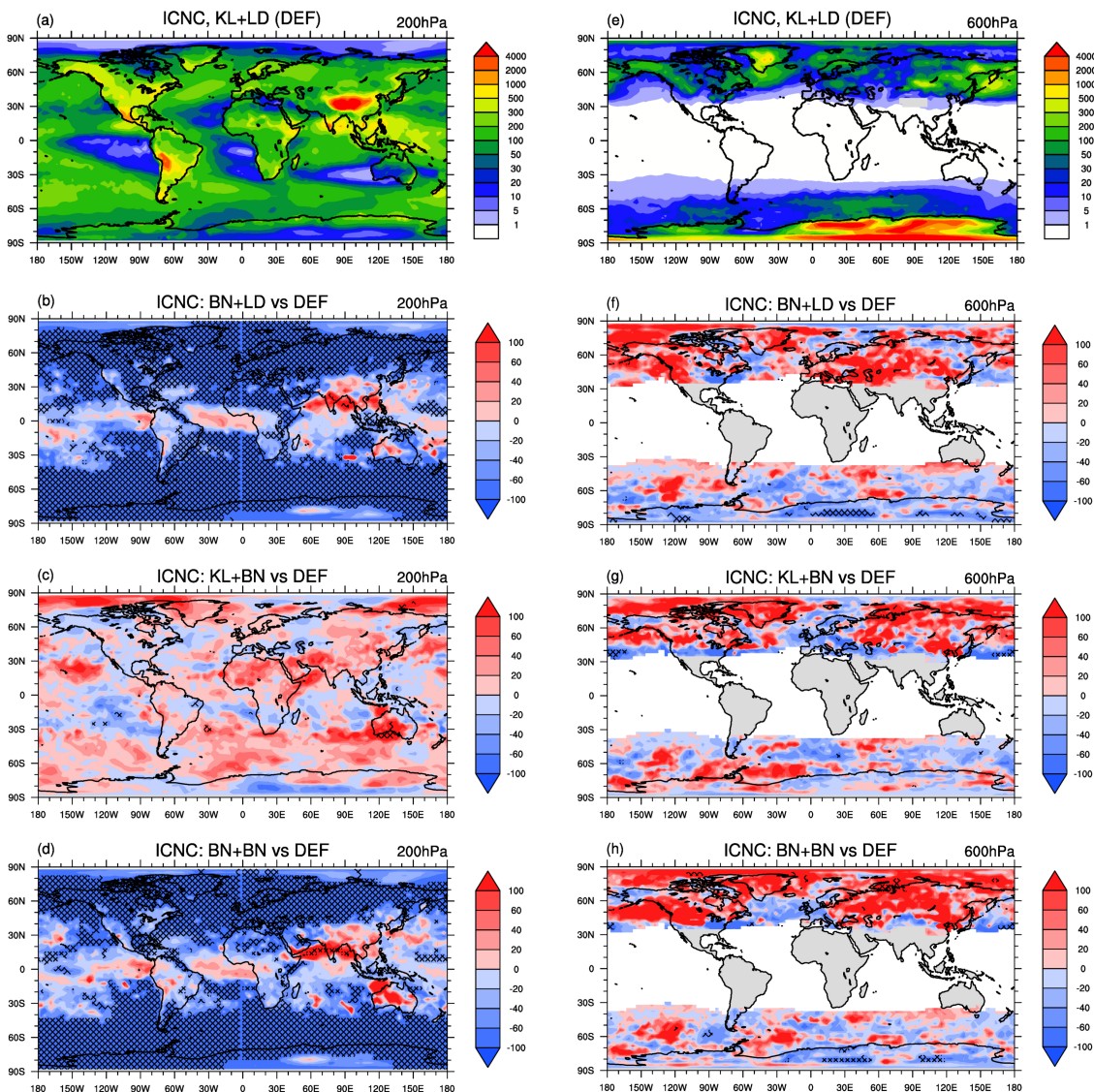

**Figure 3.** Annual means of (grid-averaged) ice crystal number concentration ($ICNC$, $[\mathrm{L}^{-1}]$) at 200 hPa (cirrus regime) and 600 hPa (mixed-phase regime) for the default simulation KL+LD and the relative percentage changes of BN+LD, KL+BN, and BN+BN with respect to it (i.e. $(experiment - DEF)/|DEF| \cdot 100$). The crossed pattern indicates areas with a significance level of 95%.

vertically integrated ice crystal number concentration changes considerably depending on the ice scheme used in the cirrus regime and in mixed-phase regime. When BN09 operates in the cirrus regime, ICNC_burden decreases by 10% due to the competition between homogeneous and heterogeneous nucleation and the PREICE effect (a similar result has been found also by Liu et al. 2012, Kuebbeler et al. 2014, and Shi et al. 2015). On the other hand, ICNC_burden increases by almost 7% when 5 BN09 is used in the mixed-phase regime, i.e. when P13 simulates heterogeneous nucleation. On a large scale, these effects

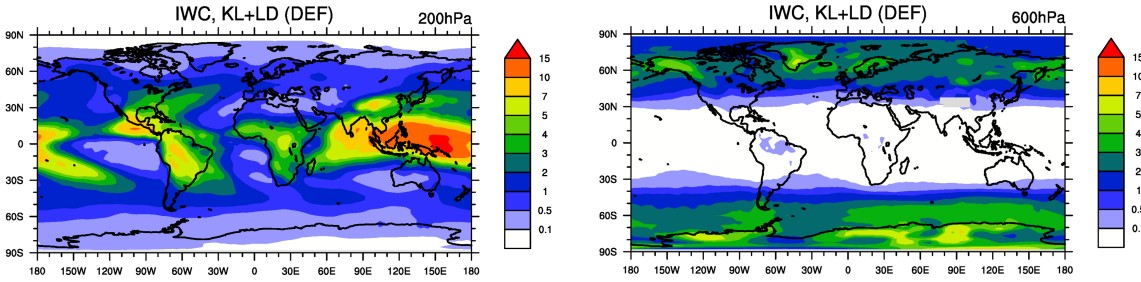

**Figure 4.** Annual means of (grid-averaged) ice water content ($IWC$, $[\mathrm{mg\,kg^{-1}}]$) at 200 hPa (cirrus regime) and 600 hPa (mixed-phase regime) for the default simulation KL+LD.

offset each other in BN+BN, where the global annual mean is basically unchanged with respect to the default simulation. Overall, the ICNC$_{\mathrm{burden}}$ values are very close to the global annual means found by Lohmann et al. (2008) and Kuebbeler et al. (2014), while they are an order of magnitude higher compared to the results of Wang and Penner (2010) and Shi et al. (2015). $ICNC_{burden,cirri}$ and $ICNC_{burden,mixed}$ are vertically integrated ICNCs in the cirrus regime and in the mixed-phase regime,

respectively. It is interesting to see quantitatively the different contributions to the total ICNC$_{\mathrm{burden}}$: ICNCs in the cirrus regime are about 6 times larger than the ICNCs in the mixed-phase regime when KL02 is used and about 5 times when BN09 is applied in the cirrus regime. In general, we observe that the variability of ICNC increases when BN09 is used. Vertically integrated cloud droplet number concentration ($CDNC_{burden}$) is basically not influenced by the choice of the ice nucleation scheme. Its values are comparable with previous modeling studies (e.g. Lohmann et al., 2007; Hoose et al., 2008; Salzmann et al., 2010;

Wang and Penner, 2010; Kuebbeler et al., 2014; Shi et al., 2015) and observations, although satellite observations are still affected by strong uncertainties (Bennartz and Rausch, 2017).

The ice water path ($IWP$) decreases by almost 7% when BN09 is used in the cirrus regime, similarly to what has been found in Kuebbeler et al. (2014), who compared simulations assuming pure homogeneous nucleation against simulations including water vapour competition. Overall, the model underestimates the IWP, also found in other studies that applied ECHAM-HAM

(e.g. Lohmann et al., 2008; Lohmann and Hoose, 2009; Kuebbeler et al., 2014; Gasparini et al., 2018), however, there are still large discrepancies among observational datasets which make problematic the validation of the models (Duncan and Eriksson, 2018). The liquid water path ($LWP$) estimates derived from satellite observations vary substantially, between 23 and 87 $\mathrm{g/m^2}$ (Li et al., 2012; Han et al., 1994). The modeled results fall within this range and the one indicated as acceptable by Mauritsen et al. (2012), which is $50 - 80\ \mathrm{g/m^2}$. The LWP variations among the experiments are much smaller than the IWP variations.

The absolute values of the shortwave cloud radiative effect ($SCRE$) and longwave cloud radiative effect ($LCRE$) are higher than those derived from satellite data, especially when KL02 is employed in the cirrus regime. However, when the net cloud radiative effect ($NCRE$) is computed, the same simulations with KL02 in the cirrus regime are closer to the observations. Looking at the absolute changes and the global distributions in the supplement file (Figure S5) it is evident that the cloud radiative effects are sensitive to the ice nucleation scheme used for cirrus clouds. Indeed, SCRE with BN09 becomes weaker

(more than 5%) because of the less efficient scattering of shortwave radiation by fewer and larger crystals. More importantly,

LCRE decreases up to $15\%$ in BN+LD because cirrus clouds, at the same, can trap less longwave radiation in the Earth-atmosphere system. As a result, NCRE becomes more negative, with statistically significance over some areas in the tropics and high latitudes, and the cooling effect is enhanced.

The total cloud cover ($TCC$) is slightly overestimated by the model (likely explaining why the cloud radiative forcing is high despite IWP being half of the observed values). However, Mauritsen et al. (2012) assert that a global model is acceptable if TCC is higher than $60\%$. The changes with respect to the default simulation are very low (below $2\%$), and the biggest change is in BN+LD where TCC reduces by $1.39\%$, since lower ICNCs leads to higher sedimentation rates. Finally, the model tends to overestimate the total precipitation ($P_{tot}$), i.e. the sum of large scale and convective precipitations, but this has also been found with other global models (e.g. Barahona et al. 2014 with GEOS-5, Shi et al. 2015 with CAM5, and Lohmann et al. 2008 and Kuebbeler et al. 2014 with ECHAM-HAM as well). When BN09 is used in the cirrus regime, $P_{tot}$ grows by $4\%$ especially because of the increase of the convective precipitation contribution, due to some feedbacks on the convective activity generated by the different ice nucleation schemes used, as mentioned in Subsection 3.2.

The annual zonal means of vertically integrated number concentration of ice crystals and cloud droplets, ice water path, liquid water path, shortwave and longwave cloud radiative effects, and total cloud cover are shown in Figure S6 (in the supplement file) and are comparable with the literature cited before. The annual zonal mean profiles show clearly that the simulations using the same ice nucleation scheme in the cirrus regime are very close to each other, i.e. KL+LD and KL+BN, and BN+LD and BN+BN (as already visible in Table 2).

Overall, the model performs well with respect to observations and the literature. Mostly, the experiments do not yield evident differences among each other at the global scale, as regional variations may cancel out, however, there are clear effects on SCRE and LCRE from changing the cirrus ice nucleation scheme. As there is not a clear indication which simulation performs better, in the next Subsection we extend our analysis including a statistical comparison with aircraft measurements.

## 4.2 Comparison with aircraft measurements

The validation of climate models with measurements from field experiments or aircraft campaigns is always limited by the fact that the models have difficulties to capture individual meteorological events. Nevertheless, here we consider a big collection of aircraft measurements recorded in 15 years, between 1999 and 2014 (*Krämer: personal communication, not yet published*). 18 field campaigns (in total, 113 flights with about 127 hours in cirrus clouds) covered Europe, Australia, Africa, Seychelles, Brazil, USA, Costa Rica, and tropical Pacific (i.e. between 25°S and 75°N) in the temperature range of $185 - 243$ K. This extensive observational data set is compared to the modeled in-cloud ICNCs in Figure 5 (*left*). The observed ICNC varies between 8 and 80 L$^{-1}$ over the entire temperature range, and the lower and upper quartiles vary between 0.6 and 300 L$^{-1}$.

Again, the simulations can be grouped in two sets according to the ice nucleation scheme used in the cirrus regime, i.e. KL+LD/KL+BN and BN+LD/BN+BN, because of their similarities. For most of the temperature range, the simulations which use KL02 in the cirrus regime overestimate the observed ICNCs (although they mostly remain below the $75^{th}$ percentile). The overestimation of ICNCs is common to other modeling studies (e.g. Wang and Penner, 2010; Liu et al., 2012; Shi et al., 2015) and especially in cold cirrus clouds (for $T < 205$ K). On the other hand, the simulations which use BN09 in the cirrus regime

| | KL+LD (DEF) | | BN+LD (2) | | KL+BN (3) | | BN+BN (4) | | OBSERVATIONS | 2 vs DEF | 3 vs DEF | 4 vs DEF |
|---|---|---|---|---|---|---|---|---|---|---|---|---|
| $CDNC_{burden}$ | 4.15 | (0.04) | 4.21 | (0.05) | 4.12 | (0.03) | 4.18 | (0.06) | $4.01^{(A)}$ | 1,32 | -0,72 | 0,72 |
| $ICNC_{burden}$ | 21.86 | (0.27) | 19.61 | (0.32) | 23.33 | (0.24) | 21.75 | (0.50) | | -10,29 | 6,72 | -0,50 |
| $ICNC_{burden,cirri}$ | 18.95 | (0.24) | 16.47 | (0.31) | 20.26 | (0.18) | 18.40 | (0.41) | | -13,09 | 6,91 | -2,90 |
| $ICNC_{burden,mixed}$ | 3.06 | (0.10) | 3.29 | (0.13) | 3.23 | (0.12) | 3.52 | (0.16) | | 7,44 | 5,52 | 14,82 |
| $LWP$ | 75.38 | (0.20) | 72.73 | (0.24) | 76.59 | (0.36) | 74.62 | (0.63) | $87.1^{(B)}$, $23.0^{(C)}$ | -3,52 | 1,61 | -1,01 |
| $IWP$ | 12.79 | (0.04) | 11.95 | (0.06) | 12.70 | (0.02) | 11.85 | (0.03) | $25.8^{(C)}$, $29.0^{(D)}$ | -6,57 | -0,70 | -7,35 |
| $SW_{NET,TOA}$ | 229.30 | (0.11) | 232.20 | (0.06) | 229.10 | (0.06) | 231.70 | (0.26) | $241.70^{(1)}$, $240.50^{(3)}$ | 1,26 | -0,09 | 1,05 |
| $LW_{TOA}$ | -224.80 | (0.20) | -230.70 | (0.16) | -224.40 | (0.10) | -230.10 | (0.12) | $-235.40^{(1)}$, $-239.80^{(3)}$ | -2,62 | 0,18 | -2,36 |
| $Imbalance_{TOA}$ | 4.52 | (0.22) | 1.53 | (0.14) | 4.65 | (0.14) | 1.58 | (0.26) | $5.87^{(1)}$, $0.71^{(3)}$ | -66,15 | 2,88 | -65,07 |
| $SCRE$ | -57.82 | (0.12) | -54.83 | (0.08) | -58.07 | (0.09) | -55.38 | (0.25) | $-48.50^{(1)}$, $-47.14^{(2)}$, $-47.04^{(3)}$ | 5,17 | -0,43 | 4,22 |
| $LCRE$ | 33.95 | (0.11) | 28.90 | (0.10) | 34.40 | (0.06) | 29.53 | (0.09) | $29.42^{(1)}$, $26.87^{(2)}$, $26.00^{(3)}$ | -14,87 | 1,33 | -13,02 |
| $NCRE$ | -23.87 | (0.18) | -25.93 | (0.10) | -23.68 | (0.14) | -25.86 | (0.27) | $-19.07^{(1)}$, $-19.7^{(2)}$, $-21.04^{(3)}$ | -8,63 | 0,80 | -8,34 |
| $TCC$ | 70.01 | (0.13) | 69.04 | (0.11) | 70.04 | (0.14) | 69.23 | (0.16) | $66.83^{(4)}$, $66.7^{(5)}$ | -1,39 | 0,04 | -1,11 |
| $P_{tot}$ | 2.902 | (1.9E-05) | 3.032 | (1.4E-05) | 2.892 | (2.6E-05) | 3.024 | (3.1E-05) | $2.624^{(6)}$, $2.669^{(7)}$ | 4,48 | -0,34 | 4,20 |

**Table 2.** Global annual means for simulations and observations. Shown are grid-averaged vertically integrated cloud droplet number concentration ($CDNC_{burden}$, $[\,10^{10}\ \mathrm{m}^{-2}]$), vertically integrated ice crystal number concentration ($ICNC_{burden}$, $[\,10^8\ \mathrm{m}^{-2}]$), vertically integrated ice crystal number concentration in the cirrus regime ($ICNC_{burden,cirri}$, $[\,10^8\ \mathrm{m}^{-2}]$), vertically integrated ice crystal number concentration in the mixed-phase regime ($ICNC_{burden,mixed}$, $[\,10^8\ \mathrm{m}^{-2}]$), grid-averaged liquid water path ($LWP$, $[\,\mathrm{g\,m}^{-2}]$) and ice water path ($IWP$, $[\,\mathrm{g\,m}^{-2}]$), net shortwave radiative flux ($SW_{NET,TOA}$, $[\,\mathrm{W\,m}^{-2}]$), longwave radiative flux ($LW_{TOA}$, $[\,\mathrm{W\,m}^{-2}]$), and radiative imbalance ($Imbalance_{TOA}$, $[\,\mathrm{W\,m}^{-2}]$) at TOA, shortwave cloud radiative effect ($SCRE$, $[\,\mathrm{W\,m}^{-2}]$), longwave cloud radiative effect ($LCRE$, $[\,\mathrm{W\,m}^{-2}]$), net cloud radiative effect ($NCRE$, $[\,\mathrm{W\,m}^{-2}]$), total cloud cover ($TCC$, $[\,\%\,]$), total precipitation ($P_{tot}$, $[\,\mathrm{mm\,day}^{-1}]$). The values in brackets are (temporal) standard deviations. The sixth column contains the annual global means computed using the satellite data from ERBE 1985-1990[1] and 2000-2006[4], CERES-SYN1deg 2004-2010[2], CERES-EBAF 2000-2016[3], MODIS-TERRA 2004-2008[5], CMAP 1970-2016[6], GPCP 19790-2009[7], and global means taken from the literature: [A] is derived from AVHRR data (Gettelman et al., 2010), [B] from NOAA-9 and NOAA-10 data (Han et al., 1994), [C] from CloudSat data (Li et al., 2012), and [D] from ISCCP data (Storelvmo et al., 2008). Last three columns show the percentage changes $[\,\%\,]$ of the experiments 2, 3, 4 with respect to the default simulation, i.e. $(experiment - DEF)/|DEF| \cdot 100$.

are very close to the observations at temperatures below 200 K and between 220 K and 230 K, while they underestimate ICNCs between 200 K and 220 K. In this temperature range the simulations can exceed the observed $25^{th}$ percentile (although remaining within the $5^{th}$ percentile). In comparison with the other two simulations, BN+LD and BN+BN always predict lower ICNCs at temperatures below 230 K, as expected because of the competition and PREICE effects. Finally, all four simulations overestimates ICNCs by one order of magnitude in the temperature range $230 - 240$ K.

Overall, the simulations BN+LD and BN+BN agree particularly well with the measurements at temperatures lower than 200 K but underestimate the ICNCs within the interval $200 - 220$ K, due to an overestimation of the competitive nucleation and PREICE effects. Barahona et al. (2010) showed that the competitive nucleation effect is small using P13. Also, Liu et al. (2012) found that BN09 (using the parameterization of Phillips et al. (2008) for heterogeneous nucleation) and BNhom produced very similar results in the cirrus regime, suggesting that the competive nucleation effect was small because of the low ICNCs formed heterogeneously. Thus, we can deduce that the PREICE effect is the one which is likely overestimated in our simulations.

Interestingly, modeled ICNCs do not show any particular trend, like also Kuebbeler et al. (2014) who used ECHAM-HAM. Differently, other studies found that ICNCs are inversely proportional with temperature, e.g. Liu et al. (2012) and Shi et al. (2015) with CAM5, indifferently if they used the ice nucleation scheme of Liu and Penner (2005) or BN09, and Barahona et al. (2010) with GEOS-5 and BN09. Such distinct behaviours are likely derived from the wide model variabilities in reproducing

subgrid-scale processes, like vertical velocity, which play a role in ice nucleation. We reiterate that ICNC is highly dependent on the vertical velocity which is usually poorly represented in terms of spatial and temporal variability (Barahona et al., 2017).

For further information, in Figure 5 (*right*) we also show the modeled in-cloud ICNCs in the mixed-phase regime, considering the same latitudes as the case before (25°S - 75°N). The simulations do not show significant differences among each other. The distinctive features are the ICNC decrease with increasing temperatures and a positive "bulge" between 265 K and

270 K caused by secondary ice production (rime splintering). The modeled ICNCs are in quite good agreement with two data sets of flight measurements taken from the projects Winter Icing Storms Project (WISP-94) and Ice in Clouds Experiment-Layer Clouds (ICE-L), which consider about 99 and 46 flight hours, respectively. It is important to stress that this comparison is less accurate than the previous one because the observations here are much more limited both in time and in space than the extensive observational data used for the cirrus regime. It should be also noted that the measurements actually concern

INPs. When the INP number is not high enough to deplete the ambient supersaturation, INP concentrations and ICNCs can correspond, however, it is well known that the two concentrations show discrepancies with increasing temperature because of the secondary ice formation (Kanji et al., 2017). Finally, ICNCs in Figure 5 (*right*) are in good agreement with the results of Heymsfield et al. (2013), also based on flight campaigns. They found that ICNCs decrease as temperature increases and are within the range $5 - 50$ L$^{-1}$ in the mixed-phase regime.

Besides the flight measurements, the recent ICNC estimates from lidar-radar satellite retrievals must be mentioned, e.g. Sourdeval et al. (2018) and Gryspeerdt et al. (2018). In particular, Gryspeerdt et al. (2018) analysed the behaviour of ICNCs within clouds as a function of temperature. Differently from Figure 5 (*left*), they showed that there is a weak temperature dependence of ICNC, which increases with decreasing temperature. On the other hand, similarly to Figure 5, they found a small increase of ICNC around $265 - 270$ K and, interestingly, a small peak at about 233 K due to orographic and frontal

regimes, which could explain our higher modeled ICNCs between 230 K and 240 K.

## 5   Conclusions

In this study we have implemented the ice nucleation scheme of Barahona and Nenes (2009) into the global chemistry-climate model EMAC. The parameterization takes into account the water vapour competition between homogeneous and heterogeneous nucleation and has been modified to consider also the depositional growth of pre-existing ice crystals. Heterogeneous nucleation

can be computed through different INP parameterizations, and we have chosen the empirical INP parameterization of Phillips et al. (2013) for our experiments. We have tested the BN09 scheme operating in the the cirrus and/or in the mixed-phase regimes and compared the results with the standard configuration of the model, which assumes that cirrus clouds form via pure

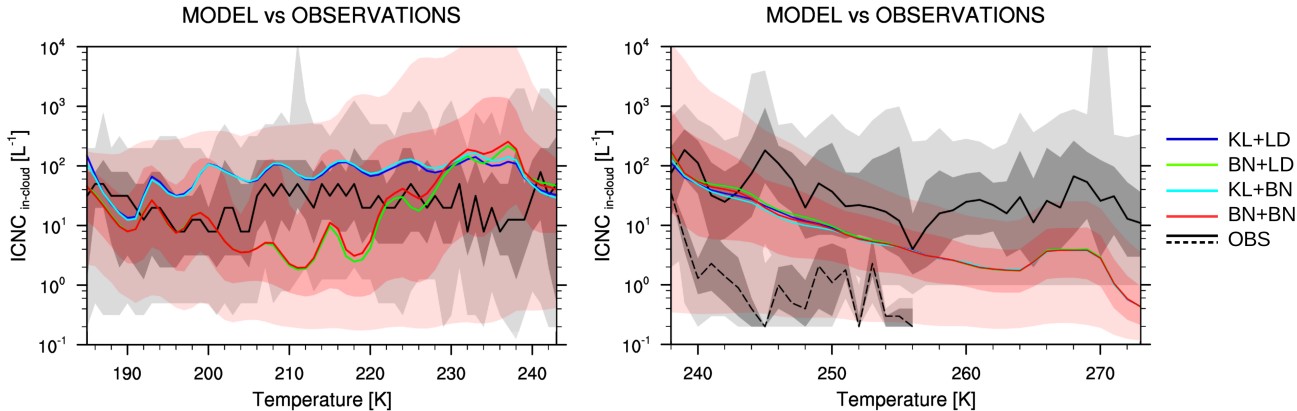

**Figure 5.** In-cloud ice crystal number concentrations ($ICNC_{in-cloud}$, $[\mathrm{L}^{-1}]$) versus temperature for modeled results and flight measurements. Medians are computed for modeled results (using daily means between $25°$S and $75°$N, masking $ICNC_{in-cloud} < 0.1\ \mathrm{L}^{-1}$, i.e. the minimum observed value) and observations, for each 1 K temperature bin. They are shown with colored lines: KL+LD (blue), BN+LD (green), KL+BN (light blue), BN+BN (red), and observations (black). Darker gray/red colors indicate the observations/BN+BN between $25^{th}$ and $75^{th}$ percentiles, while lighter gray/red colors indicate the observations/BN+BN between $5^{th}$ and $95^{th}$. (*Left*) Cirrus regime: the modeled medians are computed approximately in the range of $4-20$ km height, the observations come from Krämer (*personal communication, not yet published*). (*Right*) Mixed-phase regime: the modeled medians are computed approximately in the range of $0-20$ km height, the observations belong to the projects WISP-94 (solid line) and ICE-L (dashed line) and concern INP concentrations.

homogeneous nucleation (Kärcher and Lohmann, 2002) and uses the immersion nucleation parameterization of Lohmann and Diehl (2006) for mixed-phase clouds.

Focusing on the ice-related results, e.g. ICNC and IWC, we found that using BN09 in the cirrus regime strongly reduces the total ICNC worldwide because of the competition and PREICE effects, however, increases ICNC along the tropics. In contrast,
5 BN09 in the mixed-phase regime produces slightly higher ICNCs, especially in the NH where mineral dust particles are more abundant. We found that changing the ice nucleation scheme in the cirrus regime generates larger differences of ICNC and IWC than changing parameterization in the mixed-phase regime, that is the simulations using the same parameterization in the cirrus regime (e.g. BN+LD and BN+BN) are easily discernible from the others (LD+KL and LD+BN). Interestingly, we observed a certain dependence of ICNC and IWC in the mixed-phase regime on the parameterization used for cirrus clouds,
10 likely due to a faster sedimentation of larger ice crystals produced by BN09 in cirrus clouds at higher altitudes.

Overall, all modeled results agree well with global observations and the literature data. The comparison made with flight measurements has pointed out that ICNCs are overestimated by KL02 in the cirrus regime. BN09 agrees well with the observations in cold cirrus clouds, however, the PREICE effect is likely overestimated causing the underestimation of ICNCs between 200 K and 220 K.

15 As BN09 takes into account additional processes which were previously neglected by the standard version of the model, without consuming extra computational resources, we recommend to apply this ice nucleation scheme in future EMAC sim-

ulations. We also suggest to select P13 among the INP parameterizations available in BN09, since it incorporates the ice-nucleating ability of different aerosol species (dust, soot, bioaerosols, and soluble organics) and simulates both deposition and immersion/condensation nucleation. By using the configuration BN+BN, the EMAC model becomes one of the few GCMs which take into account in a detailed manner the complexity of ice nucleation. Finally, this work offers further material for

future GCM comparisons with focus on ICNC estimates and for future modeling evaluations against flight measurements and lidar-radar satellite retrievals.

*Code and data availability.* The Modular Earth Submodel System (MESSy) is continuously further developed and applied by a consortium of institutions. The usage of MESSy and access to the source code is licensed to all affiliates of institutions, which are members of the MESSy Consortium. Institutions can become a member of the MESSy Consortium by signing the MESSy Memorandum of Understanding.

More information can be found on the MESSy Consortium Website (http://www.messy-interface.org). All code modifications presented in this article will be included in the next version of MESSy.

**Appendix A**

In this Appendix we provide some additional technical information about the implementation of BN09 into the EMAC model. The BN09 parameterization has been added as a Fortran95 module in the submodel core layer (SMCL) of MESSy (named

as messy_cloud_ice_BN09.f90). BN09 operates in the cirrus regime and/or in the mixed-phase regime according to the calls made in the CLOUD submodel (messy_cloud_lohmann10.f90). As shown in Figure 1, BN09 computes the newly formed ice crystals in the cirrus regime when nicnc=3 and in the mixed-phase regime when limm_BN09=.TRUE., where nicnc and limm_BN09 are variables defined in the namelist-file cloud.nml (the setup of cloud.nml for the simulation BN+BN is shown in Table S1 as an example).

Other changes made during the implementation are the following ones.

- *Temperature threshold.* The original BN09 assumes the value $235\ K$ as temperature threshold between the two regimes, while the CLOUD submodel uses the value $238.15\ K$. For consistency, we used the second threshold as limit condition to call BN09, and we changed the original threshold of BN09 to the value $238.15\ K$ inside the BN09 code.

- *Number concentration and diameter of cloud droplets.* The original BN09 computes the cloud droplet number concen-

25       tration starting from the number concentration of sulfate aerosol in the Aitken mode. However, since the EMAC model computes the activated cloud droplet number concentration via other parameterizations (e.g. Abdul-Razzak and Ghan, 2000; Lin and Leaitch, 1997; Karydis et al., 2017), we provide BN09 with such variable (neglecting the corresponding computations inside the BN09 code). Unfortunately, these parameterizations do not compute the diameter of the new cloud droplets, therefore, BN09 still computes the diameter using the wet diameter of aerosol in the Aitken mode (i.e.

30       $D_c$).

*Competing interests.* The authors declare that they have no conflict of interest.

*Acknowledgements.* We would like to thank Dr. Mattia Righi from the German Aerospace Center (DLR) for the discussion on the modeled results. We acknowledge the usage of the Max Planck Computing and Data Facility (MPCDF) for the simulations performed in this work. Sylvia C. Sullivan and Athanasios Nenes acknowledge funding from a NASA Earth and Space Science Fellowship (NNX13AN74H), a

5    NASA MAP grant (NNX13AP63G), and a DOE EaSM grant (SC0007145). Moreover, Athanasios Nenes acknowledges funding by the European Research Council Consolidator Grant 726165 (PyroTRACH), Vlassis A. Karydis acknowledges support from an FP7 Marie Curie Career Integration Grant (project reference 618349), Holger Tost acknowledges funding from the Carl-Zeiss foundation, and Alexandra P. Tsimpidi acknowledges support from a DFG individual grand programme (project reference TS 335/2-1). Finally, we acknowledge the use of the programs Ferret (product of the NOAA's Pacific Marine Environmental Laboratory, http://ferret.pmel.noaa.gov/Ferret/) and NCL

10   (product of the Computational and Information Systems Laboratory at the NCAR, https://www.ncl.ucar.edu/) for the analyses and graphics in this paper.

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
