# Peer review of "Implementation of a comprehensive ice crystal formation parameterization for cirrus and mixed-phase clouds into the EMAC model (based on MESSy 2.53)"

_Geoscientific Model Development, 2018_

## Referee Comment (RC1) · Anonymous Referee #1 · 10 Apr 2018

**Review of „Implementation of a comprehensive ice crystal formation parameterization for cirrus and mixed-phase clouds into the EMAC model (based on MESSy 2.53)" by Bacer, S., Sullivan, S. C., Karydis, V. A., Barahona, D., Krämer, M., Nenes, A., Tost, H., Tsimpidi, A. P., Lelieveld, J., and Pozzer**

Summary:
Bacer et al. have implemented the ice nucleation parameterization by Barahona and Nenes (2009; BN09) into the EMAC model in both the cirrus and the mixed-phase cloud regime. For cirrus clouds this new parameterization accounts for a competition of water vapor by homogeneous and heterogeneous nucleation as well as by pre-existing ice crystals. For heterogeneous nucleation in cirrus and mixed-phase clouds the ice nucleating parameterization by Phillips et al. (2013) is selected (other ice nucleating parameterizations are available in the model). Dust and soot are used as ice nucleating particles to compare BN09 to the previous ice nucleation parameterization in EMAC. The deposition of water vapor on pre-existing ice crystals as well as heterogeneous nucleation in BN09 lead for high altitude cirrus clouds leads to an decrease in ICNC (and IWC) and an increase for lower altitude cirrus clouds compared to the previous ice nucleation parameterization which accounts only for homogeneous nucleation in cirrus clouds. For mixed-phase clouds BN09 seems to lead to an increase in ICNC compared the previous ice nucleation parameterization. A comparison of in-cloud ICNC to aircraft observations shows a better simulation at T<205K with BN09 but an underestimation of ICNC in the temperature range 205-222 K. For warmer temperatures there is no difference in ICNC of the simulations using different ice nucleation parameterizations.

General comment:
The ice nucleation parameterization implemented in the EMAC model in this study accounts for previously not represented processes and is therefore a meaningful contribution to GMD. But some details of the model parameterizations, configuration and the results are not discussed in enough detail (see specific comments below). Also the shortwave and longwave cloud radiative effects of the default simulation are too strong compared to observations. Therefore I recommend a major revision of the manuscript.

Specific comments:
P4: Give more details about the model. How are convective clouds treated? The two-moment cloud microphysics scheme in ECHAM also handles the freezing of the detrained condensate of convective clouds. Also give details how dust is computed. If dust emissions are computed online they could be quite variable between simulations. Describe how clouds and aerosol-particles interact. Droplet formation is mentioned later but should already be mentioned here. Which of the aerosol modes/species are used in the activation parameterization?
P7L29-P8L1: Soot particles are considered as ice nucleating particles (INP) for cirrus clouds (T<238 K). Whether soot particles initiate freezing at these cold temperatures and at supersaturations below the threshold for homogeneous nucleation is controversial (Kanji et al., 2017). The motivation and impact for choosing soot particles as INP for cirrus clouds need to be discussed.
P8 equations (4-5): Is the number of existing ice crystals subtracted from N_i,het or are soot and dust particles removed from the interstitial aerosol after heterogeneous nucleation? If not the INP could "freeze" several times leading to unrealistically high ICNC.

P8L17: Is the dry diameter of sulfate in the Aitken soluble mode used or the dry diameter of the Aitken soluble mode? How is the dry diameter of sulfate in the Aitken soluble mode computed? How sensitive is BN09 to this choice of INP diameter?

All figures showing zonal and annual means (Figs. 2-4,S1-3): These figures need to show some measure of significance.

P10L7-10: Why is TKE higher at lower altitude in BN+LD? Could the changes in the mixed-phase regime in Fig. 2b,f be due to increased sedimentation of larger ice crystals from cirrus clouds?

P11L1-2: What is the explanation for this decrease in IWC while ICNC increase? Are this changes significant?

P11L8-9 and L16-17: The reason for the high ICNC concentrations in the Himalaya region and Antarctica (e.g. Fig. 2 or Fig. S4) is not discussed. Due to the coarse resolution of the simulations, the topography may not be resolved well. Using a high resolution topography dataset, Gryspeerdt et al. (2017) identify cirrus clouds over Antarctica as primarily synoptic cirrus clouds not primarily orographic cirrus clouds.

P15L1-2: These high values of SCRE and LCRE in the default simulation are surprising. As can be seen from your Table 2, the observed values of both, SCRE and LCRE are lower. The default simulation needs to be retuned to better match the observed values. Is this simulation in radiative balance at the top of the atmosphere? If not the comparison of CRE of the different simulations to observations is not very meaningful. Add the net radiative balance at the top of the atmosphere to Table 2.

P16L1: Are some of the quantities in Table 2 tuned to agree with observed values (Mauritsen et al. 2012, Hourdin et al, 2016)?

P16L5: It is mentioned previously in the text that homogeneous nucleation dominates in the tropics and in the SH, whereas heterogeneous nucleation is important in the NH. Would it be possible to split the observations and the analysis in this section into the tropics and the NH extratropics?

P17L3-4: What is the reason for the better performance of BN09 compared to KL02 at low temperatures? I would assume that both schemes compute homogeneous nucleation at these low temperatures and that the vertical velocities are similar.

P17L25-26: In Fig. 2b for example an increase in ICNC in the mixed-phase regime is shown when using BN09 in the cirrus regime. How does this agree with the similarity of the ICNCs of the different simulations in the mixed-phase regime compared to the aircraft measurements?

P17L26-28: Give references for WISP-94 and ICE-L. Why are these two datasets so different (the 25[th] to 75[th] percentile do not overlap)?

P19L5-7: I agree with your point (1) and (3) but the general better performance of BN09 compared to the default parameterizations is not conclusively shown. While BN09 performs better at T < 205K, there will be fewer and optically thinner clouds at these low temperatures than at the temperature range 205-222K where BN09 agrees less well with aircraft observed in-cloud ICNC than the default parameterizations. In my opinion the additional processes computed by BN09 outweigh this drawback and BN09 should be used in future EMAC simulations but a generally better performance cannot be asserted.

P19L31-P20L2: As this is interstitial aerosol, at what relative humidity is the wet diameter of the sulfate aerosol in the Aitken mode computed?

Technical corrections:

P1L5: Only one of the multiple ice nucleating particle spectra is applied in one simulation. Rephrase the sentence as it reads now as if the multiple ice nucleating particle spectra are applied simultaneously. Also INP spectra should be replaced by INP parameterization.

P2L7: The greenhouse effects dominates for cirrus clouds e.g. Chen et al. (2000).

P2L9: Mixed-phase clouds can also occur at colder temperatures, rephrase.

P2L10-11: This is only true when deep convective clouds are included in the term mixed-phase clouds, but deep convective clouds are often named separately. As you here include deep convective clouds, this should be mentioned explicitly.

P2L11-12: Provide here references such as McCoy et al. (2016).

P2L25: Explain what you mean here by "the overestimation of vertical velocity".

P2L35-P3L1: This sentence is true for mixed-phase clouds while the sentences before and afterwards concern cirrus clouds. This is confusing, rephrase or move this sentence.

P3L12: Do you mean numerical parcel model simulations?

P6L21-22: Is this reduction done only for cirrus clouds are also for mixed-phase clouds?

P6L33: Is the cloud parcel mentioned here explicitly computed in EMAC or do the equations (1-6) provide analytical solutions for the cloud parcel?

P9L10-14: Provide references for the anthropogenic aerosol emissions and describe how natural aerosols (e.g. dust) are treated.

P8L6: Is $n_x$ the number of interstitial aerosol particles or is the number of aerosol particles in cloud droplets tracked? Please clarify.

P10L26-P11L1: Do you mean that IWC decreases where ICNC decreases?

P11L32-P13L2: The increase in IWC in equatorial regions at 200 hPa is about 5-10% (Fig. 2), I would not call this dramatic.

P13L21-23: It should be mentioned that observations of cloud droplet number concentration are uncertain (Bennartz and Rausch, 2017).

P13L27-29: The underestimation of IWP was found in previous studies using ECHAM-HAM. The IWP in ECHAM is not underestimated see e.g. Mauritsen et al. (2012).

P17L28-30: Please add the number of hours in mixed-phase clouds.

P17L30: When the measurements are for INP this needs to reflected in Fig. 5 itself (at least in the figure caption).

P18L5: Use INP parameterization instead of INP spectrum.

Caption of Fig. S4: Give references for the observational datasets.

Fig. S5 is a table not a figure.

References:

Bennartz, R. and Rausch, J.: Global and regional estimates of warm cloud droplet number concentration based on 13 years of AQUA-MODIS observations, Atmos. Chem. Phys., 17, 9815-9836, https://doi.org/10.5194/acp-17-9815-2017, 2017.

Chen, T., Rossow, W. B., and Zhang, Y.: Radiative Effects of Cloud-Type Variations, J. Climate, 13, 264–286, https://doi.org/10.1175/1520-0442(2000)013<0264:REOCTV>2.0.CO;2, 2000.

Gryspeerdt, E., Quaas, J., Goren, T., Klocke, D., and Brueck, M.: Technical note: An automated cirrus classification, Atmos. Chem. Phys. Discuss., https://doi.org/10.5194/acp-2017-723, in review, 2017.

Hourdin, F., Mauritsen, T., Gettelman, A., Golaz, J. C., Balaji, V., Duan, Q., Folini, D., Ji, D., Klocke, D., Qian, Y., Rauser, F., Rio, C., Tomassini, L., Watanabe, M., and Williamson, D.: The art and science of climate model tuning, B. Am. Meteorol. Soc., doi:10.1175/BAMS-D-15-00135.1, 2016.

McCoy, D. T., I. Tan, D. L. Hartmann, M. D. Zelinka, and T. Storelvmo (2016), On the relationships among cloud cover, mixed-phase partitioning, and planetary albedo in GCMs, J. Adv. Model. Earth Syst., 8, 650–668, doi:10.1002/2015MS000589.

---

## Referee Comment (RC2) · Anonymous Referee #2 · 10 Apr 2018

**Review of "Implementation of a comprehensive ice crystal formation parameterization for cirrus and mixed-phase clouds into the EMAC model (based on MESSy 2.53)"**

This study describes the implementation of a new ice nucleation scheme for both mixed-phase and cirrus clouds. In particular, its cirrus part includes the previously missing heterogeneous freezing on ice nucleating particles and the effect of vapour deposition on pre-existing ice.

I can imagine the manuscript is a result of hard modelling work and I believe it deserves to be published, but only after (most of ) the comments are addressed. I find the manuscript's results OK, however the authors very often only superficially describe the plotted results without explaining their causes. As the manscript presents the implementation of previously developed schemes I would really expect the authors to make the additional step forward and try to better understand their results. Moreover, I would suggest to better structure some of the introductory parts of the text.

I added many specific comments. Their main point is, however, not to demotivate the authors, but to try to help them substantially improve the quality of the manuscript.

**A summary of major points**

- Please try to understand your results in a larger detail and provide more information when needed!

- How is convection simulated in your model. How do you deal with detrainment from deep convection? Never mentioned in the paper, despite we also clearly see some responses of convection to the modifications in ice nucleation schemes.

- The introduction should be written in a more coherent way, in particular the mixed-phase part. See detailed comments.

- Keep in mind you are comparing two schemes which should already by construction give you different results for cirrus clouds as one simulates only homogeneous freezing, while the other includes also heterogeneous freezing and pre-existing ice effect.

- considering that you don't write only about cirrus, but also about mixed-phase conditions, you could very briefly mention how other relevant processes work in your model, e.g. Wegener-Bergeron-Findeisen process, secondary ice formation, autoconversion, etc.

- I would find it valuable to also briefly mention how your results compare to results from studies using the same/similar ice nucleation scheme in CAM5.

- You should write a stronger conclusion, which can be relevant also to people other than MESSY model users. I also don't think the authors made a strong enough point to convince MESSY model users who don't focus on clouds to use the new ice nucleation scheme.

**Specific comments**
**(a mixture of a few major and many minor comments)**

**Title**
- title sounds too technical: is there really a need to add *"(based on MESSy 2.53)"* . As a suggestion you could further simplify the title to something like: Implementation of a new ice phase/ice cloud parameterization in the EMAC model

**Abstract:**

Needs to be rearranged, right now is in my opinion a bit out of a logical order. I suggest:
1.) mention that you implemented BN09 for both cirrus and mixed phase clouds
2.) Only now go in details of homog. vs heterog. nucleation, aerosols, etc.

Some minor comments:
line 2: *realistically* represent => quite a bold statement
line 4: cold clouds => never defined it
lines 4-6: the sentence starting with "Furthermore" is hard to understand. Please rewrite!
line 7: Compared to the standard EMAC *results…* => …
line 10: …improves the model results…. => too vague, be more concrete, which results?

**Introduction**

The first paragraph sounds like ice nucleation and droplet activation are the only two challenging processes in the representation of clouds. Is this true?

line 15: *…clouds remain one of the most elusive components of the atmospheric system…*
I think the word elusive isn't used in a correct way here

page 2

- Wouldn't it be more logic to start with mixed phase clouds? After all, they have a larger radiative impact on climate and (regionally) on cloud feedbacks than cirrus.
- you never defined what a "mixed-phase cloud" is

-

line 4: *typically below -35℃* => why typically? depends on your definition, as there is no global definition of what a cirrus cloud is. So if you in your manuscript go for the -35°C threshold just say that firmly).
+ here you use Celsius, while throughout the whole text Kelvin. Be consistent.
(I personally don't see any advantage of using Kelvin over Celsius, but that's purely a matter of personal taste)

line 6: missing references on the radiative role of cirrus – maybe Matus and L'Ecuyer 2017, Hong et al. 2016 for some recent satellite estimate of their radiative effects or even Kienast-Sjogren et al. 2016 for lidar-based estimates, Gasparini and Lohmann 2016 for GCM modelling-based estimates.
The first two references could be cited in the context of mixed-phase CRE too.
line 10: missing some references on mixed-phase being TD unstable, or similar

The section on mixed-phase is in a poor state. It deserves at least  1-2 sentences more, giving reference for the listed processes/facts (e.g. that mixed-phase, if we define it just by temperature threshold, is probably responsible for most/a large share of precipitation, which is different from the cloud top phase classification of Mulmenstadt et al. 2015).
Are mixed phase responsible for lightning and storms? Aren't convective clouds treated by a different scheme in your model?
What do you mean by "*strong storms*". That's all written in a to ambiguous way for a scientific paper.

line 12: *The fraction of cloud ice has a profound impact on the cloud forcing in global climate models*: there's tons of references on that, why didn't the authors include any? (e.g. Tan et al., 2016 , Science, studies looking more specifically into the Southern Ocean like Vergara-Temprado et al, 2018, PNAS and many more)

*Based on modeling studies, homogeneous nucleation has been considered the dominant process for cirrus formation (e.g. Haag et al., 2003; Gettelman et al., 2012) because the concentration of liquid droplets is higher than that of INPs in the upper troposphere. However, due to the overestimation of vertical velocity this is under debate (Cziczo et al., 2013; Barahona and Nenes, 2011; Barahona et al., 2017).*

I don't think the upper 2 statements are totally correct, possibly due to a too condensed information. Early observational studies of cirrus clouds were affected by the problem of ice crystal shattering, which implied several times too large ice crystal number concentrations. Such numbers were hard to explain other than with homogeneous nucleation, and were also replicated by model studies.

Moreover, we have also numerous modelling studies (ok, Barahona et al. being one of them) showing that heterogenous nucleation might play a role in cirrus, for example: Sullivan et al., 2016, Storelvmo and Herger 2014, Penner et al. 2015, Gasparini and Lohmann 2016. I don't think there is a universal agreement on the

overestimation of vertical velocities by GCMs. A study by Joos et al. 2008 and Kärcher and Ström 2003 show a good agreement between vertical velocity observations and model updrafts. The updrafts were based on the large scale updraft and a TKE based term, which was in Joos et al. 2008 over mountains replaced by gravity waves. As Joos et al. use the same (I guess) dynamical core than the described model, we can imagine that the TKE based updrafts could be in line with observations. And Cziczo et al., 2013 also isn't talking about updraft overestimation, despite being cited for it.

Lines 26-30:
The following (very long) sentence should appear earlier in text as it defines the two ice crystal formation regimes.
*"Overall, two different regimes for ice crystal formation are distinguished: the mixed-phase regime at subfreezing temperatures between 238 K and 273 K, where ice crystals form exclusively by heterogeneous nucleation and alter the phase composition of the mixed-phase clouds, and the cirrus regime at colder temperatures (T < 238 K), where ice crystals originate via heterogeneous and/or homogeneous nucleation to form cirrus clouds. "*

end of page 2, beginning of page 3: I am missing a description of freezing in mixed-phase clouds? Why do you always refer only to cirrus, if you implemented freezing also at mixed-phase conditions?

page 3

line 10: Can you find some evidence/reference for the following sentence:
*"Including sophisticated schemes in general circulation models (GCMs) allows for a more realistic description of the variability of cloud properties and cloud radiative effects, improving the model climate predictions."*

line 18: please explain what INP spectra mean. I assume that's simply a parameterization of het ice nucleation?

line 29: *"…has been compared with the results generated via the standard model configuration"*
What kind of scheme does your standard version of the model use?

**Model description and set-up of simulations**

Convection plays a large role in global high cloud distributions and their properties. You should include some more information on how the CONVECT submodel interacts with the microphysics and cloud cover. I add some questions which could be addressed:
- How does the convective detrainment works?
- How do you compute/parameterize the size of ice crystals that are detrained from convective clouds?
- How does the scheme decide whether you detrain liquid or water (or even vapour)?

page 4

You previously defined ice crystal number concentration as ICNC. Here, you define it again as $N_i$. Please, be consistent!

page 6

This cannot be a separate paragraph:
*"Finally, the influence of the pre-existing ice particles is not taken into account. The only precaution adopted by the CLOUD submodel is the reduction of the number of aerosol particles available for ice nucleation by the existing ice particle number."*

+What do you mean with "*the only precaution*"?

lines 23-25: The text between points **2.3** and **2.3.1** is repeating the information already given before. Please remove it.

lines 26-29: You already provided the same information on page 3. Please try to avoid repetition!

page 7

line 26: Not sure that you can assume that P13 agrees better with observations in every model (thinking that vertical velocities might be different than in CAM)

**2.3.2 Implementation**

page 8

line 21: Did you define what *"M modes"* are?

line 24-25: *"They are weighted over a Gaussian updraft velocity distribution, with mean 0.1 cm s$^{-1}$ and standard deviation equal to $w_{sub}$, in order to account for the sub-grid variability (Sullivan et al., 2016) "*
Could you describe that a bit better as it is not a standard procedure in GCMs?

page 9

lines 3-4: "*Overall, BN09 is a scheme more realistic than KL02 and LD06 which improves the ice nucleation in EMAC by taking into account processes which were previously neglected (e.g. water vapour competition, influence of polydisperse aerosols, PREICE effect).*"

I think that doesn't fit in the model description part of the paper but in the results.

**3 Model results**

lines 3-4: Not sure about that. I think your Figure 3a makes me think it is not INPs but mountains that contribute most to the larger ICNC in the northern hemisphere.

Please indicate which areas are significantly different from the "DEF" case in Figures 2 and 3 by applying an appropriate statistical significance test! Same for plots S1, S2, S3.  Add +/- 1 or 2 st. deviation shading to the lines plotted in S4. You could also tentatively try to plot the 25th and 75th percentile range in Figure 5, maybe only for 1 setup due to clarity (BN+BN, I would suggest).

Please also add standard deviations to your Table 2 for a better feeling of the magnitude of changes due to changing microphysics!

ps. Do you show in-cloud or all-sky ICNC and IWC values on your figures? Mention it somewhere in text!

lines 6-7: "*This is likely due to the PREICE effect predicted by BN09, as it has been shown that BNhom and KL02 produce the same order of magnitude of ICNC (Barahona and Nenes, 2008).* "

Moreover, KL02 simulate only homogeneous nucleation, while BN09 simulate also heterogeneous nucleation at cirrus conditions. Therefore, you should point out somewhere that you are not really making an apples-to-apples comparison.

Lines 7-9: *On the other hand, ICNCs increase at lower altitudes and especially in the NH. This is due to higher TKE at lower altitudes, which impacts the updraft velocity and increases heterogeneous nucleation contribution.*

How was that done before in the REF case? Did you use only large-scale updraft? Do you consider a Gaussian distribution of vertical velocities (Sullivan et al., 2016) also in mixed-phase conditions?

line 10 and further: "*Indeed…*"

First you talk about cirrus, than mixed-phase, now cirrus again, I guess. That's confusing for the reader, which expects this sentence to refer to mixed-phase clouds. Please reorder or clarify better!

Lines 16-17: missing citation(s) at the end of the following sentence: *"Overall, the ICNC differences obtained using the various ice schemes in the mixed-phase regime are smaller (mostly within ±20%) than in the cirrus regime."*

Lines 18-20: please rephrase (cirrus don't occur throughout the year? where, why not?…)

Ice nucleation in mixed-phase may not be the main source of IWC and ICNC between 0 and -38°C. Could you estimate that from your model and comment on

that? Possible processes that might not be negligible are for instance sedimentation of ICs from cirrus or detrainment of IC from convection.

line 22: BN+LD case shows some differences with respect to DEF also in the mixed-phase regime (see Fig 2, Fig 3 f, also fig S1).
- Why is that when the mixed-phase freezing is the same? What other sources of ice exist in mixed phase?
- Can there be some radiative/dynamical/microphysical responses of mixed-phase to difference in cirrus scheme?
- There seems to be a response in convection in the tropics. Is this really the case? What caused it? Did the atmospheric stability change?

Please comment!

 [hint: by adding significance you might get by for some of the patterns by simply pointing out some differences aren't significant]

line 23: "*IWC decreases with increasing temperature, where ICNC is lower (Krämer et al., 2016),*

I don't understand what you mean with this sentence? If you look at upper troposphere, the opposite is true. While I agree with the following statement for regions between -30 and 0°C.

 *…and we find three areas with higher values over the mid-latitudes in both hemispheres and the tropics (Figure 2e).*"

I don't understand the connection with the first part of the sentence. I see you have 3 peaks of IWC which come out of your model, which is good, as the observations agree with it (please consult/refer e.g. to:
Li et al., 2012). What atmospheric features do the 3 peaks correspond to?

line 25:  "*…IWC is slightly higher because of the higher values of ICNC.*"

Does this always hold true? Add some supporting references at this point.

page 11

general comment on section 3.1

I often miss a more detailed discussion of why some of the changes occur, what caused them?  For a better process understanding I would recommend to add also zonally averaged figures of IC radii, RH (or $RH_{ice}$), temperature, for example also cloud droplet number concentration (helps for mixed-phase), and maybe cloud cover.

 **3.2 Global distributions**

You never mention why you decided for 200 and 600 hPa levels.

line 7: "*ICNCs in the cirrus regime mostly follow the precipitation pattern*"
What do you mean by precipitation patterns?
Can you also mention why does this happen, and why ICNC peaks also over mountains.

Does the ICNC global distribution compare well with recent observations by Sourdeval et al., 2018 and Gyrspeerdt et al., 2018?

"*The relative changes clearly show that BN09 used in the cirrus regime (Figure 3b, d) reduces ICNC (up to 60%) worldwide with respect to the default experiment, except over Indian and Indonesian areas*"
Again, I would like to see more explanations and not only description of figures. Why are India and Indonesia different from the rest of the world?

Line 13: "*Such a reduction occurs mostly because of the PREICE effect in the SH and the competition in the NH.*"
Is this only your speculation or do you have any evidence for it? Please show them!

Lines 16-17: "*At 600 hPa, ICNC increases towards high latitudes, in particular over Greenland (up to 2000 $L^{-1}$) and Antarctica (mostly > 2000 $L^{-1}$).*"
Why, please explain it!

Line 17: "*Interestingly, the ice nucleation scheme used in the cirrus regime affects the ICNC at the mixed-phase regime altitudes*"
I agree, that is very interesting, and therefore would be nice to understand what caused it!

Line 29-32:
"*Maritime updraft velocities are weaker, and recent work has shown that there are 30 important oceanic sources of INP (e.g. DeMott et al., 2016). These effects may combine to produce few large crystals in this Southern Pacific region.*"
What about the Intertropical Convergence Zone and peak of tropical convection in the Pacific warm pool area? Maritime aerosol cannot play a large role in a dynamically-driven detrained clouds. Moreover, you also did not include marine aerosols in the model, so I don't understand why you mentioned them.
Why don't you look at your particle radius and verify if the model is giving reasonable values in the tropical Pacific?

I guess 200 hPa is close to the level of maximum detrainment from deep convective clouds. It is therefore important to look at what size you assume for detrained ICs (I assume you use a 1-moment version of convective microphysics, so there needs to be more assumption to couple it to the stratiform microphysics).
Moreover, one of your coauthors showed how that the vertical velocities are quite high in the mentioned area (Barahona et al., 2017). I guess part of this is due to the prevailing large-scale ascent motion (quite noticeable in Joos et al.,

2008), while indeed a lot of it has to be connected to deep convection, and, in GCM modeling world, to TKE values.
Please explore that in larger detail!

page 13

line 1:
*"…but using BN09 in the cirrus regime dramatically increases IWC in equatorial regions at 200 hPa."*

Why is this the case? It would be extremely interesting to understand that, as this region plays a large role in global energy balance.
Did you change the model tuning in between? Can this happen due to changes in convection, which somehow responds to a different cirrus scheme?

On page 15 you even give a hint for that:
*"When BN09 is used in the cirrus regime, $P_{tot}$ grows by 4% especially because of the increase of the convective precipitation contribution (the large scale precipitation of all simulations remain almost constant)"*

**4 Model comparisons and observations**

Lines 4-9: This text doesn't fit into the results section, please move it to model description!

**4.1 Annual global means**

line 24: Please prove that a change of 7% is large by showing the variability (maybe add in table).

line 28: *…that applied ECHAM* => that used ECHAM-HAM

lines 15-19: You say that BN09 makes larger IC, but large scale precipitation doesn't change. That's surprising. Why?

line 21: *"The annual zonal mean profiles show clearly that the simulations using the same ice nucleation scheme in the cirrus regime are very close to each other, i.e. KL+LD and KL+BN, and BN+LD and BN+BN (as already visible in Table 2)."*

⇨ so all that hard work for nothing? Or what should I get from that?

page 16

Radiation changes for quite a bit, and this is probably a more important parameter for climate compared with ICNC, IWP, etc.

I would more strongly point SW, LW, and NET CRE anomalies, maybe even show

a lon x lat plot of them (with significance on it).

**4.2 Comparison with aircraft measurements**

line 11: Mention that you are talking about median values as means can be very different!

page 17

lines 3-4: "*From Figure 5 (left) we deduce that KL02 produces too low ICNCs in cold cirrus clouds (for T < 205 K) as well, while BN09 works better at such low temperatures*"

Isn't that interesting, considering that BN09 should give comparable results to KL02 for homogeneous freezing, while BN09 has also PREICE and heterogeneous freezing effects included. So one would rather expect just the opposite, BN09 to be lower than KL02. Why do we see the opposite? Are the results the same when comparing means instead of medians? Or do the vertical velocities calculated by the base model change for some reason between KL02 and BN09 schemes?

The comparison with aircraft data doesn't show BN09 as superior to the less realistic KL02, but rather the opposite. In particular, as there is only a small fraction of cirrus that reside at temperatures below 200 K (-73°C) in the area where most of the measurements come from (extratropics). Can you comment on that?

Did you make sure you are comparing apples-to-apples? For instance, GCMs normally simulate cirrus in the winter polar stratosphere, which might be responsible for parts a non-negligible fraction of the distribution. Better to remove them from the analysis. Also, did you normalize the model output based on the latitude not to give a too large meaning to the (numerous) polar gridpoints?

It would be interesting to look at a plot of vertical velocity in function of temperature, if you believe that to be (part of) the reason for differences between KL02 and BN09.

Lines 15-18: "*On the contrary, the simulations which consider only homogeneous nucleation in the cirrus regime show a large underestimation (even below the 5$^{th}$ percentile) at temperatures lower than 210 K, however, they are always within the observed 25$^{th}$ – 75$^{th}$ percentiles at higher temperatures.*

You show a large underestimation only below 200 K, while between 200 and 210 K both schemes seem to be comparably bad (hint on problems with vertical velocities???).

*ECHAM5 has strongly underestimated ICNC at low temperatures thus far (Kuebbeler et al., 2014), "*

Yes, but only at the very cold temperatures, which correspond only to a small fraction of cirrus and aren't the most relevant in terms of radiative (and in general climatic) impacts.

*"The implementation of BN09 has helped to alleviate this dramatic underestimation of cold cirrus ICNC (in agreement with Barahona et al., 2017). "*

As far as I recall, CAM modeling community undertook some efforts to decrease the overestimation of cold cirrus ICNC. Within various realizations of ECHAM, as it seems like, we have just the opposite problem. Too few ICNC at coldest cirrus conditions. It is not intuitive at all that such problems are alleviated by implementing a scheme, which should on average decrease the ICNC. I would love to read a discussion on this point in the corrected manuscript.

line 27: *"The simulations do not show any significant difference among each other, meaning that the parameterizations P13 and LD06 produce similar ICNC via pure heterogeneous nucleation."*

This now sounds different to discussions from section 3.2 (Figure 3, results for 600 hPa). Why?

line 28: Please add references for WISP-94 and ICE-L campaigns.

line 30: I think there's some datasets out there that extend to mixed phase temperatures. Look for instance into Heymsfield et al., 2013: Ice Cloud Particle Size Distributions and Pressure-Dependent Terminal Velocities from In Situ Observations at Temperatures from -8 to -86°C.

line 31-32: "*At mixed-phase conditions, the INP number is usually not so high that supersaturation is depleted before all particles have nucleated, so INP concentrations and ICNCs should generally correspond.*"

That isn't true for the warmer of the mixed-phase clouds. Figure 11 of the referenced Kanji et al., 2017 paper schematically illustrates that ICNC can also be higher than INP numbers due to secondary ice processes.

**Conclusions**

page 18

I would expect you to give a reason for the observed changes or lack of them after the line 15.

page 19

line 1-2: *"The comparison made with flight measurements has demonstrated that ICNCs are more realistically simulated when BN09 is used in the cirrus regime."*

This is not obvious from the data you show. Please, try to prove it in a

quantitative way with the help of some appropriate statistical methods, or rephrase the conclusions!

lines 5-9: Those are relatively weak conclusive words. Could you find some stronger statement on top of being able to include more processes in the model? Please think in the direction of why should anyone not using EMAC care about your manuscript (or consider citing it).

**References**
(in the order in which they appear in comments to the manuscript)

Matus and L'Ecuyer, 2017: The role of cloud phase in Earth's radiation budget
Hong et al., 2016: Assessing the radiative effects of global ice clouds based on CloudSat and CALIPSO measurements
Kienast-Sjögren et al., 2016: Radiative properties of mid-latitude cirrus clouds derived by automatic evaluation of lidare measurements
Gasparini and Lohmann et al., 2016: Why cirrus cloud seeding cannot substantially cool the planet
Mülmenstädt et al., 2015: Frequency of occurrence of rain from liquid-,mixed-, and ice-phase clouds derived from A-Train satellite retrievals.
Tan et al., 2016: Observational constraints on mixed-phase clouds imply higher climate sensitivity
Vergara-Temprado et al., 2018: Strong control of Southern Ocean cloud reflectivity by ice-nucleating particles
Sullivan et al., 2016: Understanding cirrus ice crystal number variability for different het. ice nucleation spectra
Storelvmo and Herger, 2014: Cirrus cloud susceptibility to the injection of ice nuclei in the upper troposphere
Penner et al., 2015: Can cirrus cloud seeding be used for geoengineering?
Joos et al., 2008: Orographic cirrus in the global climate model ECHAM5
Kärcher and Ström, 2003: The roles of dynamical variability and aerosols in cirrus cloud formation
Cziczo et al., 2013: Clarifying the dominant sources and mechanisms of cirrus cloud formation
Li et al., 2012: An observationally based evaluation of cloud ice water in CMIP3 and CMIP5 GCMs and contemporary reanalyses using contemporary satellite data, look for CIWP
Sourdeval et al., 2018: Ice crystal number concentration estimates from lidar-radar satellite remote sensing. Part 1: Method and evaluation
Gyrspeerdt et al., 2018: Ice crystal number concentration estimates from lidar-radar satellite retrievals . Part 2 : Controls on the ice crystal number concentration
Heymsfield et al., 2013: Ice Cloud Particle Size Distributions and Pressure-Dependent Terminal Velocities from In Situ Observations at Temperatures from -8 to -86°C

---

## Author Comment (AC1) · 5 Jun 2018

We really thank the anonymous Referee for the very constructive comments. Below we report our replies.

**SPECIFIC COMMENTS**

**P4:** Give more details about the model. How are convective clouds treated? The twomoment cloud microphysics scheme in ECHAM also handles the freezing of the detrained condensate of convective clouds.

As the Referee #2 has raised the same issue, the following information about convection has been added in Section 2.1.

"The CONVECT submodel contains multiple convection parameterizations (Tost et al., 2006). In this work the scheme of Tiedtke (1989) has been used. Convective cloud microphysics is highly simplified and neither explicit aerosol activation into liquid droplets nor aerosol effects in the ice formation processes are taken into account, i.e. convective microphysics is solely based on temperature and updraft strength. Detrainment from convection is treated by taking updraft (and downdraft) concentrations of water vapour and cloud condensate and the corresponding massflux detrainment rates into account. These are merged including turbulent detrainment (i.e. exchange of mass through the cloud edges) and organised detrainment (i.e. organized outflow at cloud top). The detrained water vapour is added to the large-scale water vapour field, while the detrained cloud condensate is directly used as a source term for cloud condensate by the large-scale cloud scheme (i.e. the CLOUD submodel), which considers the detrained condensate either liquid or ice depending on the temperature (if T < 238 K the phase is ice) and the updraft velocity. The size and numbers of the detrained condensate are not taken into account explicitly."

**P4:** Also give details how dust is computed. If dust emissions are computed online they could be quite variable between simulations.

Offline dust emissions from the AEROCOM data set were used in all simulations. We added this information in Section 2.4 (lines 9-13, P9), rather than at P4.

"All simulations have been run for 6 years (1 year as spin-up time plus 5 years for the analysis) using emissions starting from the year 2000 (GFEDv3.1 from van der Werf et al., 2010 for biomass burning and CMIP5-RCP4.5 from Clarke et al., 2007 for anthropogenic emissions). As in Pozzer et al. (2012), dust is offline prescribed using monthly emission files based on the AEROCOM data set (Dentener et al., 2006). Also volcanic and secondary organic aerosol emissions are based on AEROCOM, while GFEDv3.1 and CMIP5-RCP4.5 have been used to simulate emissions of black carbon and organic aerosols, respectively. Finally, aerosol climatologies have been used for the interactions with radiation (Tanre et al., 1984) and heterogeneous chemistry (Aquila et al., 2011). Prescribed climatologies of sea surface temperatures (SST) and sea-ice concentrations (SIC) from AMIP (30 years: 1980-2009) have been used as boundary conditions."

**P4:** Describe how clouds and aerosol-particles interact. Droplet formation is mentioned later but should already be mentioned here. Which of the aerosol modes/species are used in the activation parameterization?

We added the following information in Section 2.1, P4 (and we reduced lines 17-19, P9).

"Cloud droplet formation is parameterized by the "unified activation framework"

(UAF) (Kumar et al., 2011; Karydis et al., 2011). It is an advanced physically based parameterization which merges two theories:  $\kappa$ -Köhler theory (KT) (Petters and Kreidenweis, 2007), which governs the activation of soluble aerosols, and Frenkel-Halsey-Hill adsorption activation theory (FHH-AT) (Kumar et al., 2009), which describes the droplet activation due to water adsorption onto insoluble aerosols (e.g., mineral dust). Aerosol modes that consist of only soluble material follow the KT, and the required effective hygroscopicity ( $\kappa$ ) is calculated based on the chemical composition of the mode as described by the ISORROPIA thermodynamic equilibrium model (Fountoukis and Nenes, 2007). Aerosol modes that consist of an insoluble core with soluble coating follow the UAF scheme, which takes into account the effects of adsorption and absorption on the cloud condensation nuclei (CCN) activity of the mixed aerosol. More details about the UAF scheme and its implementation in the EMAC model can be found in Karydis et al. (2017)."

P7L29-P8L1: Soot particles are considered as ice nucleating particles (INP) for cirrus clouds (T<238 K). Whether soot particles initiate freezing at these cold temperatures and at super-saturations below the threshold for homogeneous nucleation is controversial (Kanji et al., 2017). The motivation and impact for choosing soot particles as INP for cirrus clouds need to be discussed.

We chose DU and BC as INPs for a technical reason explained probably too poorly at lines 25-27, P9. Phillips et al. (2013, P13) can consider the contribution of four species DU, BC, BIO, and soluble OC to immersion/condensation and deposition nucleation modes. However, the default configuration of EMAC (i.e. simulation KL+LD) takes into account only the contribution of DU and BC to immersion nucleation in the mixed-phase regime via LD06. Therefore, we decided to include only these species (DU and BC) for the computations of P13.

We modified the sentence at line 1, P8: "Dust and soot, the aerosol species considered in this work for the reasons explained in Section  $2.4, \dots$ ".

In Section 2.4, we made the last paragraph preciser: "The P13 parameterization is used to simulate deposition and immersion/condensation nucleation whenever BN09 is called (for the reasons explained in Subsection 2.3). Since LD06 takes into account only dust and soot for immersion nucleation, we set the same aerosol species as contributions for P13 and turned off the biological and organic contributions."

**P8:** equations (4-5): Is the number of existing ice crystals subtracted from  $N_{i,het}$  or are soot and dust particles removed from the interstitial aerosol after heterogeneous nucleation? If not the INP could "freeze" several times leading to unrealistically high ICNC.

The number of new ice crystals formed heterogeneously is not subtracted to the interstitial aerosol, however, the reduction of aerosols is taken into account by the SCAV submodel, which simulates the nucleation scavenging. This means that the number of aerosols available for ice nucleation is "updated" by SCAV and there is no risk of counting several times the same particles as INP.

P8L17: Is the dry diameter of sulfate in the Aitken soluble mode used or the dry diameter of the Aitken soluble mode? How is the dry diameter of sulfate in the Aitken soluble mode computed?

How sensitive is BN09 to this choice of INP diameter?

Thanks for pointing it out. As aerosols are internally mixed, the diameters are only computed for different aerosol modes (not for different aerosol species), thus, we actually used the dry diameter of the Aitken soluble mode (not the diameter of sulfate). We corrected lines 17, P8 and 1, P20.

The dry diameter of the Aitken soluble mode is computed by the aerosol model GMXe. It is then used by BN09 to compute the diameter of cloud droplets using an approximation (linearly dependent on the dry diameter of the Aitken soluble mode) derived from the equilibrium calculations proposed by Lewis et al. 2008.

Based on few tests using the BN09 offline, the dependency to the diameter is very weak.

All figures showing zonal and annual means (Figs. 2-4,S1-3): These figures need to show some measure of significance.

We estimated the statistical significance using the Welch's t-test and we marked the areas with 95% level of significance in all plots which show relative percentage changes. All figures were modified accordingly.

**P10L7-10:** Why is TKE higher at lower altitude in BN+LD?**

Could the changes in the mixed-phase regime in Fig. 2b, f be due to increased sedimentation of larger ice crystals from cirrus clouds?

The sentence regarding TKE does not explain why there is a positive bias in the mixed-phase regime in the comparison of BN+LD with KL+LD (i.e. Figure 2b), therefore we removed such sentence.

As noticed by the Referee, BN09 produces larger ice crystals which sediment faster from cirrus clouds, thus, lines 8-10 were modified as follows.

"Interestingly, ICNCs at lower altitudes are also influenced by the ice nucleation parameterization used in the cirrus regime. In fact, there is an increase of ICNCs in the mixed-phase regime probably due to a faster sedimentation of the larger ice crystal produced by BN09 in cirrus clouds, especially in the NH where there are larger sources of efficient ice-nucleating mineral dust."

**P11L1-2:** What is the explanation for this decrease in IWC while ICNC increase? Are this changes significant?

The new plots (with significance levels) show that, in the mixed-phase regime, the IWC decrease and the ICNC increase overlap and are significant only in a small area, at high latitude and around 700 hPa. A possible reason could be attributed to the different ice crystal sizes in this area.

Lines 1-4, P11 were changed: "On the other hand, IWC in KL+BN slightly reduces (up to 20%) in the mixed-phase regime in areas where ICNC increases, especially in the NH at high latitudes (Figure 2g). This could be due to the the different sizes of ice crystals, however, the areas with significance are rather small."

P11L8-9, L16-17: The reason for the high ICNC concentrations in the Himalaya region and Antarctica (e.g. Fig. 2 or Fig. S4) is not discussed. Due to the coarse resolution of the simulations, the topography may not be resolved well. Using a high resolution topography dataset, Gryspeerdt et al. (2017) identify cirrus clouds over Antarctica as primarily synoptic cirrus clouds not primarily orographic cirrus clouds.

Naturally, the resolution of the topography is quite coarse, about 300 km x 300 km at the equator, and both the Himalayan region and Antarctica are represented as wide and very high plateaus (see Figure 1 of this document). The high values of ICNCs over the Himalayan region (and Andes) at 200 hPa are related to the high values of

the turbulent contribution to the vertical velocity ( $w_{sub}$ , Figure 2-left of this document). On the other hand, the high values of ICNCs over Antarctica at 600 hPa can be related to the high values of both  $w_{sub}$  (Figure 2-right) and ice supersaturation in this area. We discussed it at lines 9-8 and 16-17, P11.

"ICNCs in the cirrus regime (Figure 3a) show areas with high values over land and in correspondence with mountainous regions, e.g. the Rocky Mountains, Andes, and Tibetan Plateau with ICNCs > 500 L-1. Such pattern is strongly related to the turbulent contribution of the vertical velocity  $w_{sub}$  and in agreement with Gryspeerdt et al. (2017), who detected in these areas mostly orographic cirrus clouds. Figure 3a also shows higher ICNCs around the edge of the Antarctic ice sheet and over those regions which experience a strong convective activity, i.e. the Inter Tropical Convergence Zone (ITCZ) and the Tropical Warm Pool (TWP), as observed in Sourdeval et al. (2018)."

"At 600 hPa, ICNCs increase towards high latitudes, in particular over Greenland (up to 2000 L-1) and Antarctica (mostly > 2000 L-1) (Figure 3e). It must be said that, due to the very low temperatures in the the latter region, even at 600 hPa the conditions are typical of the cirrus regime, and the high ICNCs can be related to the high values of both  $w_{sub}$  and ice supersaturation. Gryspeerdt et al. (2017) found that cirrus clouds over Antarctica have primarily synoptic origin. However, differently from Figure 3e, observations do not present such a high peak of ICNC over Antarctica (Gryspeerdt et al. 2018; Sourdeval et al., 2018)."

**P15L1-2:** These high values of SCRE and LCRE in the default simulation are surprising. As can be seen from your Table 2, the observed values of both, SCRE and LCRE are lower. The default simulation needs to be retuned to better match the observed values. Is this simulation in radiative balance at the top of the atmosphere? If not the comparison of CRE of the different simulations to observations is not very meaningful. Add the net radiative balance at the top of the atmosphere to Table 2.

Actually, the default simulation is not in radiative balance at TOA (the imbalance is about 4.5 W/m2). Nevertheless, we preferred to keep the physical parameters constant in all simulations, so to show all the differences arising when using the BN09 algorithm, also from the radiative point of view. In addition, the simulations using BN09 in the cirrus regime are well balanced, and in line with the work of Roeckner et al. (2004, Table 1). We agree that the comparison of CRE with observations can be less meaningful, but we preferred to show it for completeness.

As suggested, we added in Table 2 the radiative fluxes of SW, LW, and the net imbalance at TOA.

**P16L1:** Are some of the quantities in Table 2 tuned to agree with observed values (Mauritsen et al. 2012, Hourdin et al, 2016)?

None of the quantities in Table 2 has been tuned.

**P16L5:** It is mentioned previously in the text that homogeneous nucleation dominates in the tropics and in the SH, whereas heterogeneous nucleation is important in the NH. Would it be possible to split the observations and the analysis in this section into the tropics and the NH extratropics?

Unfortunately, we cannot split the observational data set into tropics and midlatitudes because this analysis is a work in progress by Krämer et al. (paper in preparation).

**P17L3-4:** What is the reason for the better performance of BN09 compared to KL02 at low temperatures? I would assume that both schemes compute homogeneous nucleation at these low temperatures and that the vertical velocities are similar.**

Unfortunately, Figure 5 is affected by an error made during the post-processing and has been replaced by Figure 3 of this document. The results in the mixed-phase regime remain basically unchanged (right plot). In the cirrus regime (left plot), the simulations KL+LD and KL+BN undergo big differences at temperatures below 225 K, and the strong underestimation at very cold temperatures is not evident anymore. The simulations BN+LD and BN+BN show only slight changes which make them a bit closer to the observations (in the intervals 185-190 K and 202-226 K). We are sorry for the mistake. Now, at very cold temperatures ICNCs simulated using BN09 in the cirrus regime are lower than the ICNCs computed by KL+LD and KL+BN, as expected.

The text in Section 4.2 has been modified accordingly to the new Figure. Moreover, we mentioned some comparisons with other modeling studies as the Referee #2 suggested.

"Again, the simulations can be grouped in two sets according to the ice nucleation scheme used in the cirrus regime, i.e. KL+LD/KL+BN and BN+LD/BN+BN, because of their similarities. For most of the temperature range, the simulations which use KL02 in the cirrus regime overestimate the observed ICNCs (although they mostly remain below the 75th percentile). The overestimation of ICNCs is common to other modeling studies (e.g. Wang and Penner, 2010, Liu et al., 2012, and Shi et al., 2015) and especially in cold cirrus clouds (for T < 205 K). On the other hand, the simulations which use BN09 in the cirrus regime are very close to the observations at temperatures below 200 K and between 220 K and 230 K, while they underestimate ICNCs between 200 K and 220 K. In this temperature range the simulations can exceed the observed 25th percentile (although remaining within the 5th percentile). In comparison with the other two simulations, BN+LD and BN+BN always predict lower ICNCs at temperatures below 230 K, as expected because of the competition and PREICE effects. Finally, all four simulations overestimates ICNCs by one order of magnitude in the temperature range 230 – 240 K.

Overall, the simulations BN+LD and BN+BN agree particularly well with the measurements at temperatures lower than 200 K but underestimate the ICNCs within the interval 200 - 220 K, due to an overestimation of the competitive nucleation and PREICE effects. Barahona et al. (2010) showed that the competitive nucleation effect is small using P13. Also, Liu et al. (2012) found that BN09 (using the parameterization of Phillips et al., 2008 for heterogeneous nucleation) and BNhom produced very similar results in the cirrus regime, suggesting that the competive nucleation effect was small because of the low ICNCs formed heterogeneously. Thus, we can deduce that the PREICE effect is the one which is likely overestimated in our simulations. Interestingly, modeled ICNCs do not show any particular trend, like also Kuebbeler et al. (2014) who used ECHAM-HAM. Differently, other studies found that ICNCs are inversely proportional with temperature, e.g. Liu et al. (2012) and Shi et al. (2015) with CAM5, indifferently if they used the ice nucleation scheme of Liu and Penner (2005) or BN09, and Barahona et al. (2010) with GEOS-5 and BN09. Such distinct behaviours are likely derived from the wide model variability in reproducing subgrid-scale processes, like vertical velocity, which play a role in ice nucleation. We reiterate that ICNC is highly dependent on the vertical velocity which is usually poorly represented in terms of spatial and temporal variability (Barahona et al., 2017). "

The lines 9-11, P1 (Abstract) changed to: "Overall, ICNCs agree well with the observations, especially in cold cirrus clouds (at temperatures below 205 K), although they are underestimated between 200 K and 220 K. As BN09 takes into account processes which were previously neglected by the standard version of the model, it is recommended for future EMAC simulations."

The lines 2-4, P19 (Conclusions) changed to: "Overall, all modeled results agree well with global observations and the literature data. The comparison made with flight measurements has pointed out that ICNCs are overestimated by KL02 in the cirrus regime. BN09 agrees well with the observations in cold cirrus clouds, however, the PREICE effect is likely overestimated causing the underestimation of ICNCs between 200 K and 220 K."

**P17L25-26:** In Fig. 2b for example an increase in ICNC in the mixed-phase regime is shown when using BN09 in the cirrus regime. How does this agree with the similarity of the ICNCs of the different simulations in the mixed-phase regime compared to the aircraft measurements?**

It must be said that the "similarities" in the mixed-phase regime shown in Figure 5 (right) can be actually equal to absolute differences of 200 1/L for temperatures below 250 K (please note the log scale on the vertical axis). In fact, if we consider, for example, the ICNC values at T=238 K in Figure 3-right of this document, we can observe that the simulations BN+LD (green) and BN+BN (red) are higher than DEF (blue) by almost 200 1/L. ICNCs shown in Figure 3-right are nothing else that the mean computed along the latitudes of the ICNCs shown in Figure 4-left in this document (to be precise, Figure 3 actually shows the medians, which are a bit smaller than the means). The absolute differences between DEF and BN+KL are shown in Figure 4-center. If we average the differences along the latitude (Figure 4-right), we find that the differences at T=238 K are about 150 1/L. Thus, the differences shown in Figure 2 (of the manuscript, left column) are strongly smoothed by averaging along the latitudinal dimension, reducing the differences in the mixed-phase regime of Figure 3-right.

The reason why the simulations are different from the observations is explained at lines 28-29, P17, why the two data sets of observations are different is explained in the next point.

**P17L26-28:** Give references for WISP-94 and ICE-L.

Why are these two datasets so different (the 25th to 75th percentile do not overlap)?

We added the references of both projects.

For the project WISP-94 an optical array probe for airborne measurements was used, while the data of project ICE-L come from the Continuous Flow Diffusion Chamber (more information can be found in the "Supporting information" of DeMott et al. 2010). Thus, the differences between the two data sets are due to the employment of different instruments and to the ice particle shattering which affects the probe measurements (producing a positive bias).

**P19L5-7:** I agree with your point (1) and (3) but the general better performance of BN09 compared to the default parameterizations is not conclusively shown. While BN09 performs better at T < 205K, there will be fewer and optically thinner clouds at these low temperatures than at the temperature range 205-222K where BN09 agrees less well with aircraft observed in-cloud ICNC than the default parameterizations. In my opinion the additional processes computed by BN09 outweigh this drawback and BN09 should be used in future EMAC simulations but a generally better performance cannot be asserted.

The Referee is right. A better performance of BN09 could not and cannot (with the new Figure 3) be established. We modified lines 5-9, P19 as follows.

"As BN09 takes into account additional processes which were previously neglected by the standard version of the model, without consuming extra computational resources, we recommend to apply this ice nucleation scheme in future EMAC simulations. We also suggest to select P13 among the INP parameterizations available in BN09, since it incorporates the ice-nucleating ability of different aerosol species (dust, soot, bioaerosols, and soluble organics) and simulates both deposition and immersion/condensation nucleation."

**P19L31-P20L2:** As this is interstitial aerosol, at what relative humidity is the wet diameter of the sulfate aerosol in the Aitken mode computed?

As mentioned before (see point P8L17), we used the dry diameter of the Aitken soluble mode.

The wet diameter is calculated by the aerosol model GMXe based on the relative humidity computed online by the model.

**TECHNICAL CORRECTIONS**

**P1L5:** Only one of the multiple ice nucleating particle spectra is applied in one simulation. Rephrase the sentence as it reads now as if the multiple ice nucleating particle spectra are applied simultaneously.

Also INP spectra should be replaced by INP parameterization.

We changed the sentence: "Furthermore, the influence of chemically-heterogeneous, polydisperse aerosols is considered by applying one of the multiple ice nucleating particle parameterizations which are included in BN09 to compute the heterogeneously formed ice crystals."

We changed the words "spectrum" and "spectra" with "parameterization(s)" in the whole manuscript.

**P2L7:** The greenhouse effects dominates for cirrus clouds e.g. Chen et al. (2000).

We added such information: "... they scatter solar radiation back into the space (albedo effect) and absorb and re-emit longwave terrestrial radiation (greenhouse effect). Differently from other types of clouds, cirrus clouds produce a net warming at the top of the atmosphere (TOA) (e.g. Chen et al. 2000, Hong et al., 2016, Matus and L'Ecuyer, 2017)."

**P2L9:** Mixed-phase clouds can also occur at colder temperatures, rephrase.

We used the temperature threshold  $-35^{\circ}$ C (e.g. Lohmann et al. 2009), expressed in Kelvin as requested by the Referee#2.

We changed the sentence to: "In addition, mixed-phase clouds consist of both supercooled liquid cloud droplets and ice crystals and appear at subfreezing temperatures above 238 K." **P2L10-11:** This is only true when deep convective clouds are included in the term mixed-phase clouds, but deep convective clouds are often named separately. As you here include deep convective clouds, this should be mentioned explicitly.

We preferred to remove such sentence as it was outside the issues treated in this paper. Rather, we included some short information about precipitation and cloud electrification here and about secondary ice production at line 31, P2.

"As ice crystals can grow quickly to precipitation-sized particles, precipitation is mainly formed in mixed-phase clouds, while precipitation from cirrus clouds does not usually reach the surface (Lohmann 2017). The mixed phase is also important for cloud electrification and intracloud lightning, which occur through the in-cloud charge separation via a transition from supercooled raindrops to graupel over the mixed-phase temperature range (Korolev et al. 2017)."

"The cirrus regime ... The mixed-phase regime ... In the latter regime, besides primary nucleation, another mechanism which controls ICNCs is the secondary ice production, i.e. the production of new ice crystals via the multiplication of preexisting ice particles without the action of INPs."

**P2L11-12:** Provide here references such as McCoy et al. (2016).

Done.

**P2L25:** Explain what you mean here by "the overestimation of vertical velocity".

We added some new lines to explain it better.

"Based on modeling studies, homogeneous nucleation has been considered the dominant process for cirrus formation (e.g. Haag et al., 2003; Gettelman et al., 2012) because the concentration of liquid droplets is higher than that of INPs in the upper troposphere. However, some field measurements found a predominance of heterogeneous nucleation and lower ice crystal number concentrations (ICNCs) than produced by homogeneous nucleation (e.g. Cziczo et al., 2013; Jensen et al., 2013). What process is dominant is still under debate, although recent studies suggested the overestimation of the vertical velocity as possible cause of the discrepancy between modeled results and observations (e.g. Barahona and Nenes, 2011; Zhou et al., 2016; Barahona et al., 2017)."

**P2L35-P3L1:** This sentence is true for mixed-phase clouds while the sentences before and afterwards concern cirrus clouds. This is confusing, rephrase or move this sentence.**

The PREICE effect concerns both the mixed-phase and the cirrus regimes and does not include the "condensation onto pre-existing cloud droplets". We removed that part of the sentence and we reformulated lines 32-35, P2 and 1-2, P3 as follows.

"This competition between homogeneous and heterogeneous nucleation for water vapour drastically affects the ICNC in the cirrus regime, even at low INP concentrations (Kärcher and Lohmann, 2003; Spichtinger and Cziczo, 2010). On the other hand, both in the cirrus regime and in the mixed-phase regime, water vapour can also be reduced by depositional growth onto pre-existing ice crystals and ice crystals carried into the cloud via convective detrainment and advective transport, thus, inhibiting ice nucleation. The impact of pre-existing ice crystals (PREICE) can be especially important in cirrus clouds, when..."

**P3L12:** Do you mean numerical parcel model simulations?**

Yes, we corrected the expression.

**P6L21-22: Is this reduction done only for cirrus clouds are also for mixed-phase clouds?**

It is actually done only for cirrus clouds. We modified the sentence to: "The only expedient adopted by the CLOUD submodel is to reduce the number of aerosol particles available for ice nucleation by the existing ice particle number in the cirrus regime."

**P6L33:** Is the cloud parcel mentioned here explicitly computed in EMAC or do the equations (1-6) provide analytical solutions for the cloud parcel?

We used here the expression "cloud parcel" just to explain what happens when INPs overcome a certain threshold. There are no explicit computations of cloud parcels. To avoid misunderstandings, we simply deleted the part "that develops in the cloud parcel".

**P9L10-14:** Provide references for the anthropogenic aerosol emissions and describe how natural aerosols (e.g. dust) are treated.

We improved the description of aerosol emissions between lines 9-13, P9 as written previously at point P4.

**P8L6:** Is  $n_x$  the number of interstitial aerosol particles or is the number of aerosol particles in cloud droplets tracked? Please clarify.

According to Phillips et al. 2008 and 2013,  $n_x$  is the number of aerosol particles "including interstitial IN and IN immersed in cloud liquid". However, with the implementation of BN09 in EMAC, P13 uses only interstitial aerosols. We clarified this in Sections 2.3.1 and 2.3.2:

"...,  $n_x$  is the number concentration of aerosol particles (interstitial and INP immersed in cloud droplets) of species X, ..." at line 7, P8.

"... of interstitial aerosol of species X (which can be ..." at line 19, P8.

Indeed, in immersion/condensation ice nucleation parameterizations it is usually assumed that each INP corresponds to exactly one cloud droplet which freezes when the INP reaches its characteristic freezing temperature, as discussed in Paukert et al. 2017.

**P10L26-P11L1:** Do you mean that IWC decreases where ICNC decreases?

Yes, we meant that, but actually the sentence is unclear.

We rephrased it: "The relative changes in Figure 2f show a pattern very similar to Figure 2b, therefore, IWC decreases where ICNC reduces (and vice versa) when BN09 is used in the cirrus regime."

**P11L32-P13L2:** The increase in IWC in equatorial regions at 200 hPa is about 5-10% (Fig. 2), I would not call this dramatic.

We deleted this part, as IWC does not increase "dramatically" but increases where also ICNCs increase.

**P13L21-23:** It should be mentioned that observations of cloud droplet number concentration are uncertain (Bennartz and Rausch, 2017).

Done. Now, the sentence is:

"Vertically integrated cloud droplet number concentration  $(CDNC_{burden})$  is not influenced by the choice of the ice nucleation scheme. Its values are comparable with previous modeling studies (e.g. Lohmann et al., 2007; Hoose et al., 2008; Salzmann et al., 2010; Wang and Penner, 2010; Kuebbeler et al., 2014; Shi et al., 2015) and observations, although satellite observations are still affected by strong uncertainties (Bennartz and Rausch, 2017)."

**P13L27-29:** The underestimation of IWP was found in previous studies using ECHAM-HAM. The IWP in ECHAM is not underestimated see e.g. Mauritsen et al. (2012).

Thanks, we changed "ECHAM" with "ECHAM-HAM".

P17L28-30: Please add the number of hours in mixed-phase clouds.

We added this information: "The modeled ICNCs are in rather good agreement with two data sets of flight measurements taken from the projects Winter Icing Storms Project (WISP-94) and Ice in Clouds Experiment-Layer Clouds (ICE-L), which consider about 99 and 46 flight hours, respectively."

**P17L30:** When the measurements are for INP this needs to reflected in Fig. 5 itself (at least in the figure caption).

We added such information in the caption of Figure 5.

**P18L5:** Use INP parameterization instead of INP spectrum.

We changed the words "spectrum" and "spectra" with "parameterization(s)" in the whole manuscript.

Caption Fig. S4: Give references for the observational datasets.

Done.

Fig. S5 is a table not a figure.

Thanks, we corrected it.

**References**

- Aquila, V., et al.: MADE-in: a new aerosol microphysics submodel for global simulation of insoluble particles and their mixing state, Geosci. Model Dev., 4, 325–355, 2011.
- DeMott, P.J., Prenni, A.J., Liu, X., Kreidenweis, S.M., Petters, M.D., Twohy, C.H., Richardson, M.S., Eidhammer, T., and Rogers, D.C.: Predicting global atmospheric ice nuclei distributions and their impacts on climate, PNAS, 107, 25, 11217-11222, 2010.
- Fountoukis, C., and Nenes, A.: ISORROPIA II: a computationally efficient thermodynamic equilibrium model for K+-Ca2+-Mg2+-NH44-Na+-SO24-NO-3-Cl--H2O aerosols, Atmos. Chem. Phys., 7, 4639–4659,2007.
- Hong, Y., Liu, G., and Li, J.-L. F.: Assessing the Radiative Effects of Global Ice Clouds Based on CloudSat and CALIPSO Measurements, Journal of Climate, 29, 7651-7674, 2016.
- Karydis, V.A., Kumar, P., Barahona, D., Sokolik, I.N., and Nenes, A.: On the effect of dust particles on global cloud condensation nuclei and cloud droplet number, J. Geophys. Res.-Atmos., 116, 2011.
- Kumar, P., Sokolik, I.N., and Nenes, A.: Parameterization of cloud droplet formation for global and regional models: including adsorption activation from insoluble CCN, Atmos. Chem. Phys., 9, 251–2532, 2009.
- Kumar, P., Sokolik, I. N., and Nenes, A.: Cloud condensation nuclei activity and droplet activation kinetics of wet processed regional dust samples and minerals, Atmos. Chem. Phys., 11, 8661–8676, 2011.

- Lewis, E.R.: An examination of Köhler theory resulting in an accurate expression for the equilibrium radius ratio of a hygroscopic aerosol particle valid up to and including relative humidity 100%, J. Geophysical Res., 113, D03205, 2008.
- Lohmann, U., and Hoose, C.:Sensitivity studies of different aerosol indirect effects in mixed-phase clouds, Atmos. Chem. Phys., 9, 8917-8934, 2009.
- Matus, A.V., and L'Ecuyer, T.S.: The role of cloud phase in Earth's radiation budget, Geophys. Res. Atmos., 122, 2559-2578, 2017.
- Paukert, M., Hoose, C., and Simmel, M.: Redistribution of ice nuclei between cloud and rain droplets: Parameterization and application to deep convective clouds, J. Adv. Model. Earth Syst., 9, 514-535, 2017.
- Petters, M. D., and Kreidenweis, S. M.: A single parameter representation of hygroscopic growth and cloud condensation nucleus activity, Atmos. Chem. Phys., 7, 1961–1971, 2007.
- Roeckner, E., Brokopf, R., Esch, M., Giorgetta, M., Hagemann, S., Kornblueh, L., Manzini, E., Schlese, U., and Schulzweida, U.: The atmospheric general circulation model ECHAM5, Part II, Report No. 354, Max-Planck-Institut für Meteorologie, 2004.
- Tanre, D., Geleyn, J.-F., and Slingo, J.M.: First results of the introduction of an advanced aerosol-radiation interaction in the ECMWF low resolution global model, in: Aerosols and their climatic effects, edited by: Gerber, H. and Deepak, A., A. Deepak Pub., 133-177, 1984.
- Tost, H., Jöckel, P., and Lelieveld, J.: Influence of different convection parameterisations in a GCM, Atmos. Chem. Phys., 6, 5475–5493, 2006.

Figure 1: Topography (m) of T42 horizontal resolution derived from surface geopotential.

---

## Author Comment (AC2) · 5 Jun 2018

We really thank the anonymous Referee for the constructive and fruitful comments. Below we report our replies to the "Specific comments".

**Title**

Title sounds too technical: is there really a need to add "(based on MESSy 2.53)". As a suggestion you could further simplify the title to something like: Implementation of a new ice phase/ice cloud parameterization in the EMAC model.

According to the GMD rules of "Manuscript composition" for the authors, the title has to be "concise but informative, including model name and version number if a model description paper". Therefore, "(based on MESSy 2.53)" in the title is a journal requirement and we cannot change it.

**Abstract**

Needs to be rearranged, right now is in my opinion a bit out of a logical order. I suggest: 1.) mention that you implemented BN09 for both cirrus and mixed phase clouds 2.) Only now go in details of homog. vs heterog. nucleation, aerosols, etc.

If possible, we would prefer to leave the order as it is. Our logic is:

1) Saying in one sentence what is the manuscript about (the first two lines in the Abstract).

2) Describing the parameterization in order to inform the reader about the capabilities of the "tool" that we will employ.

3) Answering the question: How do we use it? We can use it in both regimes.

4) Major results.

Some minor comments:

**line 2: realistically represent => quite a bold statement**

As BN09 takes into account processes which were previously neglected by EMAC, we changed the sentence to:

"A comprehensive ice nucleation parameterization has been implemented in the global chemistry-climate model EMAC to improve the representation of ice crystal number concentrations (ICNCs)."

line 4: cold clouds = never defined it

We wrote "cirrus clouds" instead of "cold clouds".

Moreover, since the PREICE effect was actually added in this work (it was not considered by the original BN09 algorithm), this sentence was changed to:

"The parameterization of Barahona and Nenes (2009, hereafter BN09) allows the treatment of ice nucleation taking into account the competition for water vapour between homogeneous and heterogeneous nucleation in cirrus clouds."

Rather, we mentioned the PREICE effect at line 7:

"BN09 has been modified in order to consider the pre-existing ice crystal effect and implemented to operate both in the cirrus and in the mixed-phase regimes."

lines 4-6: the sentence starting with "Furthermore" is hard to understand. Please rewrite!

This sentence was changed as follows (see also Referee #1, P1L5):

"Furthermore, the influence of chemically-heterogeneous, polydisperse aerosols is considered by applying one of the multiple ice nucleating particle parameterizations which are included in BN09 to compute the heterogeneously formed ice crystals." line 7: Compared to the standard EMAC "results"... => ...

We changed the word "results" with "parameterizations" (they are KL02 and LD06).

line 10: ...improves the model results... => too vague, be more concrete, which results?

Such sentence was modified as written in our reply to Referee #1, P17L3-4: "Overall, ICNCs agree well with the observations, especially in cold cirrus clouds (at temperatures below 205 K), although they are underestimated between 200 K and 220 K. As BN09 takes into account processes which were previously neglected by the standard version of the model, it is recommended for future EMAC simulations."

**Introduction**

**1st paragraph** sounds like ice nucleation and droplet activation are the only two challenging processes in the representation of clouds. Is this true?**

At line 15 it is written that the representation of clouds is one of the major challenges in climate studies, and this holds generally for many processes which occur in clouds (e.g. cloud phase transitions, INP characteristics influencing ice nucleation, secondary ice production mechanisms, aerosol-clouds interactions). Then, we just mentioned the liquid droplet activation and we focused on the ice crystal formation because it is the main process considered in this work. For clarity, we changed the sentence to:

"Nevertheless, clouds remain one of the less understood components of the atmospheric system, and their representation in models (including processes like cloud droplet formation, ice nucleation, cloud phase transitions, secondary ice production, aerosol-cloud interactions) is one of the major challenges in climate studies (IPCC, 2013; Seinfeld et al., 2016)."

**P2L15:** I think the word elusive isn't used in a correct way here.

We changed "most elusive" with "less understood".

**P2L4-13:** Wouldn't it be more logic to start with mixed phase clouds? After all, they have a larger radiative impact on climate and (regionally) on cloud feedbacks than cirrus.

Through the whole manuscript we firstly wrote about the cirrus regime and then the mixed-phase regime, e.g. in Sections 2.3.1, 3.2, 4.2 (with Figures 3, 4, 5). We simply decided to follow the order "from high altitudes to low altitudes" (or "from low temperatures to warmer temperatures") like the CLOUD submodel does (see Figure 1). For consistency with the rest of the manuscript, we would prefer to keep this order also in the Introduction.

P2L4-13: you never defined what a "mixed-phase cloud" is.

Mixed-phase clouds are defined at lines 7-9, P2 (where we changed the temperature threshold to 238 K).

**P2L4:** typically below -35° => why typically? depends on your definition, as there is no global definition of what a cirrus cloud is. So if you in your manuscript go for the -35° threshold just say that firmly.

+ here you use Celsius, while throughout the whole text Kelvin. Be consistent. (I personally don't see any advantage of using Kelvin over Celsius, but that's purely a matter of personal taste)

Thanks for pointing this out. We removed the word "typically".

We used Kelvin because it is the unit of the International System and it is used by the model. To be consistent through the whole manuscript we changed the temperature values from Celsius to Kelvin in the Introduction: line 4 and line 8, P1: 238 K.

P2L6: missing references on the radiative role of cirrus - maybe Matus and L'Ecuyer 2017, Hong et al. 2016 for some recent satellite estimate of their radiative effects or even Kienast-Sjogren et al. 2016 for lidar-based estimates, Gasparini and Lohmann 2016 for GCM modelling-based estimates. The first two references could be cited in the context of mixed-phase CRE too.

We cited Matus and L'Ecuyer 2017, Hong et al. 2016 for the cirrus warming effect. For the mixed-phase clouds, we added the sentence:

"Mixed-phase clouds generates a net cooling at TOA, although the estimates of their radiative effects are complicated by the coexistence of both ice and liquid cloud phases (Matus and L'Ecuyer, 2017)."

**P2L10:** missing some references on mixed-phase being TD unstable, or similar.

We added the following citations: "(e.g. Korolev et al., 2007; Korolev et al., 2017)."

**P2L11-13:** The section on mixed-phase is in a poor state. It deserves at least 1-2 sentences more, giving reference for the listed processes/facts (e.g. that mixed-phase, if we define it just by temperature threshold, is probably responsible for most/a large share of precipitation, which is different from the cloud top phase classification of Mulmenstadt et al. 2015).

Are mixed phase responsible for lightning and storms?

Aren't convective clouds treated by a different scheme in your model?

What do you mean by "strong storms". That's all written in a to ambiguous way for a scientific paper.

We added the following information regarding mixed-phase clouds (see also Referee#1, P2L10-11).

"As ice crystals can grow quickly to precipitation-sized particles, precipitation is mainly formed in mixed-phase clouds, while precipitation from cirrus clouds does not usually reach the surface (Lohmann 2017). The mixed phase is also important for cloud electrification and intracloud lightning, which occur through the in-cloud charge separation via a transition from supercooled raindrops to graupel over the mixed-phase temperature range (Korolev et al. 2017)."

Convective clouds are treated separately by the CONVECT submodel, but this information is given in Section 2.1.

As written in the reply to Referee #1 (P2L10-11), we removed the sentence mentioning "strong storm" in favour of the information written before.

**P2L12:** there's tons of references on that, why didn't the authors include any? (e.g. Tan et al., 2016, Science, studies looking more specifically into the Southern Ocean like Vergara-Temprado et al, 2018, PNAS and many more)

We added the citations: McCoy et al., 2016; Tan et al., 2016; Vergara-Temprado et al., 2018.

**P2L23-26:** I don't think the upper 2 statements are totally correct, possibly due to a too condensed information. Early observational studies of cirrus clouds were affected by the problem of ice crystal shattering, which implied several times too large ice crystal number

concentrations. Such numbers were hard to explain other than with homogeneous nucleation, and were also replicated by model studies.

Moreover, we have also numerous modelling studies (ok, Barahona et al. being one of them) showing that heterogenous nucleation might play a role in cirrus, for example: Sullivan et al., 2016, Storelvmo and Herger 2014, Penner et al. 2015, Gasparini and Lohmann 2016.

I don't think there is a universal agreement on the overestimation of vertical velocities by GCMs. A study by Joos et al. 2008 and Kärcher and Ström 2003 show a good agreement between vertical velocity observations and model updrafts. The updrafts were based on the large scale updraft and a TKE based term, which was in Joos et al. 2008 over mountains replaced by gravity waves. As Joos et al. use the same (I guess) dynamical core than the described model, we can imagine that the TKE based updrafts could be in line with observations.

And Cziczo et al., 2013 also isn't talking about updraft overestimation, despite being cited for it.

We extended these lines to better explain this issue (see also Referee #1, P2L25).

"Based on modeling studies, homogeneous nucleation has been considered the dominant process for cirrus formation (e.g. Haag et al., 2003; Gettelman et al., 2012) because the concentration of liquid droplets is higher than that of INPs in the upper troposphere. However, some field measurements found a predominance of heterogeneous nucleation and lower ice crystal number concentrations (ICNCs) than produced by homogeneous nucleation (e.g. Cziczo et al., 2013; Jensen et al., 2013). What process is dominant is still under debate, although recent studies suggested the overestimation of the vertical velocity as possible cause of the discrepancy between modeled results and observations (e.g. Barahona and Nenes, 2011; Zhou et al., 2016; Barahona et al., 2017)."

We did not cite Kärcher and Ström (2003) and Joos et al. (2008) because the fact that simulated vertical velocity (given by the sum of large-scale vertical velocity and subgrid-scale TKE component) is in good agreement with observations does not mean that it is not underestimated (Fig. 9 in Kärcher and Ström (2003) shows that the modeled vertical velocity is underestimated, although it has definitely improved with respect to the previous representation, i.e. large-scale vertical velocity). We thought it was more appropriate to cite these two references in Section 2.2 (line 18, P6):

"Other studies, e.g. Kärcher and Ström (2003) and Joos et al. (2008), showed that w is in good agreement with vertical velocity observations."

**P2L26-30:** The following (very long) sentence should appear earlier in text as it defines the two ice crystal formation regimes.**

We split the sentence into smaller sentences and we also exchanged their order (because we have always firstly written about cirrus clouds and then about mixed-phase clouds):

"Overall, two different regimes for ice crystal formation are distinguished. The cirrus regime at cold temperatures (T < 238 K), where ice crystals originate via heterogeneous and homogeneous nucleation to form cirrus clouds. The mixed-phase regime at subfreezing temperatures between 238 K and 273 K, where ice crystals form exclusively via heterogeneous nucleation and alter the phase composition of the mixed-phase clouds."

However, we preferred to leave these lines in the same position because they mention the ice nucleation mechanisms (homogeneous and heterogeneous) which are described before.

**P2L31-35:** I am missing a description of freezing in mixed-phase clouds? Why do you always refer only to cirrus, if you implemented freezing also at mixed-phase conditions?

Ice nucleation in mixed-phase clouds is described at lines 18-23, P2. At the end of page 2 and beginning of page 3, we describe the competition for water vapour between homogeneous and heterogeneous nucleation (in the cirrus regime) and the PREICE effect, i.e. the two processes which will be considered by BN09. This part has been slightly changed after the request by the Referee#1 (please, see point P2L35-P3L1).

**P3L10:** Can you find some evidence/reference for the following sentence: "Including sophisticated schemes in general circulation models (GCMs) allows for a more realistic description of the variability of cloud properties and cloud radiative effects, improving the model climate predictions"

We cited: Lohmann and Feichter (2005) and Barahona et al. (2014).

**P3L18:** please explain what INP spectra mean. I assume that's simply a parameterization of het ice nucleation?

Exactly. We replaced the words "spectrum" and "spectra" with "parameterization(s)" (as requested by the Referee #1).

**P3L29:** What kind of scheme does your standard version of the model use?

The standard configuration of EMAC and its schemes are described in Sections 2.1. and 2.2.

**Model description and set-up of simulations**

P4: Convection plays a large role in global high cloud distributions and their properties. You should include some more information on how the CONVECT submodel interacts with the microphysics and cloud cover. I add some questions which could be addressed: - How does the convective detrainment works?

- How do you compute/parameterize the size of ice crystals that are detrained from convective clouds?

- How does the scheme decide whether you detrain liquid or water (or even vapour)?

We added the following information (see also Referee #1, P4):

"The CONVECT submodel contains multiple convection parameterizations (Tost et al., 2006). In this work the scheme of Tiedtke (1989) has been used. Convective cloud microphysics is highly simplified and neither explicit aerosol activation into liquid droplets nor aerosol effects in the ice formation processes are taken into account, i.e. convective microphysics is solely based on temperature and updraft strength. Detrainment from convection is treated by taking updraft (and downdraft) concentrations of water vapour and cloud condensate and the corresponding massflux detrainment rates into account. These are merged including turbulent detrainment (i.e. exchange of mass through the cloud edges) and organised detrainment (i.e. organized outflow at cloud top). The detrained water vapour is added to the large-scale water vapour field, while the detrained cloud condensate is directly used as a source term for cloud condensate by the large-scale cloud scheme (i.e. the CLOUD submodel), which considers the detrained condensate either liquid or ice depending on the temperature (if T < 238 K the phase is ice) and the updraft velocity. The size and numbers of the detrained condensate are not taken into account explicitly."

**P4L25:** You previously defined ice crystal number concentration as ICNC. Here, you define it again as Ni. Please, be consistent!

We replaced " $N_i$ " with "ICNC". Moreover, we replaced " $N_l$ " with "CDNC" and we removed " $q_w$ ", " $q_i$ ", " $q_l$ " as they are never used later in the text.

**P6L20-21:** This cannot be a separate paragraph. What do you mean with "the only precaution"?

> We merged these lines to the previous paragraph. We changed "precaution" with "expedient". We mean that this is the unique, approximate way of the CLOUD submodel to take into account the pre-existing ice crystals.

**P6L23-25:** The text between points 2.3 and 2.3.1 is repeating the information already given before. Please remove it.

Done.

P6L26-29: You already provided the same information on page 3. Please try to avoid repetition!

We deleted lines 26-27, P3 but we left lines 16-18, P3 because we consider this the minimum information to introduce BN09. Here, line 26, P6 gives new information, while the next sentence was slightly changed:

"It explicitly considers the competition for water vapour between homogeneous and heterogeneous nucleation in the cirrus regime, the influence of chemically-heterogeneous, polydisperse aerosols acting as INPs, and allows to use different heterogeneous nucleation parameterizations."

**P7L26:** Not sure that you can assume that P13 agrees better with observations in every model (thinking that vertical velocities might be different than in CAM)

Naturally, we cannot assume that P13 agrees better with observations in every model, but the sensitivity studies suggest that generally P13 performs better than the other INP parameterizations listed at lines 19-20, P7. Barahona et al. (2010), who compared various INP parameterizations available in BN09 using another global model, the Global Modeling Initiative (GMI), found that PDA08 (which is the previous version of P13) better agrees with observations.

**Implementation**

**P8L21:** Did you define what "M modes" are?

We defined "M" at line 21: M can be K (Aitken), A (accumulation) or C (coarse). We modified the sentence to make it clearer:

"...the diameters  $D_M$  are not distinguished among aerosol species but only among the modes (Aitken (K), accumulation (A), coarse (C), i.e. M = K, A, C) which the species belong to."

P8L24-25: Could you describe that a bit better as it is not a standard procedure in GCMs?

This procedure is actually performed by the BN09 parameterization. We explained it better and we moved these lines to the end of Section 2.3.1.

"In order to account for sub-grid variabilities, the output variables of BN09 which depend on the vertical velocity (f(w)) are weighted over a Gaussian updraft velocity

distribution by numerically calculating the integral (Morales and Nenes, 2010; Sullivan et al. 2016):

$$\overline{f(w)} = \frac{\int_0^\infty f(w')P(w')dw'}{\int_0^\infty P(w')dw'}$$
(1)

where P(w') is the Gaussian probability density function of sub-grid vertical velocities (w') with mean 0.1 cm s-1 and standard deviation equal to  $w_{sub}$ ."

**P9L3-4:** I think that doesn't fit in the model description part of the paper but in the results.

We moved this sentence in Section 3 (before Section 3.1), and we added the new sentence: "In this Section we investigate the changes and the effects obtained by using BN09 in the different regimes."

**Model results**

**P10L3-4:** Not sure about that. I think your Figure 3a makes me think it is not INPs but mountains that contribute most to the larger ICNC in the northern hemisphere.

Looking at the maps (Figures 3 and S2) and according to the literature cited at line 11, indeed both factors (big mountain chains and more INPs in the NH) contribute to higher ICNCs in the NH. We rephrased the sentence:

"..., while they are much higher over the mid-latitudes in the Northern Hemisphere (NH) because of larger INP concentrations and the influence of big mountain chains, e.g. Rocky Mountains and the Himalayas (Figure 2a)."

Figures: Please indicate which areas are significantly different from the "DEF" case in Figures 2 and 3 by applying an appropriate statistical significance test! Same for plots S1, S2, S3.

Add +/-1 or 2 st. deviation shading to the lines plotted in S4.

You could also tentatively try to plot the 25th and 75th percentile range in Figure 5, maybe only for 1 setup due to clarity (BN+BN, I would suggest).

We estimated the statistical significance using the Welch's t-test and we marked the areas with 95% level of significance in all plots which show relative percentage changes. All figures were modified accordingly.

We plotted the error bars for +/- one standard deviation (only for the simulation BN+BN) in Figure S4.

We plotted 5th-95th and 25th-75th percentiles of BN+BN in the comparison with flight measurements (see Figure 1 of this document).

**Table 2:** Please also add standard deviations to your Table 2 for a better feeling of the magnitude of changes due to changing microphysics!

Done. We attributed to each annual global mean the (temporal and spatial) standard deviation.

Figures: Do you show in-cloud or all-sky ICNC and IWC values on your figures? Mention it somewhere in text!

We used in-cloud ICNCs only in Figure 5 (as specified in the caption). We added the information "(grid-averaged)" in the captions of the other Figures.

**P10L6-7:** Moreover, KL02 simulate only homogeneous nucleation, while BN09 simulate also heterogeneous nucleation at cirrus conditions. Therefore, you should point out somewhere that you are not really making an apples-to-apples comparison.

Actually, here we compared BNhom (not BN09) and KL02, but we explained it better.

"As ice crystals are formed almost exclusively via homogeneous nucleation here (not shown) and BNhom and KL02 produce the same order of magnitude of ICNCs (Barahona and Nenes, 2008), the negative bias is likely due to the PREICE effect predicted by BN09."

**P10L8-9:** How was that done before in the REF case? Did you use only large-scale updraft? Do you consider a Gaussian distribution of vertical velocities (Sullivan et al., 2016) also in mixed-phase conditions?

As answered to Referee #1, the sentence regarding TKE does not explain why there is a positive bias in the mixed-phase regime in the comparison of BN+LD with KL+LD (i.e. Figure 2b), thus, we removed such sentence.

KL02, like BN09, uses  $w_{sub} = 0.7 * \sqrt{TKE}$ , not the large-scale updraft.

BN09 uses a Gaussian distribution of vertical velocities (in the cirrus regime in BN+LD, in the mixed-phase regime in KL+BN, in both regimes in BN+BN).

**P10L10:** First you talk about cirrus, than mixed-phase, now cirrus again, I guess. That's confusing for the reader, which expects this sentence to refer to mixed-phase clouds. Please reorder or clarify better!

We reordered and rephrased lines 6-13 as follows (including also what we replied to Referee#1, P10L7-10).

"As ice crystals are formed almost exclusively via homogeneous nucleation here (not shown) and BNhom and KL02 produce the same order of magnitude of ICNCs (Barahona and Nenes, 2008), the negative bias is likely due to the PREICE effect predicted by BN09. Indeed, it has been demonstrated that homogeneous nucleation dominates in the upper troposphere in the tropics and in the SH (Haag et al., 2003; Liu et al., 2012; Barahona et al., 2017), while heterogeneous nucleation is important in the NH (Cziczo et al., 2009), where cirrus clouds are formed from a combination of homogeneous and heterogeneous processes. Interestingly, ICNCs at lower altitudes are also influenced by the ice nucleation parameterization used in the cirrus regime. In fact, there is an increase of ICNCs in the mixed-phase regime probably due to a faster sedimentation of the larger ice crystal produced by BN09 in cirrus clouds, especially in the NH where there are larger sources of efficient ice-nucleating mineral dust. Overall ..."

**P10L17:** *missing citation(s) at the end of the following sentence.**

This assertion refers to our results. Probably the expression "using the various ice schemes in the mixed-phase regime" is confusing. We modified it: "Overall, the ICNC deviations in the mixed-phase regime obtained using the two different parameterizations are smaller (mostly within  $\pm 20\%$ ) than in the cirrus regime."

**P10L18-20:** please rephrase (cirrus don't occur throughout the year? where, why not?...)

We are sorry, the word "whole" is missing: "Since cirrus clouds do not occur throughout the whole year, ...". **P10:** Ice nucleation in mixed-phase may not be the main source of IWC and ICNC between 0 and -38°C. Could you estimate that from your model and comment on that? Possible processes that might not be negligible are for instance sedimentation of ICs from cirrus or detrainment of IC from convection.

The Referee is right but, unfortunately, we do not have these tendencies (sedimentation and detrainment) stored as output. Thus, we cannot quantify their contributions but we mentioned these sources for the mixed-phase regime in the revised manuscript.

**P10L22:** BN+LD case shows some differences with respect to DEF also in the mixed-phase regime (see Fig 2, Fig 3 f, also fig S1).

- Why is that when the mixed-phase freezing is the same? What other sources of ice exist in mixed phase?

- Can there be some radiative/dynamical/microphysical responses of mixed-phase to difference in cirrus scheme?

- There seems to be a response in convection in the tropics. Is this really the case? What caused it? Did the atmospheric stability change? Please comment!

Previously, in P10L10 we mentioned the sedimentation of larger ice crystals from cirrus clouds as a possible cause for the increment of ICNCs at lower altitudes in BN+LD with respect to DEF. Naturally, changes of cloud phase (ice and supercooled liquid) in mixed-phase clouds influence radiative fluxes and dynamics, because of the "self-maintaining feedback pathway between liquid water, radiation, and turbulence" (Morrison et al., 2011), and there can be a response in convection. Indeed, the net cooling found in Table 2 (NCRE) can decrease the static stability and enhance updrafts, which in turn affects ICNCs. However, we did not investigate these aspects (linked, among the other things, to convective clouds which are treated by an independent submodel) because they are not the focus of this study. This paper wants to describe the implementation of BN09 and analyse the products of BN09, without expanding to other atmospheric effects which will be studied in a future scientific paper (while this remains a paper about model developments).

**P10L23:** I don't understand what you mean with this sentence? If you look at upper troposphere, the opposite is true. While I agree with the following statement for regions between -30 and 0°C.

The sentence was not properly correct. We changed lines 23-25 as follows.

"IWC pattern (Figure 2e) qualitatively follows the ICNC distribution. It is quite symmetrical between the two hemispheres except at high latitudes in the NH, where IWC is slightly higher because of the higher values of ICNC. Particularly, IWC exhibits three local maxima: two over the mid-latitudes in both hemispheres and one in the tropics, associated to storm tracks and deep convections, respectively (Li et al., 2012). These features are in agreement with satellite observations, e.g. Waliser et al. (2009), Li et al. (2012)."

P10L23: I don't understand the connection with the first part of the sentence. I see you have 3 peaks of IWC which come out of your model, which is good, as the observations agree with it (please consult/refer e.g. to: Li et al., 2012). What atmospheric features do the 3 peaks correspond to?

Please, see our previous answer (P10L23).

**Global distribution**

**P11L7:** You never mention why you decided for 200 and 600 hPa levels.

We indicated in parenthesis that the two levels are representative for the cirrus regime and the mixed-phase regime (we wrote this also in the caption of Figures 3 and 4). We specified it better at lines 7-8, P11: "Figure 3 shows the global distributions of ICNC annual means at two different altitudes: 200 hPa (where temperatures vary between 200 K and 220 K) to represent the cirrus regime and 600 hPa (where temperatures are approximately between 240 K and 260 K) to represent the mixedphase regime."

**P11L8:** What do you mean by precipitation patterns? Can you also mention why does this happen, and why ICNC peaks also over mountains.

Does the ICNC global distribution compare well with recent observations by Sourdeval et al., 2018 and Gyrspeerdt et al., 2018?

Thanks for the interesting references. We changed the sentence (also according to the Referee #1's comment P11L8-9) as follows.

"ICNCs in the cirrus regime (Figure 3a) show areas with high values over land and in correspondence with mountainous regions, e.g. the Rocky Mountains, Andes, and Tibetan Plateau with ICNCs > 500 L-1. Such pattern is strongly related to the turbulent contribution of the vertical velocity  $w_{sub}$  and in agreement with Gryspeerdt et al. (2017), who detected in these areas mostly orographic cirrus clouds. Figure 3a also shows higher ICNCs around the edge of the Antarctic ice sheet and over those regions which experience a strong convective activity, i.e. the Inter Tropical Convergence Zone (ITCZ) and the Tropical Warm Pool (TWP), as observed in Sourdeval et al. (2018)."

**P11L11-12:** Again, I would like to see more explanations and not only description of figures. Why are India and Indonesia different from the rest of the world?

The new ice crystals produced BN09 (i.e. the output which is then passed to the CLOUD submodel) are almost exclusively formed via homogeneous nucleation at 200 hPa, therefore the reduction is actually due to the PREICE effect. We rephrased all the lines 11-13, P11.

"The relative changes clearly show that BN09 used in the cirrus regime (Figure 3b, d) reduces ICNC (up to 60%) worldwide with respect to the default experiment, and the ICNC annual global mean drops to 137  $L^{-1}$  (i.e. more than 30%). Such a reduction occurs mostly because of the PREICE effect, being the ice crystals mainly of homogeneous origin at this altitude. However, there are positive biases along the ITCZ and over the TWP area. As the concentrations of new ice crystals produced by BN09 are not particularly remarkable in these regions (not shown), deep convection is likely to play a role. Indeed, there is a certain response of the convective activity to the choice of the ice nucleation scheme used in the cirrus regime."

**P11L13:** Is this only your speculation or do you have any evidence for it? Please show them! Please, see our previous answer (P11L11-12).

**P11L16-17: Why, please explain it!**

As answered to Referee #1 (P11L8-9, L16-17), we wrote:

"At 600 hPa, ICNCs increase towards high latitudes, in particular over Greenland (up to 2000  $L^{-1}$ ) and Antarctica (mostly > 2000  $L^{-1}$ ) (Figure 3e). It must be said

that, due to the very low temperatures in the the latter region, even at 600 hPa the conditions are typical of the cirrus regime, and the high ICNCs can be related to the high values of both  $w_{sub}$  and ice supersaturation. Gryspeerdt et al. (2017) found that cirrus clouds over Antarctica have primarily synoptic origin. However, differently from Figure 3e, observations do not present such a high peak of ICNC over Antarctica (Gryspeerdt et al. 2018; Sourdeval et al., 2018)."

**P11L18-19:** I agree, that is very interesting, and therefore would be nice to understand what caused it!**

We added some comments in the manuscript at lines 7-10, P10 because this can be seen already in Figure 2. Thus, we changed lines 18-20, P11 to:

"Figure 3f confirms what already noticed in Figure 2b, that is the ice nucleation scheme used in the cirrus regime affects the ICNC at the mixed-phase regime altitudes predicting higher ICNCs especially in the NH."

**P11L29-31:** What about the Intertropical Convergence Zone and peak of tropical convection in the Pacific warm pool area? Maritime aerosol cannot play a large role in a dynamically-driven detrained clouds.

Moreover, you also did not include marine aerosols in the model, so I don't understand why you mentioned them.

Why don't you look at your particle radius and verify if the model is giving reasonable values in the tropical Pacific?

The Referee is right. We did not consider marine aerosols as potential INPs. We removed the sentences at lines 29-31.

Although BN09 and KL02 produce different sizes of newly formed ice crystals, both schemes present the highest radius values over the TWP (at 200 hPa), as can be seen in Figure 2 of this document, and this impacts on the IWC of Figure 4 (left). Lines 28-31 were changed as follows.

"Nevertheless, two interesting features appear. First, the high IWC values (> 10 mg kg-1) over the TWP at 200 hPa, where ICNCs are not particularly high. This is probably caused by the bigger radii of the newly formed ice crystals simulated in this area, both by KL02 and BN09. Second ..."

**P11L29:** I guess 200 hPa is close to the level of maximum detrainment from deep convective clouds. It is therefore important to look at what size you assume for detrained ICs (I assume you use a 1-moment version of convective microphysics, so there needs to be more assumption to couple it to the stratiform microphysics).

Moreover, one of your coauthors showed how that the vertical velocities are quite high in the mentioned area (Barahona et al., 2017). I guess part of this is due to the prevailing large-scale ascent motion (quite noticeable in Joos et al., 2008), while indeed a lot of it has to be connected to deep convection, and, in GCM modeling world, to TKE values. Please explore that in larger detail!

We added some information about the CONVECT submodel and how it interacts with the CLOUD submodel in the revised manuscript as written at point P4. The sizes of detrained condensate particles are not computed explicitly and we cannot quantify them (as mentioned also at point P10).

The Referee is right, Barahona et al. (2017) in Fig. 4 show high standard deviations in vertical velocity over ITCZ and TWP, however, our  $w_{sub}$  (Figure 3-left of this document) is generally smaller worldwide. Thus, we cannot easily attribute the high values of IWC over the TWP area to TKE and so to deep convection. **P13L1:** Why is this the case? It would be extremely interesting to understand that, as this region plays a large role in global energy balance. Did you change the model tuning in between? Can this happen due to changes in convection, which somehow responds to a different cirrus scheme?

On page 15 you even give a hint for that: "When BN09 is used in the cirrus regime, Ptot grows by 4% especially because of the increase of the convective precipitation contribution (the large scale precipitation of all simulations remain almost constant)"

We deleted this sentence as IWC does not increase "dramatically" but increases where also ICNCs increase.

**Model comparisons and observations**

- **P13L4-9:** This text doesn't fit into the results section, please move it to model description! Done.
- **P13L24:** Please prove that a change of 7% is large by showing the variability (maybe add in table).

It must be stressed that 7% changes are based on global annual means. Although the difference is not statistically significant, it is still remarkable to have such a change on a global scale. We agree with the Referee that the sentence was not correctly formulated and we changed it removing the text "is quite sensitive to the ice scheme used". Following the Referee's suggestion, we added the variability (one standard deviation) of the calculated fields in Table 2.

**P13L28:** ... that applied ECHAM => that used ECHAM-HAM

Done.

**P15L15-19:** You say that BN09 makes larger IC, but large scale precipitation doesn't change. That's surprising. Why?

> These lines of the text refer to annual global means, so the sentence "the large scale precipitation of all simulations remain almost constant" is actually not appropriate because there are regional and local differences (as shown in Figure 4 of this document). This sentence has been removed.

P15L21-23: so all that hard work for nothing? Or what should I get from that?

We are not sure about what the Referee means here. With this sentence we want to point out that the biggest differences among the four simulations occur between the simulations which use different ice nucleation parameterizations in the cirrus regime, i.e. KL+LD and KL+BN are clearly different from BN+LD and BN+BN. This is what we refer to also at line 15, P16.

**P16L21-23:** Radiation changes for quite a bit, and this is probably a more important parameter for climate compared with ICNC, IWP, etc. I would more strongly point SW, LW, and NET CRE anomalies, maybe even show a lon x lat plot of them (with significance on it).

We included the global distributions of SCRE, LCRE, and NCRE (with levels of significance) in the supplement file and we changed lines 4-8, P15.

"Looking at the percentage changes and the global distributions in the supplement file (Figure S4) it is evident that the cloud radiative effects are sensitive to the ice nucleation scheme used for cirrus clouds. Indeed, SCRE increases more than 5% with BN09 because of the less efficient scattering of shortwave radiation by fewer and larger crystals. More importantly, LWCR decreases up to 15% in BN+LD because cirrus clouds, at the same, can trap less longwave radiation in the Earth-atmosphere system. As a result, NCRE diminishes with statistically significance over some areas in the tropics and high latitudes, and the cooling effect is enhanced."

**P16L11: Mention that you are talking about median values as means can be very different!**

In the caption of Figure 5 there are all the specifics about the plots (kind of statistics: median, kind of variable: in-cloud ICNC, spatial coverage, vertical coverage).

**P17L3-4:** Isn't that interesting, considering that BN09 should give comparable results to KL02 for homogeneous freezing, while BN09 has also PREICE and heterogeneous freezing effects included. So one would rather expect just the opposite, BN09 to be lower than KL02. Why do we see the opposite?

Are the results the same when comparing means instead of medians? Or do the vertical velocities calculated by the base model change for some reason between KL02 and BN09 schemes?

Unfortunately, Figure 5 in the manuscript is affected by an error made during the post-processing and has been replaced by Figure 1 of this document (see also Referee#1, P17L3-4). The results in the mixed-phase regime remain basically unchanged (right plot). In the cirrus regime (left plot), the simulations KL+LD and KL+BN undergo big differences at temperatures below 225 K, and the strong underestimation at very cold temperatures is not evident anymore. The simulations BN+LD and BN+BN show only slight changes which make them a bit closer to the observations (in the intervals 185-190 K and 202-226 K). We are sorry for the mistake. Now, at very cold temperatures ICNCs simulated using BN09 in the cirrus regime are lower than the ICNCs computed by KL+LD and KL+BN, as expected.

The text in Section 4.2 has been modified accordingly to the new Figure. Moreover, we mentioned some comparisons with other modeling studies (see also Referee #1, P17L3-4).

"Again, the simulations can be grouped in two sets according to the ice nucleation scheme used in the cirrus regime, i.e. KL+LD/KL+BN and BN+LD/BN+BN, because of their similarities. For most of the temperature range, the simulations which use KL02 in the cirrus regime overestimate the observed ICNCs (although they mostly remain below the 75th percentile). The overestimation of ICNCs is common to other modeling studies (e.g. Wang and Penner, 2010, Liu et al., 2012, and Shi et al., 2015) and especially in cold cirrus clouds (for T < 205 K). On the other hand, the simulations which use BN09 in the cirrus regime are very close to the observations at temperatures below 200 K and between 220 K and 230 K, while they underestimate ICNCs between 200 K and 220 K. In this temperature range the simulations can exceed the observed 25th percentile (although remaining within the 5th percentile). In comparison with the other two simulations, BN+LD and BN+BN always predict lower ICNCs at temperatures below 230 K, as expected because of the competition and PREICE effects. Finally, all four simulations overestimates ICNCs by one order of magnitude in the temperature range 230 – 240 K.

Overall, the simulations BN+LD and BN+BN agree particularly well with the measurements at temperatures lower than 200 K but underestimate the ICNCs within the interval 200 - 220 K, due to an overestimation of the competitive nucleation and PREICE effects. Barahona et al. (2010) showed that the competitive nucleation

effect is small using P13. Also, Liu et al. (2012) found that BN09 (using the parameterization of Phillips et al. 2008 for heterogeneous nucleation) and BNhom produced very similar results in the cirrus regime, suggesting that the competive nucleation effect was small because of the low ICNCs formed heterogeneously. Thus, we can deduce that the PREICE effect is the one which is likely overestimated in our simulations. Interestingly, modeled ICNCs do not show any particular trend, like also Kuebbeler et al. (2014) who used ECHAM-HAM. Differently, other studies found that ICNCs are inversely proportional with temperature, e.g. Liu et al. (2012) and Shi et al. (2015) with CAM5, indifferently if they used the ice nucleation scheme of Liu and Penner (2005) or BN09, and Barahona et al. (2010) with GEOS-5 and BN09. Such distinct behaviours are likely derived from the wide model variability in reproducing subgrid-scale processes, like vertical velocity, which play a role in ice nucleation. We reiterate that ICNC is highly dependent on the vertical velocity which is usually poorly represented in terms of spatial and temporal variability (Barahona et al., 2017). "

The means are higher than the medians (Figure 5-left of this document), but the results are similar: ICNCs simulated using BN09 in the cirrus regimes are lower than ICNCs simulated using KL02, and the latter ones overestimate the observations. Vertical velocity does not change between KL02 and BN09, a part for the PREICE correction in BN09.

**P17L4:** The comparison with aircraft data doesn't show BN09 as superior to the less realistic KL02, but rather the opposite. In particular, as there is only a small fraction of cirrus that reside at temperatures below 200 K (-73°C) in the area where most of the measurements come from (extratropics). Can you comment on that?

Please, see our previous answer (P17L3-4).

**P17L3-4:** Did you make sure you are comparing apples-to-apples? For instance, GCMs normally simulate cirrus in the winter polar stratosphere, which might be responsible for parts a non-negligible fraction of the distribution. Better to remove them from the analysis.

Also, did you normalize the model output based on the latitude not to give a too large meaning to the (numerous) polar gridpoints?

Actually, we did not simulate polar stratospheric clouds (moreover, the high latitudes are basically excluded from the analysis as the latitudinal coverage is 25S - 75N).

We did not normalized the model output. We did it in Figure 5-right of this document, taking into account the volumes of the different grid boxes, and we can observe that the results change only slightly. As the observations are not homogeneously distributed any weighted result would have the same flow of the original plot.

**P17L3-4:** It would be interesting to look at a plot of vertical velocity in function of temperature, if you believe that to be (part of) the reason for differences between KL02 and BN09.

Most of this section changed accordingly to the new Figure. Please, see our previous answer P17L3-4.

We did not plot the vertical velocity in function of temperature because now the ICNCs simulated by BN+LD and BN+BN are lower than KL+LD and KL+BN, as expected.

**P17L15-18:** You show a large underestimation only below 200 K, while between 200 and 210 K both schemes seem to be comparably bad (hint on problems with vertical velocities???).

Most of this section changed accordingly to the new Figure. Please, see our previous answer P17L3-4.

**P17L18:** Yes, but only at the very cold temperatures, which correspond only to a small fraction of cirrus and aren't the most relevant in terms of radiative (and in general climatic) impacts.

In the new manuscript also this sentence was changed.

P17L20-21: As far as I recall, CAM modeling community undertook some efforts to decrease the overestimation of cold cirrus ICNC. Within various realizations of ECHAM, as it seems like, we have just the opposite problem. Too few ICNC at coldest cirrus conditions. It is not intuitive at all that such problems are alleviated by implementing a scheme, which should on average decrease the ICNC. I would love to read a discussion on this point in the corrected manuscript.

Please, see our previous answer P17L3-4.

**P17L25-26:** This now sounds different to discussions from section 3.2 (Figure 3, results for 600 hPa). Why?

Rather than Section 3.2, this is different (the Referee is right) to what discussed about Figure 2c in Section 3.1. P13 actually produces less new ice crystals than LD06. The differences in Figure 5 (right) are small because of the "effect of smoothing" derived by averaging along the latitude, as we showed in our reply to Referee#1 (point P17L25-26). We deleted the assertion "... P13 and LD06 produce similar ICNC ..."

**P17L27-28:** Please add references for WISP-94 and ICE-L campaigns.

Done.

P17L30: I think there's some datasets out there that extend to mixed phase temperatures. Look for instance into Heymsfield et al., 2013: Ice Cloud Particle Size Distributions and Pressure-Dependent Terminal Velocities from In Situ Observations at Temperatures from -8 to -86°C.

> Thanks for the reference. We added the following sentence at the end of the paragraph.

> "Finally, ICNCs in Figure 5 (right) are in good agreement with the results of Heymsfield et al. (2013), also based on flight campaigns. They found that ICNCs decrease as temperature increases and are within the range 5-50  $L^{-1}$  in the mixed-phase regime. Besides the flight measurements, the recent ICNC estimates from lidar-radar satellite retrievals must be mentioned, e.g. Sourdeval et al. 2018 and Gryspeerdt et al. 2018. In particular, Gryspeerdt et al. 2018 analysed the behaviour of ICNCs within clouds as a function of temperature. Differently from Figure 5 (left), they showed that there is a weak temperature dependence of ICNC, which increases with decreasing temperature. On the other hand, similarly to Figure 5, they found a small increase of ICNC around 265–270 K and, interestingly, a small peak at about 233 K due to orographic and frontal regimes, which could explain our higher modeled ICNCs between 230 K and 240 K."

P17L30-32: That isn't true for the warmer of the mixed-phase clouds. Figure 11 of the referenced Kanji et al., 2017 paper schematically illustrates that ICNC can also be higher than INP numbers due to secondary ice processes.

We rephrased the sentence.

"It should be also noted that the measurements actually concern INPs. When the INP number is not high enough to deplete the ambient supersaturation, INP concentrations and ICNCs can correspond, however, it is well known that the two concentrations show discrepancies with increasing temperature because of the secondary ice formation (Kanji at al. 2017)."

**Conclusions**

**P18L15:** I would expect you to give a reason for the observed changes or lack of them after the line 15.

We provided the reasons at lines 5-8, P15. They are not new discoveries so we thought not to repeat them here in the Conclusions.

**P19L1-2:** This is not obvious from the data you show. Please, try to prove it in a quantitative way with the help of some appropriate statistical methods, or rephrase the conclusions!

The sentence changed (see also Referee #1, P17L3-4):

"Overall, all modeled results agree well with global observations and the literature data. The comparison made with flight measurements has pointed out that ICNCs are overestimated by KL02 in the cirrus regime. BN09 agrees well with the observations in cold cirrus clouds, however, the PREICE effect is likely overestimated causing the underestimation of ICNCs between 200 K and 220 K."

**P19L5-9:** Those are relatively weak conclusive words. Could you find some stronger statement on top of being able to include more processes in the model? Please think in the direction of why should anyone not using EMAC care about your manuscript (or consider citing it).

At line 13, P18, we added the sentence:

"We found that changing the ice nucleation scheme in the cirrus regime generates larger differences of ICNC and IWC than changing parameterization in the mixedphase regime, that is the simulations using the same parameterization in the cirrus regime (e.g. BN+LD and BN+BN) are easily discernible from the others (LD+KL and LD+BN). Interestingly, we also observed a certain dependence of ICNC and IWC in the mixed-phase regime on the parameterization used for cirrus clouds."

As pointed out by the Referee, it is expected that the EMAC Community will be interested in this paper more than others, however, this work will be useful for future model comparisons with focus on ICNC estimates.

The last paragraph was changed as follows (see also Referee #1, P17L3-4 and P19L5-7).

"Overall, all modeled results agree well with global observations and the literature data. The comparison made with flight measurements has pointed out that ICNCs are overestimated by KL02 in the cirrus regime. BN09 agrees well with the observations in cold cirrus clouds, however, the PREICE effect is likely overestimated causing the underestimation of ICNCs between 200 K and 220 K.

As BN09 takes into account additional processes which were previously neglected by the standard version of the model, without consuming extra computational resources, we recommend to apply this ice nucleation scheme in future EMAC simulations. We also suggest to select P13 among the INP parameterizations available in BN09, since it incorporates the ice-nucleating ability of different aerosol species (dust, soot, bioaerosols, and soluble organics) and simulates both deposition and immersion/condensation nucleation. By using the configuration BN+BN, the EMAC model becomes one of the few GCMs which take into account in a detailed manner the complexity of ice nucleation. Finally, this work offers further material for future GCM comparisons with focus on ICNC estimates and for future modeling evaluations against flight measurements and lidar-radar satellite retrievals."

**References**

- Jöckel, P., et al.: Earth System Chemistry integrated Modelling (ESCiMo) with the Modular Earth Submodel System (MESSy) version 2.51, Geosci. Model Dev., 9, 1153–1200, 2016.
- Klingmüller, K., et al.: Revised mineral dust emissions in the atmospheric chemistry-climate model EMAC (MESSy 2.52 DU Astithal KKDU2017 patch), Geosci. Model Dev., 11, 989-1008, 2018.
- Korolev, A.: Limitations of the Wegener-Bergeron-Findeisen Mechanism in the Evolution of Mixed-Phase Clouds, Journal of the Atmospheric Sciences, 64, 3372-3375, 2007.
- Lohmann, U. and Feichter, J.: Global indirect aerosol effects: a review, Atmos. Chem. Phys., 5, 715-737, 2005.
- McCoy, D. T., et al.L, On the relationships among cloud cover, mixed-phase partitioning, and planetary albedo in GCMs, J. Adv. Model. Earth Syst., 8, 650-668, 2016.
- Morales, R. and Nenes, A.: Characteristic updrafts for computing distribution-averaged cloud droplet number and stratocumulus cloud properties, J. Geophysical Res., 115, D18220, 2010.
- Morrison, H., et al.: Resilience of persistent Arctic mixed-phase clouds, Nature Geoscience, 5, 11-17, 2012.
- Waliser, D. E., et al.: Cloud ice: A climate model challenge with signs and expectations of progress, J. Geophysical Res., 114, D00A21, 2009.

Figure 1: New Figure 5. Modeled in-cloud ICNC and flight measurements versus temperature. Lines are medians of KL+LD (blue), BN+LD (green), KL+BN (light blue), BN+BN (red), and observations (black). Shaded areas indicate 5th-95th and 25th-75th percentiles of observations and BN+BN.

---

## Author Response (AR1)

Dear Editor,

in this document we report the modifications implemented in the manuscript in order to satisfy the Referees' comments. As the manuscript has undergone major revisions, only the main changes are listed below, but all changes (including minor/technical corrections) are visible in the marked-up manuscript version attached at the end of this document.

Yours faithfully,

Sara Bacer

**Main changes made in the manuscript**

The main changes regard Section 3 (Model results) and Section 4 (Model comparisons and observations). There are only few modifications in the Introduction and Section 2. The Abstract has been slightly changed to satisfy the Referee's comments. The Conclusions have been extended and slightly modified as well.

Along the whole manuscript, new references (suggested by the Referees) have been included. In the supplement file, the annual means of cloud radiative effects (SCRE, LCRE, NCRE) have been added.

**1 Introduction**

- The description of mixed-phase clouds has been extended.
- The issue about the "overestimation of vertical velocity" has been explained better.

**2 Model description and set-up of simulations**

**2.1 EMAC model**

- Information about the CONVECT submodel, how convective clouds are treated and interact with the CLOUD submodel has been provided.
- The description of the cloud droplet formation parameterization (UAF) has been added.

**2.3 Ice nucleation parameterization BN09**

- The way how BN09 output variables are weighted over a Gaussian updraft velocity distribution has been explained more extensively in Subsection 2.3.1 (Scheme characteristics) and removed from Subsection 2.3.2 (Implementation).

**2.4 Setup of simulations**

- The aerosol emissions used for the simulations have been explained more clearly.
- A better explanation why only dust and black carbon are used by P13 has been provided at the end of the Section.

**3 Model results**

The level of significance at 95% has been marked in all figures showing relative changes.

**3.1 Annual zonal means**

- In general, more explanations have been provided in the analysis of Figure 2.

- The observation that ICNCs in the mixed-phase regime are influenced by the ice nucleation scheme used for cirrus clouds has been added.

- The last paragraph regarding IWC has been improved with clearer explanations and comparisons with observations (new citations).

**3.2 Global distributions**

- Also here, more explanations have been provided in the analysis of Figure 3.
- The ICNC pattern in Figure 3a has been described more deeply. Some features have been linked to the turbulent contribution of the vertical velocity. The comparison with the recent lidar-radar satellite retrievals is also new.
- The biases in Figure 3b have been better analysed, and the possibile influence by convection has been mentioned.
- The issue of high ICNCs over Antarctica has been discussed more and compared with observations.
- The last paragraph regarding IWC has been changed to satisfy the Referees' comments.

**4 Model comparison and observations**

**4.1 Annual global means**

- Standard deviations have been included in Table 2.
- Two new variables ($ICNC_{burden,cirri}$ and $ICNC_{burder,mixed}$) have been added in Table 2 and described in the text.

**4.2 Comparison with aircraft measurements**

This Subsection has substantially changed.

- The text about the comparison with flight measurements in the cirrus regime has changed after the correction of Figure 5-left.
- The fact that the observations in the mixed-phase regime are actually of INPs (and not ICNCs) has been commented more accurately. Moreover, the comparison with a new reference (suggested by Referee#2) has been added.
- A new paragraph at the end mentions some recent works which report lidar-radar satellite retrievals of ICNCs and compares the modeled results with them.

**5 Conclusions**

- The Conclusions have been corrected accordingly to the changes made before in the manuscript.
- In particular, the last paragraph has been modified. The assertion that the model "performs generally better" has been deleted, and the importance of the model capability to simulate new processes after the implementation has been stressed.
- At the end, some new lines remark the importance of this work from a more general point of view.

[revised manuscript text omitted]

---

## Referee Report (RR1)

**Review of Bacer et al., round 2**

The authors have nicely responded to most of the questions and the manuscript is much clearer now. I think the paper is almost ready to be published. However, I still see one major issue, which the authors should urgently address (if not here then at least in future studies).  I also add a few more suggestions, which may improve bits of the manuscript.

Detraining ice water content, but not ice crystal number

Please point out the issue of detraining ice mass but not number from deep convective clouds. Maybe also add a note on it in the conclusions, as most readers do not carefully read all details of the model description.
I understand that detrainment and coupling with convection hasn't been the main focus of the study, however – given its importance for global climate, it should be mentioned in a clearer way somewhere in the text.
(where I take this point to acknowledge that the authors significantly improved the description of their model and its coupling to the detrainment)

The IC radius of more than 100 μm at the formation timestep (as shown by Figure 2 of the response document) is clearly not realistic at the 200 hPa level in areas dominated by detrainment. You also mentioned that at such levels you expect homogeneous nucleation to dominate, which is, again, not compatible with such large ice crystal radii in locations dominated by convective detrainment (e.g. Jensen et al., 2009) or very thin TTL cirrus clouds (e.g. Jensen et al., 2015). See also Heymsfield et al., 2014, Heymsfield et al., 2017 for a more general perspective. Naively, I could imagine that your nucleation scheme would take that into account and homogeneously nucleate enough ice crystals, but that clearly doesn't seem to be the case.

Many GCMs solve the problem by assuming the size of detrained ice crystals in order to get their number. This is also an oversimplification, which however prevents the formation of huge ice crystals and the decoupling between (large) ice water content and (relatively small) ice crystal number observed in your results with the BN09 scheme over the tropics. There might indeed be other, more elegant fixes of the detrainment number and large ice crystal problem.

**Minor comments**

General comment: ICNC changes
I am puzzled to see how little the ICNC burden for cirrus clouds changed. Why is that?
What confuses me is that you take the 200 hPa level as representative for cirrus. However, the ICNC changes there are significantly different compared with the integrated ICNC burden for cirrus.

I think you actually nicely prepared the answer to my question with Figure 4 in the response document to reviewer 1. I find the binning by temperature very informative. Could you add one in the paper and comment it briefly in the text? It would be great to see the anomalies of BN+LD, KL+BN, and BN+BN with respect to DEF. Also, I like its right panel too as it clearly summarizes the main ice crystal number changes (if you extended it to the lowest temperature range). I think extending the panel beyond 30°S, where the heterogeneous freezing doesn't occur too frequently, should also not affect its general interpretation Maybe this right panel with the extended temperature range alone would be enough to strengthen some of your conclusions, if you include it in the manuscript.

To sum up, I miss a discussion in section 3.1 (and conclusions) which is more clearly pointing at the (1) ICNC increase in mixed phase due to heterogeneous freezing, (2) ICNC increase around 230 K due to heterogeneous freezing in cirrus, (3) ICNC decrease at the coldest temperatures due to the PREICE effect. Is there a possibility the last mechanism is plausible from the text, however it would strengthen your findings.

[Figure]

Page 2, lines 31-33:
Homogeneous nucleation is dominant? Yes, but based on some modelling studies. Not all of them. At least that's what one of the authors nicely showed in a previous study (see plot below from Barahona et al., 2017). Of course, I would agree that homogeneous nucleation is the dominant source of ice crystal number or similar statements, but not when looking at ice crystal nucleation events.

**Figure 8.** Monthly mean homogeneous ice nucleation frequency for the Tropical (latitude −30° to 30°) and the Northern (NH, latitude 30° to 60°), and Southern (SH, latitude −30° to −60°) extratropical regions.

Page 4, line 33:
-You referenced the wrong Tost et al., 2006 paper (looking at the references)
-Do you use the scheme of Tiedtke 1989 or Tiedke 1989 with modifications from Nordeng 1994? There are some significant differences between the two, so it is better to be precise here.

Page 7, line 20:

*"The only expedient adopted by the CLOUD submodel is…."*
That word (expedient) does not make sense in the context used. Please use simple and understandable words instead.

Page 11, line 18:

Cziczo et al., 2009 is not a good reference for northern hemisphere, being dominated by heterogeneous nucleation, as it talks/speculates about lead-containing natural dust. Li et al., 2012, Storelvmo et al., 2014, Gasparini et al., 2016 and also Barahona et al., 2017 show some of that, although being highly model and parameterization dependent.

Page 12, line 29-30:
*"Such a reduction…at this altitude."* please rephrase this sentence as it is not grammatically correct.

Page 14, line 13-14:
Could you add a sentence explaining why the mixed-phase is less sensitive to the change in ice nucleation scheme?

Page 14, line 19-20
These mentioned ice crystals are simply too large compared to any kind of observations at such levels of the atmosphere.

Page 16, lines 7-9:
I would either rephrase this or remove the reference to Lohmann et al. 2008 as the sentence gets very hard to understand, when explaining that Lohmann et al. 2008 did not include the competition effects.

Page 16, line 10:
also Gasparini et al., 2018 for a recent ECHAM-HAM paper showing IWP anomalies

Page 16, line 11:
add a reference for your liquid water path estimates

Page 16, line 13:
Duncan and Eriksson 2018 might be a fresh good reference for IWP variations between observation and reanalysis datasets.

Page 16, line 19-20:
SCRE becomes less strong. I would not call that an "increase", despite its value becoming less negative.
For me an increase in SCRE would imply a stronger reflection of clouds and consequently a more negative SCRE.

Page 16, line 21:
*NCRE diminishes*
please rephrase also here to a more clear expression – maybe "NCRE becomes more negative"

Page 17, line 5-8:
If your scheme produces fewer and larger ice crystals that indeed might lead to increased precipitation. However, this will contribute to the large-scale precipitation budget, and not to the convective one.

Your interpretation of the result is therefore not explaining the increase in convective precipitation.

Page 17, line 10-13:
I would argue that the changes in LCRE and SCRE are quite considerable and not negligible (as you also mentioned earlier in the text).

Figure S4:
I would suggest to rather plot the radiative anomalies in absolute terms, using $W/m^2$ as a unit. Relative changes are tricky to interpret, in particular when looking at SCRE (and NCRE).

Table 2:
Why do you compute also a spatial standard deviation when showing globally averaged results? Please use just global annual means, and get the (temporal) standard deviation from those 5 points (I agree 5 points are rather at the lower limit, but that is certainly more meaningful than looking at variability in both space and time).

References

Jensen et al., 2009: On the importance of small ice crystals in tropical anvil cirrus
Jensen et al., 2015: The NASA airborne Tropical Tropopause Experiment
Heymsfield et al., 2014: Relationships between ice water content and volume extinction coefficient from in situ observations for temperatures from -0 to -86°C: Implications for spaceborne lidar retrievals.
Heymsfield et al., 2017: Cirrus clouds
Storelvmo et al., 2014: Cirrus cloud susceptibility to the injection of ice nuclei in the upper troposphere
Duncan and Eriksson, 2018: An update on global atmospheric ice estimates from satellite observations and reanalyses
Gasparini et al., 2018: Cirrus cloud properties as seen by the CALIPSO satellite and ECHAM-HAM global climate model

---

## Author Response (AR2)

Dear Editor,

in this document we report our point-by-point replies to the anonymous Referee and your comments, and the marked-up version of the manuscript (attached at the end).

We have implemented all the minor comments of the Referee and yours. We did not address completely the comments about the "detrained ice" issue raised by the Referee as it goes beyond the scope of this manuscript.

Best regards,

Sara Bacer

**Reply to the Referee's comments**

We really appreciate the careful work of revision carried out by the Referee and we are grateful for the helpful comments. Below there are our replies.

**Major Issue**

*Detraining ice water content, but not ice crystal number.*

We explained the issue raised by the Referee in Subsection 2.1 (EMAC model), between L1 and L9 P5. We notice that we did not specify that the CLOUD submodel assumes a temperature dependent radius to estimate the ice crystal number from ice mass detrained from convection. Thus, we substituted the sentence at L10, P5 ("The size and numbers of the detrained condensate are not taken into account explicitly") with:
"The number of detrained ice crystals is estimated from the ice condensate detrained from convection by assuming an only temperature dependent radius."
Nevertheless, the ice crystal size and the detrainment from convection such as the distinction of liquid vs. in-situ formed ice go beyond the scope of this manuscript and will be subject of future studies.
We also changed the sentence at L19-20 P14 from: "This is probably caused by the bigger radii of the newly formed ice crystals simulated in this area, both by KL02 and BN09" to "This is probably caused by the larger radius of ice crystals simulated in this area" since we cannot exclude that also the radius of detrained ice crystals is too large (unfortunately, this radius is not an output of our simulations).

**Minor Comments**

*General comment: ICNC changes.*

- *I am puzzled to see how little the ICNC burden for cirrus clouds changed. Why is that? What confuses me is that you take the 200 hPa level as representative for cirrus. However, the ICNC changes there are significantly different compared with the integrated ICNC burden for cirrus.*

  ICNC$_{burden}$ is the vertical integration of ice crystal number concentrations derived from different sources, ice nucleation but also convective detrainment, instantaneous freezing, and secondary ice production. On the other hand, ICNC at 200 hPa is mainly produced by ice nucleation, and partially by convective detrainment and instantaneous freezing (we do not to show plots about this because they are part of a future work). Thus, it is correct to expect that ICNC$_{burden}$ changes are different from ICNC changes (which are mainly due to the different ice nucleation schemes used in the cirrus regime).

- *I think you actually nicely prepared the answer to my question with Figure 4 in the response document to reviewer 1. I find the binning by temperature very informative. Could you add one in the paper and comment it briefly in the text? It would be great to see the anomalies of BN+LD, KL+BN, and BN+BN with respect to DEF. Also, I like its right panel too as it clearly summarizes the main ice crystal number changes (if you extended it to the lowest temperature range).*
  *I think extending the panel beyond 30S, where the heterogeneous freezing doesn't occur too frequently, should also not affect its general interpretation. Maybe this right panel with the extended temperature range alone would be enough to strengthen some of your conclusions, if you include it in the manuscript.*

We plotted the annual zonal means as functions of latitude and temperature and the absolute changes (extending the lower temperature range) as requested by the Referee, see Figure 1 of this file. The last row of plots considers the averages computed over different latitudinal coverages: 90S – 90N and 25S – 27N (i.e. the same considered in the comparison with flight measurements in Figure 5 of the manuscript). Indeed, the absolute changes show a similar behaviour (as said by the Referee). Since Figure 1 is equivalent to Figure 2 of the manuscript, we added it in the supplement file and wrote the following sentences:

"The absolute changes of ICNC annual zonal means computed as a function of latitude and temperature (Figure S1 in the supplement file) show that ICNCs in BN+KL are lower than the default case by 300 $L^{-1}$ at temperatures below 220 K." (at L13 P11 of the manuscript).

"This is also evident from Figure S1 in the supplement file (last row), where the absolute changes are, in average, between 200 and $-200$ $L^{-1}$ when BN09 is used in the cirrus regime and between 50 and $-50$ $L^{-1}$ when comparing KL+BN with KL+LD." (at L3 P12 of the manuscript).

- *To sum up, I miss a discussion in section 3.1 (and conclusions) which is more clearly pointing at the (1) ICNC increase in mixed phase due to heterogeneous freezing, (2) ICNC increase around 230 K due to heterogeneous freezing in cirrus, (3) ICNC decrease at the coldest temperatures due to the PREICE effect. Is there a way you could prove the last point? It sounds plausible from the text, however, a more direct proof would strengthen your findings.*

We addressed the first point at L19-21 P11.
Regarding the second point, we did not write explicitly about this ICNC increase which is close to the transition between the two cloud regimes. However, as the heterogeneous contribution is very small in comparison with the homogeneous nucleation (not shown), it is more likely that the ICNC increase is due to a faster sedimentation of larger IC from higher altitudes, as written at L20-21 P11.
We already addressed the third point at L14-15 P11 and later in Subsection 4.2 (P18). To prove that ICNC decreases at lower temperatures because of the PREICE effect, we compared two simulations (using the BN+BN configuration) with and without the PREICE effect. The difference has a pattern really similar to Figure 2d (ICNC: BN+BN vs DEF). Therefore, we have a confirmation that the differences at those altitudes/temperatures are due to the PREICE effect. This information will be part of a future study.

**P2 L31-33:** *Homogeneous nucleation is dominant? Yes, but based on some modelling studies. Not all of them. At least that's what one of the authors nicely showed in a previous study (see plot below from Barahona et al., 2017). Of course, I would agree that homogeneous nucleation is the dominant source of ice crystal number or similar statements, but not when looking at ice crystal nucleation events.*

We slightly changed the sentence and added some citations:
"In several modeling studies, homogeneous nucleation has been considered the dominant process for cirrus formation (e.g. Haag et al., 2003; Hendricks et al. 2011; Gettelman et al., 2012; Barahona et al. 2014) because ... "

**P4 L33:** *You referenced the wrong Tost et al., 2006 paper (looking at the references). Do you use the scheme of Tiedtke 1989 or Tiedke 1989 with modifications from Nor-*

*deng 1994? There are some significant differences between the two, so it is better to be precise here.*

Thanks for noticing it. We corrected the reference in the revised version.
We used the Tiedtke (1989) scheme with modifications by Nordeng (1994). We specified it at L33 P4:
"In this work the scheme of Tiedtke (1989) with modifications by Nordeng (1994) has been used."

**P7 L20:** *"The only expedient adopted by the CLOUD submodel is..." That word (expedient) does not make sense in the context used. Please use simple and understandable words instead.*

We made the sentence simpler: "Finally, the influence of the pre-existing ice particles is not taken into account. The CLOUD submodel simply reduces the number of aerosol particles available for ice nucleation by the existing ice particle number in the cirrus regime".

**P11 L18:** *Cziczo et al., 2009 is not a good reference for northern hemisphere, being dominated by heterogeneous nucleation, as it talks/speculates about lead-containing natural dust. Li et al., 2012, Storelvmo et al., 2014, Gasparini et al., 2016 and also Barahona et al., 2017 show some of that, although being highly model and parameterization dependent.*

Thanks for pointing it out. We cited: "Li et al., 2012; Kuebbeler et al. 2014; Storelvmo et al., 2014; Shi et al. 2015; Gasparini et al., 2016; Barahona et al., 2017".

**P12 L29-30:** *"Such a reduction occurs mostly because of the PREICE effect, being the ice crystals mainly of homogeneous origin at this altitude" Please rephrase this sentence as it is not grammatically correct.*

We changed the sentence to: "As the ice crystals are mainly of homogeneous origin at this altitude, such a reduction is probably due to the PREICE effect."

**P14 L13-14:** *Could you add a sentence explaining why the mixed-phase is less sensitive to the change in ice nucleation scheme?*

In the model, ICNC is the sum of tendencies associated to different processes, such as sedimentation, secondary ice production, aggregation, ice nucleation. A possible explanation is that the rate associated to heterogeneous nucleation has a low effect, in comparison to the other processes, on the final ICNC. Thus, ICNC is only slightly perturbed from the choice of the ice nucleation scheme used in the mixed-phase regime.
We added the following sentence in the manuscript at L3 P12:
"Possibly, the rate associated to heterogeneous nucleation in the mixed-phase regime is masked by other processes, like sedimentation and aggregation, which also contribute to ICNC in this regime."
Moreover, we slightly changed L13 P14, from:
"In general, we find that the ICNC in the mixed-phase regime is less sensitive to the ice nucleation scheme changes than the ICNC in the cirrus regime" to
"Thus, we can reiterate that the ICNC in the mixed-phase regime is less sensitive to the ice nucleation scheme changes than the ICNC in the cirrus regime."

**P14 L19-20:** *These mentioned ice crystals are simply too large compared to any kind of observations at such levels of the atmosphere.*

Please, see our first reply. The ice crystal size goes beyond the scope of this manuscript and will be subject of future studies.

**P16 L7-9:** *I would either rephrase this or remove the reference to Lohmann et al. 2008 as the sentence gets very hard to understand, when explaining that Lohmann et al. 2008 did not include the competition effects.*

We removed the second citation: "The ice water path ($IWP$) decreases by almost 7% when BN09 is used in the cirrus regime, similarly to what has been found in Kuebbeler et al. (2014), who compared simulations assuming pure homogeneous nucleation against simulations including water vapour competition."

**P16 L10:** *Also Gasparini et al., 2018 for a recent ECHAM-HAM paper showing IWP anomalies.*

We cited Gasparini et al. 2018.

**P16 L11:** *Add a reference for your liquid water path estimates.*

The LWP values are actually taken from Table 2, and the references are written in the caption of the Table.
We specified "..., between 23 and 87 $g/m^2$ (Li et al. 2012 and Han et al. 1994)."

**P16 L13:** *Duncan and Eriksson 2018 might be a fresh good reference for IWP variations between observation and reanalysis datasets.*

We modified the sentence at L9-10 P16: "Overall, the model underestimates the IWP, also found in other studies that applied ECHAM-HAM (...), however, there are still large discrepancies among observational datasets which make problematic the validation of the models (Duncan and Eriksson 2018)."

**P16 L19-20:** *SCRE becomes less strong. I would not call that an "increase", despite its value becoming less negative. For me an increase in SCRE would imply a stronger reflection of clouds and consequently a more negative SCRE.*

It sounds actually better. We replaced "SCRE increases more than 5% with BN09 because of..." with "SCRE with BN09 becomes weaker (more than 5%) because of..."

**P16 L21:** *"NCRE diminishes". Please rephrase also here to a more clear expression – maybe "NCRE becomes more negative".*

We changed the sentence to: "As a result, NCRE becomes more negative, with statistically significance over..."

**P17 L5-8:** *If your scheme produces fewer and larger ice crystals that indeed might lead to increased precipitation. However, this will contribute to the large-scale precipitation budget, and not to the convective one. Your interpretation of the result is therefore not explaining the increase in convective precipitation.*

The Referee is right. Our results about precipitation shows that the changes are mostly caused by the convective precipitation changes, but the sentence at L5-7 P17 does not explain this. As written at L32-33 P12, there are some feedbacks in the convection activity due to the ice nucleation scheme used. Thus, we wrote:
"When BN09 is used in the cirrus regime, $P_{tot}$ grows by 4% especially because of the increase of the convective precipitation contribution, due to some feedbacks on the convective activity generated by the different ice nucleation schemes used, as mentioned in Subsection 3.2."

**P17 L10-13:** *I would argue that the changes in LCRE and SCRE are quite considerable and not negligible (as you also mentioned earlier in the text).*

Actually, we wrote the changes in percentages (5% and 15%) at L14-22 P16 and that "There are clear effects on SCRE and LCRE from changing the cirrus ice nucleation scheme" at L13-15 P17. In order to highlight this sentence, we changed it to:
"Mostly, the experiments do not yield evident differences among each other at the global scale, as regional variations may cancel out, however, there are clear effects on SCRE and LCRE from changing the cirrus ice nucleation scheme."

**Figure S4:** *I would suggest to rather plot the radiative anomalies in absolute terms, using $W/m^2$ as a unit. Relative changes are tricky to interpret, in particular when looking at SCRE (and NCRE).*

Done.

**Table 2:** *Why do you compute also a spatial standard deviation when showing globally averaged results? Please use just global annual means, and get the (temporal) standard deviation from those 5 points (I agree 5 points are rather at the lower limit, but that is certainly more meaningful than looking at variability in both space and time).*

We changed the standard deviations following the Referee's suggestion.

[Figure]

Figure 1: New Figure for the supplement file. Annual zonal mean of (grid-averaged) ice crystal number concentration (ICNC, $[L^{-1}]$) for the default simulation KL+LD and the absolute changes of BN+LD, KL+BN, and BN+BN with respect to it as functions of latitude and temperature. (*Last row*) Absolute changes of annual zonal means averaged along the latitude 90S – 90N (left) and 25S – 75N (right): (BN+LD)-(DEF) in green, (KL+BN)-(DEF) in cyan, and (BN+BN)-(DEF) in red. Only the plot on the left has been added in the supplement.

**Reply to the Editor's comments**

We really thank the Editor for his wok and his helpful comments. Below we report our replies.

1. *I read multiple times the term cold/warm temperature which is a common mistake in cloud physics literature. Temperatures are high or low and tell whether an air parcel or some other material is warm or cold. Please change to low/high temperature.*

   Done.

2. *A bin model may explicitly compute the size distribution, but a two moment scheme certainly does not. Please reformulate p.5, l.25.*

   Thanks for pointing it out. We changed the sentence to:
   "The advantage of using a two-moment scheme is that it allows aerosol-cloud interactions improving calculations of cloud microphysical processes and radiative transfer."

3. *p.8, l.20 "required to KEEP (?) s_ max below s_ hom."*

   Done.

4. *p.10, l.19: I guess what you want to say is that "the standard deviations are constants depending only on the mode". The word "differentiated" is misleading.*

   Yes. We changed to: "the standard deviations $\sigma_M$ are constant depending only on the mode ..."

5. *Sometimes you use the word "to comment" where I would favour the word "to remark" (e.g. p.11, l.25).*

   We used "remarked" at L25 P11 and L1 P13.

6. *p.12, l.5: as a function of LATITUDE AND altitude.*

   Done.

7. *replace big by large: p.14,l.33; p.15, l.24*

   Done.

8. *p.17, l.5: "(in total, 113 flights with about ...)"*

   Done.

9. *Add short explanation after p.20, l.12 why cirrus param affects mixed phase cloud properties.*

[revised manuscript text omitted]

---

## Author Response (AR3)

In this short notice the authors would like to thank the Editor and to inform him that some affiliations have been updated in the last uploaded version of the manuscript.

Best regards,

Sara Bacer